# Sleeping Reinforcement Learning

**Simone Drago** [* 1]  **Marco Mussi** [* 1]  **Alberto Maria Metelli** [1]

## Abstract

In the standard Reinforcement Learning (RL) paradigm, the action space is assumed to be fixed and immutable throughout the learning process. However, in many real-world scenarios, not all actions are available at every decision stage. The available action set may depend on the current environment state, domain-specific constraints, or other (potentially stochastic) factors outside the agent's control. To address these realistic scenarios, we introduce a novel paradigm called *Sleeping Reinforcement Learning*, where the available action set varies during the interaction with the environment. We start with the simpler scenario in which the available action sets are revealed at the beginning of each episode. We show that a modification of UCBVI achieves regret of order $\widetilde{\mathcal{O}}(H\sqrt{SAT})$, where $H$ is the horizon, $S$ and $A$ are the cardinalities of the state and action spaces, respectively, and $T$ is the learning horizon. Next, we address the more challenging and realistic scenario in which the available actions are disclosed only at each decision stage. By leveraging a novel construction, we establish a minimax lower bound of order $\Omega(\sqrt{T2^{A/2}})$ when the availability of actions is governed by a Markovian process, establishing a statistical barrier of the problem. Focusing on the statistically tractable case where action availability depends only on the current state and stage, we propose a new optimistic algorithm that achieves regret guarantees of order $\widetilde{\mathcal{O}}(H\sqrt{SAT})$, showing that the problem shares the same complexity of standard RL.

## 1. Introduction

In recent years, Reinforcement Learning (RL, Sutton & Barto, 2018) has demonstrated remarkable success in solv-

ing sequential decision-making problems mainly in simulated environments. Nowadays, we witness an increasing demand to transition its capabilities from simulation to real-world applications. However, to move RL to practical domains, we still experience notable challenges that need to be addressed from both the algorithmic and theoretical perspectives. One of these challenges concerns *actions' availability*. Indeed, in the standard RL framework, the set of actions that can be played in a given state is assumed to be known and immutable throughout the interaction. However, in several real-world sequential decision-making problems, it may not be possible to play some of the actions under some circumstances.

**Motivation.** Consider the scenario depicted in Figure 1a, in which we want to control a physical system (e.g., a robot) characterized by a given action space $\mathcal{A}$ made of all the actions the agent can perform. At every stage $h$, the RL agent observes the state of the environment $s_h$ and decides the action $a_h \in \mathcal{A}$ to play. Then, such an action is usually validated by a *low-level controller* (LLC), which checks whether the action is feasible or not (depending, for instance, on some physical constraints or safety reasons). On the one hand, if feasible, the action is executed as is. On the other hand, if the action is not feasible, the low-level controller overrides it with another action, $\widetilde{a}_h \in \mathcal{A}$, such as one suggested by a baseline policy or the "closest" feasible action (according to some domain-specific metric). From the agent's perspective, the LLC can be considered as a part of the environment. However, when the agent is unaware that certain actions are infeasible (and may only realize it *after* the LLC intervenes), the performance of the learned policy can be significantly harmed (see Example 1). If the agent is instead aware of the available actions, we ideally want it to select an action among those deemed feasible by the LLC. This scenario can be addressed by adopting a solution in which a *low-level filter* makes the RL agent aware of which actions $\mathcal{A}_h \subseteq \mathcal{A}$ are available *before* actually making a decision (Figure 1b).

This problem of learning in scenarios with varying action availability is widely studied and discussed for Multi-Armed Bandits (MABs, Lattimore & Szepesvári, 2020) under the name of "Sleeping" MABs. This research line comprises stochastic and adversarial choices for both rewards and action availability (Kleinberg et al., 2008; Kanade et al.,

---

*Equal contribution [1]Politecnico di Milano, Milan, Italy. Correspondence to: Simone Drago <simone.drago@polimi.it>.

*Proceedings of the 42$^{nd}$ International Conference on Machine Learning*, Vancouver, Canada. PMLR 267, 2025. Copyright 2025 by the author(s).

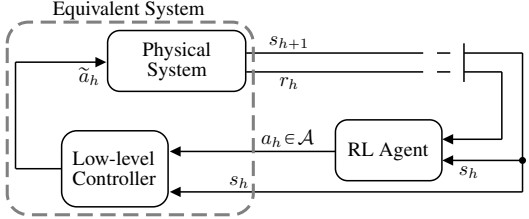

(a) *Classic* Reinforcement Learning.

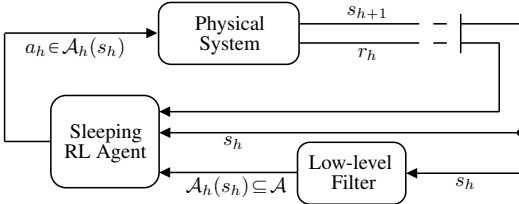

(b) *Sleeping* Reinforcement Learning.

*Figure 1.* Example of a possible interaction protocol for *Classic* and *Sleeping* Reinforcement Learning.

2009; Saha et al., 2020; Nguyen & Mehta, 2024) and several notions of regret (Gaillard et al., 2023). In RL, instead, this problem remains unaddressed despite its practical relevance.[1] This work aims to fill such a gap in the literature.

**Original Contributions.** In this paper, we propose the novel paradigm of *Sleeping Reinforcement Learning* (SleRL) and study it from a theoretical perspective. The contributions of this work are summarized as follows:

- In Section 2, we formally introduce the framework of the episodic finite-horizon *Sleeping Markov Decision Processes* (*SleMDPs*) setting, a generalization of MDPs in which the available action set changes throughout the interaction. Then, we introduce the two types of action set disclosure, namely *per-episode* and *per-stage* disclosures. Finally, we present two stochastic models governing the action availability, namely *Markovian* and *independent*.
- In Section 3, we consider the simpler case of *per-episode disclosure*, where the agent is informed of the available action sets *at the beginning of each episode*. We introduce the definitions of value function, optimality, and regret. Furthermore, we present `Action-Restricted UCBVI` (`AR-UCBVI`), an algorithm based on `UCBVI` (Azar et al., 2017) and analyze its regret, showing that it matches the lower bound of standard RL up to logarithmic factors for sufficiently large $T$ (Theorem 3.2).
- In Section 4, we address the more realistic and challenging setting of *per-stage disclosure*, where the available action set is revealed *for the current stage only*, imme-

diately before the agent selects an action. We address the general scenario in which the action availabilities are governed by a Markovian process and illustrate how the problem can be framed as solving an *augmented MDP* in which the available action set is incorporated in the state. Based on this transformation, we define the value function, optimality, and regret. Then, through a novel construction (Figure 3), we demonstrate a statistical barrier of this setting, showing that an exponential dependence on the number of actions $A$ is unavoidable in the regret, proving a lower bound of order $\Omega(H\sqrt{SAT2^{A/2}})$ (Theorem 4.2).

- In Section 5, we turn to the tractable case of *independent per-stage disclosure*, where the action availabilities are sampled independently at every stage. We propose a novel optimistic algorithm, `Sleeping UCBVI` (`S-UCBVI`), that extends the classical `UCBVI` with the estimate of the action availability probabilities and appropriately defined new bonuses. We show that `S-UCBVI` enjoys a regret bound of order $\widetilde{\mathcal{O}}(H\sqrt{SAT})$ for sufficiently large $T$ (Theorem 5.2), matching the lower bound, up to logarithmic factors.

Related works are discussed in Appendix B. Omitted proofs are provided in Appendices C and D. A numerical validation of the work is provided in Appendix E.

## 2. The Sleeping MDPs Setting

In this section, we first introduce the notation and the standard episodic finite-horizon MDP setting, and then we present the novel *Sleeping MDPs* (SleMDPs) framework.

**Notation.** Given $a, b \in \mathbb{N}$ with $a < b$, we define $[\![a, b]\!] := \{a, a+1, \dots, b\}$ and $[\![a]\!] := [\![1, a]\!]$. Given a finite set $\mathcal{X}$, we denote as $\Delta(\mathcal{X})$ the probability simplex over $\mathcal{X}$, with $|\mathcal{X}|$ its cardinality and with $\mathcal{P}(\mathcal{X})$ its power set. Let $q \in \Delta(\mathcal{X})$, we denote its support as $\operatorname{supp}(q) = \{x \in \mathcal{X} : q(x) > 0\}$.

**Markov Decision Processes.** We define a finite-horizon undiscounted MDP as a tuple $\mathcal{M} := (\mathcal{S}, \mathcal{A}, P, R, H, \overline{s})$, where $\mathcal{S}$ and $\mathcal{A}$ are the state and action spaces, respectively, $H$ is the horizon of the episode, $P : \mathcal{S} \times \mathcal{A} \times [\![H]\!] \to \Delta(\mathcal{S})$ is the stage-dependent transition probability distribution, $R : \mathcal{S} \times \mathcal{A} \times [\![H]\!] \to [0, 1]$ is the deterministic reward function, assumed to be known,[2] and $\overline{s} \in \mathcal{S}$ is the initial state. We assume the state space and the action space are finite sets, and we denote their cardinalities as $|\mathcal{S}| =: S < +\infty$ and $|\mathcal{A}| =: A < +\infty$. The agent's behavior is modeled with a Markovian policy $\pi : \mathcal{S} \times [\![H]\!] \to \Delta(\mathcal{A})$. The agent interacts with the environment for $K$ episodes of length $H$, and we denote with $T = KH$ the total number of decisions.

**Sleeping Markov Decision Processes.** A SleMDPs is

---

[1]The scenario in which not all the actions are available for every state in a *deterministic* manner is discussed under the name of *action masking* (Huang & Ontañón, 2022), see Appendix B.

[2]This is a mild assumption that can be removed with no additional complexity, as learning the transition probability $P$ is more challenging than learning the reward function $R$.

---

**Algorithm 1:** Interaction Protocol — Per-episode.

1 **for** $k \in [\![K]\!]$ **do**
2     Agent observes $\mathcal{A}_{k,h}(s), \ \forall h \in [\![H]\!], s \in \mathcal{S}$
3     **for** $h \in [\![H]\!]$ **do**
4        Agent observes state $s_{k,h}$
5        Agent plays $a_{k,h} \in \mathcal{A}_{k,h}(s_{k,h})$
6        Environment returns $r_{k,h}$ and $s_{k,h+1}$
7     **end**
8 **end**

---

**Algorithm 2:** Interaction Protocol — Per-stage.

1 **for** $k \in [\![K]\!]$ **do**
2     **for** $h \in [\![H]\!]$ **do**
3        Agent observes state $s_{k,h}$
4        Agent observes $\mathcal{A}_{k,h}$
5        Agent plays $a_{k,h} \in \mathcal{A}_{k,h}$
6        Environment returns $r_{k,h}$ and $s_{k,h+1}$
7     **end**
8 **end**

---

defined as a tuple $\mathcal{M} := (\mathcal{S}, \mathcal{A}, C, P, R, H, \overline{s})$, where $(\mathcal{S}, \mathcal{A}, P, R, H, \overline{s})$ is standard MDP as defined above, and $C$ is the action availability model, that will be characterized later. Formally, for every episode $k \in [\![K]\!]$, at every stage $h \in [\![H]\!]$, for the current state $s_{k,h} \in \mathcal{S}$, the available action set $\mathcal{A}_{k,h}(s) \subseteq \mathcal{A}$ is an element of the power set $\mathcal{P}(\mathcal{A})$, selected according to the model $C$. We assume that at least one action can always be played (i.e., $\mathcal{A}_{k,h}(s_{k,h}) \neq \{\}, \forall k \in [\![K]\!], h \in [\![H]\!], s_{k,h} \in \mathcal{S}$).

**Action Disclosure.** The available actions can be revealed to the agent either at the beginning of the current episode for every stage and state, i.e., *per-episode disclosure*, or when the agent is asked to choose an action in the current stage and state, i.e., *per-stage disclosure*. The two interaction protocols are presented in Algorithms 1 and 2, respectively.

**Example 1.** *To illustrate the effect of the* per-episode *(PE) and* per-stage *(PS) disclosures of action availability and the effect of a* low-level controller *(LLC), we consider the MDP in Figure 2a, whose goal is to go from initial state A to the absorbing state C. We have two paths to do so. First, we can play action $F_s$ (i.e., forward safe) in state A, which deterministically leads to C with a reward of $-2$. Second, we can play action $F$ (i.e., forward) in state A and deterministically reach B without costs ($r = 0$). Then, in state B, for all subsequent stages, action $F$ is available with probability $p \in [0,1]$. If $F$ is not available, we can voluntarily stay in B and wait (i.e., play action $S_v$), obtaining a reward of $-1$. In state B, there is also another action $S_f$ (i.e., stay forced), that makes the agent remain in state B and receive a reward of $-2$. Action $S_f$ is employed by the LLC to override forbidden actions (i.e., the attempt to play $F$ when it is not available). We now compute and plot as a function of $p$ (Figure 2b) the optimal value functions for*

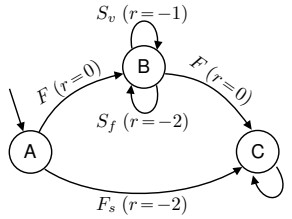

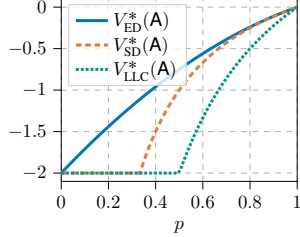

(a) Illustrative Sleeping MDP.    (b) Optimal value functions as a function of $p$.

*Figure 2.* Illustrative example.

*PE and PS disclosure and for LLC (see Appendix A for a complete discussion):*

- *PE **disclosure**. Since we know whether action $F$ is available in state B at the beginning of the episode, we play $F$ in state A if action $F$ is available in state B in stages $h \in \{2,3\}$ getting either reward $0$ or $-1$ (playing $S_v$ in stage $h = 2$); otherwise we play $F_s$ getting $-2$ as reward.*

- *PS **disclosure**. Here, we do not know if action $F$ will be available in state B. If we play $F$ in state A, we may wait several stages in B playing action $S_v$ until $F$ becomes available and getting $-(1-p)/p$ as expected total reward. Instead, if we play $F_s$ in state A, we get $-2$ as reward.*

- *LLC. Here, when we are in state B, we do not observe if action $F$ is available. Since the LLC overrides $F$ with $S_v$ when $F$ is not available, if we decide to go through state B, we will get $-2(1-p)/p$ as expected total reward; otherwise, by playing $F_s$ in state A, we get $-2$ as reward.*

*As expected, the value functions are sorted as: $V_{PE}^*(A) \geq V_{PS}^*(A) \geq V_{LLC}^*(A)$, where $V_{PE}^*(A), V_{PS}^*(A)$, and $V_{LLC}^*(A)$ represent the optimal state value function in state A in the PE, PS, and LLC cases, respectively (see also Section 6).*

**Action Availability Models.** We admit the available action sets to be chosen in a *stochastic* way as follows:

- *Independent* (referred as ind) action availabilities: we define $C = C^{\mathrm{ind}} : \mathcal{S} \times [\![H]\!] \to \Delta(\mathcal{P}(\mathcal{A}))$. For every action subset $\mathcal{B} \subseteq \mathcal{A}$, state $s \in \mathcal{S}$, and stage $h \in [\![H]\!]$, $C_h^{\mathrm{ind}}(\mathcal{B}|s) = \Pr(\mathcal{A}_{k,h} = \mathcal{B}|s_{k,h} = s)$ represents the probability that $\mathcal{B}$ is the available action set in state $s$ at stage $h$. Notably, the availability of an action does not depend on whether it was available in the past.

- *Markovian* (referred as Markov) action availabilities: we define $C = C^{\mathrm{Markov}} : \mathcal{S}^2 \times \mathcal{A} \times [\![H]\!] \times \mathcal{P}(\mathcal{A}) \to \Delta(\mathcal{P}(\mathcal{A}))$. For every action subsets $\mathcal{B}, \mathcal{B}' \subseteq \mathcal{A}$, states $s, s' \in \mathcal{S}$, action $a \in \mathcal{A}_{k,h}$, and stage $h \in [\![H]\!]$, $C_h^{\mathrm{Markov}}(\mathcal{B}'|s', s, a, \mathcal{B}) = \Pr(\mathcal{A}_{k,h} = \mathcal{B}'|s_{k,h} = s', s_{k,h-1} = s, a_{k,h-1} = a, \mathcal{A}_{k,h-1} = \mathcal{B})$ represents the probability that $\mathcal{B}'$ is the available action set observed in state $s'$ at stage $h$, conditioned to the fact that $\mathcal{B}$ was the available action set, $s$ the state, and $a$ the played action at

stage $h - 1$, respectively.[3]

## 3. Per-episode Disclosure

In this section, we face the simpler case in which the available actions are revealed *at the beginning* of the episode for every state and stage. We start by discussing the notions of policy, value function, optimality, and regret. Then, we present an algorithm matching the regret of standard RL. We highlight that the per-episode disclosure can be seen as a generalization of *action masking* (see Appendix B).

**Policies, Value Functions, and Optimality.** The action availabilities $\mathcal{A}_{k,h}(s)$ are revealed at the beginning of each episode $k \in [\![K]\!]$, for every state $s \in \mathcal{S}$ and stage $h \in [\![H]\!]$. Thus, we restrict the policy space to the policies that play available actions only, i.e., $\Pi_k := \{\pi : \mathcal{S} \times [\![H]\!] \to \Delta(\mathcal{A}) \text{ s.t. } \forall(s,h) \in \mathcal{S} \times [\![H]\!] : \mathrm{supp}(\pi_h(\cdot|s)) \subseteq \mathcal{A}_{k,h}(s)\}$. For episode $k$ and policy $\pi \in \Pi_k$, we denote with $V_h^\pi(s)$ the value function in state $s$ at stage $h$ following policy $\pi$ and with $Q_h^\pi(s,a)$ the state-action value function, *restricted* to the available actions $a \in \mathcal{A}_{k,h}(s)$. For episode $k$, we denote as *optimal policy* any policy fulfilling $\pi_k^* \in \arg\max_{\pi \in \Pi_k} V_1^\pi(\bar{s})$ and the optimal value functions as $V_{k,h}^*(s) = V_h^{\pi_k^*}(s)$ and $Q_{k,h}^*(s,a) = Q_h^{\pi_k^*}(s,a)$. An optimal policy $\pi_k^*$ can be retrieved as the greedy policy w.r.t. $Q_h^*$, restricting to the available actions using a variation of value iteration, namely `Action-Restricted Value Iteration` (AR-VI, Algorithm 3).[4]

**Regret.** We evaluate the performance of an algorithm $\mathfrak{A}$ in terms of cumulative regret over the $K \in \mathbb{N}$ episodes against the optimal policy $\pi_k^*$ *of each episode*.

**Definition 3.1** (Per-episode Disclosure Regret). *Let $\mathfrak{A}$ be an algorithm playing sequence of policies $(\pi_k)_{k=1}^K \in \times_{k=1}^K \Pi_k$, we define the* per-episode (PE) *disclosure regret as:*

$$R_{PE}(\mathfrak{A}, T) := \sum_{k \in [\![K]\!]} \left( V_{k,1}^*(\bar{s}) - V_1^{\pi_k}(\bar{s}) \right). \quad (1)$$

Since we know which actions are available *at every stage and for every state* in advance, we can afford to compete against the best policy $\pi_k^*$ in every episode. Indeed, regardless of the fact that the action availabilities $\mathcal{A}_{k,h}(s)$ are chosen in a *stochastic* or *adversarial* way, we are able to tackle the problem as if we were in a different MDP (defined in terms of the available actions) in each episode $k$.

**Lower Bound.** We now present a minimax lower bound

---

[3]With little abuse, we are using the same symbol even for stage $h = 1$, where $s_{h-1}$, $a_{h-1}$, and $\mathcal{A}_{h-1}$ are not defined. In such a case, we consider $C_1^{\mathsf{Markov}}(\mathcal{B}'|s') = \Pr(\mathcal{A}_{k,1} = \mathcal{B}'|s_{k,1} = s')$ as the *initial-action availability* distribution.

[4]This is totally equivalent to solving an MDP with action sets that depend on the state and stage.

---

**Algorithm 3:** `Action-Restricted Value Iteration` (AR-VI) for episode $k$.

**Input:** Sleeping MDP $\mathcal{M} = (\mathcal{S}, \mathcal{A}, C, P, R, H, \bar{s})$, Available actions $\mathcal{A}_{k,h}(s)$, $\forall h \in [\![H]\!], s \in \mathcal{S}$

1   $V_{k,H+1}^*(s) = 0$
2   **for** $h \in \{H, H-1, \ldots, 1\}$ **do**
3     $Q_{k,h}^*(s,a) = R_h(s,a) + \mathbb{E}_{s' \sim P_h(\cdot|s,a)}\left[V_{k,h+1}^*(s')\right],$
         $\forall a \in \mathcal{A}_{k,h}(s), s \in \mathcal{S}$
4     $V_{k,h}^*(s) = \max_{a \in \mathcal{A}_{k,h}(s)} Q_{k,h}^*(s,a), \ \forall s \in \mathcal{S}$
5   **end**
6   **return**
    $\pi_{k,h}^*(s) \in \arg\max_{a \in \mathcal{A}_{k,h}(s)} Q_{k,h}^*(s,a), \ \forall s \in \mathcal{S}, h \in [\![H]\!]$

---

for SleMDPs with per-episode disclosure.

**Theorem 3.1** (Lower Bound – Per-episode Disclosure). *For any algorithm $\mathfrak{A}$, there exists an instance of Sleeping MDP such that, for $T \geqslant \Omega(H^2 SA)$, the per-episode disclosure regret satisfies:*

$$\mathbb{E}\left[R_{PE}(\mathfrak{A}, T)\right] \geqslant \Omega\left(H\sqrt{SAT}\right).$$

*Proof.* The proof directly follows from that of (Domingues et al., 2021, Theorem 9).[5] We have to lower bound the regret on the worst instance of SleMDPs with per-episode disclosure. Since the SleMDPs per-episode disclosure are a generalization of MDPs (an MDP is a SleMDP where $\mathcal{A}_{k,h}(s) = \mathcal{A}, \forall s \in \mathcal{S}, h \in [\![H]\!], k \in [\![K]\!]$), the regret lower bound for MDPs holds for SleMDPs too. $\square$

**Algorithm.** To learn in the per-episode scenario, we modify `UCBVI` (Azar et al., 2017) to handle the action availabilities. We design `Action-Restricted UCBVI` (AR-UCBVI, Algorithm 4), the optimistic counterpart of the AR-VI (Algorithm 3). From a high-level perspective, besides the fact that the maximization in Bellman's equation is computed over the available actions $\mathcal{A}_{h,k}(s)$ only, AR-UCBVI has to carefully handle the optimism to guarantee a monotonicity property of the sequence of estimated state-action value functions. The algorithm starts by initializing the visitation counters (line 1). Then, for every episode $k \in [\![K]\!]$, the algorithm observes the action availabilities $\mathcal{A}_{k,h}(s)$ (line 3) and estimates the transition model $\widehat{P}_{k,h}(s'|s,a)$ (line 4):

$$\widehat{P}_{k,h}(s'|s,a) = \frac{N_{k,h}(s,a,s')}{N_{k,h}(s,a)}, \quad (2)$$

where $N_{k,h}(s,a)$ is the number of times action $a$ was played in state $s$ in stage $h$, and $N_{k,h}(s,a,s')$ is the number of times the next state was $s'$. Then, AR-UCBVI runs optimistic value iteration to obtain optimistic estimates of the optimal action-value function $Q_{k,h}^*(s,a)$. As mentioned above, this step requires more attention w.r.t. that of (Azar et al.,

---

[5]This result differs from the one of (Domingues et al., 2021) due to the different notation adopted.

**Algorithm 4:** `Action-Restricted UCBVI (AR-UCBVI)`.

1 **Initialize**: $N_{0,h}(s,a,s') = 0$, $N_{0,h}(s,a) = 0$,
   $N_{0,h}(s) = 0$, $\forall (s,a,s',h) \in \mathcal{S} \times \mathcal{A} \times \mathcal{S} \times [\![H]\!]$
2 **for** $k \in [\![K]\!]$ **do**
3    Agent observes $\mathcal{A}_{k,h}(s)$, $\forall h \in [\![H]\!], s \in \mathcal{S}$
4    Estimate $\widehat{P}_{k,h}(s'|s,a)$ as in Eq. (2)
5    //Compute $\widehat{V}_{k,\cdot}(\cdot), \widehat{Q}_{k,\cdot}(\cdot,\cdot)$ for episode $k$
6    $\widehat{Q}_{0,h}^k(s,a) = H - h + 1$, $\forall (s,a,h) \in \mathcal{S} \times \mathcal{A} \times [\![H]\!]$
7    **for** $j \in [\![k]\!]$ **do**
8      Initialize $\widehat{V}_{j,H+1}^k(s) = 0, \forall s \in \mathcal{S}$
9      **for** $h = \{H, H-1, \ldots, 1\}$ **do**
10        **for** $s \in \mathcal{S}$ **do**
11          Compute $\widehat{Q}_{j,h}^k(s,a), \forall a \in \mathcal{A}_{k,h}(s)$
            as in Eq. (3)
12          Compute $\widehat{V}_{j,h}^k(s) = \max\limits_{a \in \mathcal{A}_{k,h}(s)} \widehat{Q}_{j,h}^k(s,a)$
13        **end**
14      **end**
15    **end**
16    //Play optimistically for episode $k$
17    Agent observes state $s_{k,1}$
18    **for** $h \in [\![H]\!]$ **do**
19      Agent plays $a_{k,h} \in \arg\max\limits_{a \in \mathcal{A}_{k,h}(s_{k,h})} \widehat{Q}_{k,h}^k(s_{k,h}, a)$
20      Environment returns $r_{k,h}$ and $s_{k,h+1}$
21      Increment counters
22    **end**
23 **end**

2017). Indeed, the original `UCBVI`, to ensure a monotonically non-increasing sequence of the estimates $\widehat{Q}_{k,h}(s,a)$, limits the current estimate $\widehat{Q}_{k,h}(s,a)$ (computed with all samples up to episode $k-1$) to the previous episode estimate $\widehat{Q}_{k-1,h}(s,a)$ (computed with all samples up to episode $k-2$). This operation makes no sense in a SleMPD since the action availabilities $\mathcal{A}_{k,h}(s)$ and $\mathcal{A}_{k-1,h}(s)$ may change between consecutive episodes, making $\widehat{Q}_{k-1,h}(s,a)$ no longer an optimistic estimate of the true $Q_{k,h}^*(s,a)$. For this reason, we have to compute the sequence of optimistic value functions $\widehat{Q}_{j,h}^k(s,a)$ for the action availabilities $\mathcal{A}_{k,h}(s)$ *of episode $k$* using the estimates $\widehat{P}_{j,h}$ *of all episodes $j$ with $j \in [\![k]\!]$*. This way, we make use of all the samples collected so far (even in episodes $j$ with action availabilities different from the current ones $\mathcal{A}_{k,h}(s)$). Furthermore, this ensures that the sequence of estimates $\widehat{Q}_{j,h}^k(s,a)$ is monotonically non-increasing in $j$. As shown in Algorithm 4 (lines 6-15), `AR-UCBVI` starts from $h = H$ and goes backward computing the optimistic $\widehat{Q}_{j,h}^k(s,a)$ for every $a \in \mathcal{A}_{k,h}(s)$:

$$\widehat{Q}_{j,h}^k(s,a) = \min\Big\{\widehat{Q}_{j-1,h}^k(s,a), \quad (3)$$
$$R_h(s,a) + \sum_{s' \in \mathcal{S}} \widehat{P}_{j,h}(s'|s,a)\widehat{V}_{j,h+1}^k(s') + b_{j,h}^{Q,k}(s,a)\Big\},$$

where $b_{j,h}^{Q,k}(s,a)$ is the exploration bonus obtained from

a refined analysis of `UCBVI` (Drago et al., 2025) and is defined as:

$$b_{j,h}^{Q,k}(s,a) := \sqrt{\frac{4\overline{L}\widehat{\mathbb{V}}_{j,h}^k(s,a)}{N_{j,h}(s,a)}} + \frac{7H\overline{L}}{3N_{j,h}(s,a)}$$
$$+ \sqrt{\frac{8\mathbb{E}_{s' \sim \widehat{P}_{j,h}(\cdot|s,a)}[\overline{b}_{j,h+1}^{Q,k}(s')]}{N_{j,h}(s,a)}},$$

where $\widehat{\mathbb{V}}_{j,h}^k = \mathbb{V}\mathrm{ar}_{s' \sim \widehat{P}_{j,h}(\cdot|s,a)}[\widehat{V}_{j,h+1}^k(s')]$ is the empirical variance of the next-state value estimate, $\overline{b}_{j,h+1}^{Q,k}(s') = \min\{84^2 H^3 S^2 A\overline{L}/N_{j,h+1}(s'), H^2\}$ is the additional bonus, and $\overline{L} = \ln(5HSAT/\delta)$. Then, we compute the value estimate as $\widehat{V}_{j,h}^k(s) = \max_{a \in \mathcal{A}_{k,h}(s)} \widehat{Q}_{j,h}^k(s,a)$. Finally, the algorithm plays an action greedily w.r.t. the optimistic estimate $\widehat{Q}_{k,h}^k(s,a)$ (lines 17-22).

The following result provides the regret of `AR-UCBVI`, showing that learning in a SleMDP with per-episode disclosure does not increase the regret w.r.t. standard RL.

**Theorem 3.2** (Upper Bound – Per-episode Disclosure). *For any $\delta \in (0,1)$, with probability $1 - \delta$, the per-episode disclosure regret of `AR-UCBVI` is bounded by:*

$$R_{PE}(\text{AR-UCBVI}, T) \leqslant 34H\overline{L}\sqrt{SAT} + 2500H^4 S^2 A\overline{L}^2,$$

*where $\overline{L} = \ln(5HSAT/\delta)$. For $T \geqslant \Omega(H^6 S^3 A)$, selecting $\delta = 1/T$, we have:*

$$\mathbb{E}\left[R_{PE}(\text{AR-UCBVI}, T)\right] \leqslant \widetilde{\mathcal{O}}\left(H\sqrt{SAT}\right),$$

*where the expectation is taken w.r.t. the stochasticity of the environment.*

*Proof.* The proof of this theorem follows the one of (Azar et al., 2017, Theorem 2). The key challenge is the computation of the bonus $b_{j,h}^{Q,k}$. Indeed, Lemma 17 of (Azar et al., 2017) heavily relies on the fact that the sequence $\widehat{V}_{k,h}(s) - V_h^*(s)$ is monotonically non-increasing in $k$. This is not the case for our sequence $\widehat{V}_{k,h}^k(s) - V_{k,h}^*(s)$ since, as already explained, the action availabilities $\mathcal{A}_{k,h}(s)$ change across episodes. For this reason, we resort to the sequence $\widehat{V}_{j,h}^k(s) - V_{k,h}^*(s)$ for *fixed $k$* which is monotonically non-increasing in $j$. This allows us to apply Lemma 16 of (Azar et al., 2017) *pretending* to have played in the MDP with action availabilities of episode $k$, i.e., $\mathcal{A}_{k,h}(s)$, for all episodes $j \in [\![k]\!]$ and, ultimately, getting the bonus as in Lemma 17. Notice that we apply the *pigeonhole principle* considering $\mathcal{A}_{k,h}(s) = \mathcal{A}$ which represents the worst case. $\square$

## 4. Per-stage Disclosure: Markovian Case

We now analyze the realistic scenario where the set of available actions is revealed for the current stage only with no knowledge of future availabilities. In this sec-

tion, we focus on Markovian availabilities i.e., $\mathcal{A}_{k,h} \sim C_h^{\mathsf{Markov}}(\cdot|s_{k,h}, s_{k,h-1}, a_{k,h-1}, \mathcal{A}_{k,h-1})$.

**Augmented MDP, Policies, Value Functions, and Optimality.** To address this scenario, we can map the SleMDP with an *augmented MDP* in which we encode action availability sets in the state representation.

**Definition 4.1** (Augmented MDP). *Let* $\mathcal{M} := (\mathcal{S}, \mathcal{A}, C, P, R, H, \overline{s})$ *be a SleMDP, we define the* augmented MDP $\widetilde{\mathcal{M}} := (\widetilde{\mathcal{S}}, \mathcal{A}, \widetilde{P}, \widetilde{R}, H, \widetilde{P}_0)$, *with:*

- *augmented state space* $\widetilde{\mathcal{S}} := \mathcal{S} \times \mathcal{P}(\mathcal{A})$;
- *augmented transition probability, defined for every* $\widetilde{s} = (s, \mathcal{B})$, $\widetilde{s}' = (s', \mathcal{B}') \in \widetilde{\mathcal{S}}$, $a \in \mathcal{A}$, *and* $h \in [\![H]\!]$ *as:* $\widetilde{P}_h(\widetilde{s}'|\widetilde{s}, a) = P_h(s'|s, a)C_h^{\mathsf{Markov}}(\mathcal{B}'|s', s, a, \mathcal{B})$;
- *reward function, defined for every* $\widetilde{s} = (s, \mathcal{B}) \in \widetilde{\mathcal{S}}$ *and* $a \in \mathcal{A}$ *as:* $\widetilde{R}(\widetilde{s}, a) = R(s, a)$;
- *initial augmented-state distribution, defined for every* $\widetilde{s} = (s, \mathcal{B}) \in \widetilde{\mathcal{S}}$ *as* $\widetilde{P}_0(\widetilde{s}) = C_1^{\mathsf{Markov}}(\mathcal{B}|\overline{s})\mathbb{1}\{s = \overline{s}\}$.

Note that the augmented MDP has a state space with cardinality $|\widetilde{\mathcal{S}}| = S2^A$, exponential in $A$. The policy space in the augmented MDP is defined as $\widetilde{\Pi} := \{\widetilde{\pi} : \widetilde{\mathcal{S}} \times [\![H]\!] \to \Delta(\mathcal{A})$ s.t. $\forall \widetilde{s} = (s, \mathcal{B}) \in \widetilde{\mathcal{S}} :$ $\mathrm{supp}(\widetilde{\pi}(\cdot|\widetilde{s})) \subseteq \mathcal{B}\}$, to ensure that only available actions are played.[6] For a policy $\widetilde{\pi}$, augmented state $\widetilde{s} = (s, \mathcal{B})$, available action $a \in \mathcal{B}$, and stage $h \in [\![H]\!]$, we denote with $\widetilde{V}_h^{\widetilde{\pi}}(\widetilde{s}) = \widetilde{V}_h^{\widetilde{\pi}}((s, \mathcal{B}))$ and $\widetilde{Q}_h^{\widetilde{\pi}}(\widetilde{s}, a) = \widetilde{Q}_h^{\widetilde{\pi}}((s, \mathcal{B}), a)$ the value and state-action value functions, respectively, in the augmented MDP, the latter *restricted* to the available actions $a \in \mathcal{B}$. An optimal policy for the augmented MDP is any policy such that $\widetilde{\pi}^* \in \arg\max_{\widetilde{\pi} \in \widetilde{\Pi}} \widetilde{V}^{\widetilde{\pi}}$ and the optimal value functions as $\widetilde{V}_h^*(\widetilde{s}) = \widetilde{V}_h^{\widetilde{\pi}^*}(\widetilde{s})$ and $\widetilde{Q}_h^*(\widetilde{s}, a) = \widetilde{Q}_h^{\widetilde{\pi}^*}(\widetilde{s}, a)$. An optimal policy $\widetilde{\pi}^*$ can be obtained as greedy w.r.t. $\widetilde{Q}^*$.[7] The optimal value functions can be computed using value iteration on the augmented MDP (see Algorithm 5).

Considering the protocol of Algorithm 2, we define the value function for the original SleMDP $V_1^{\widetilde{\pi}}(\overline{s})$ in the initial state $\overline{s} \in \mathcal{S}$ as the expectation of that of the augmented MDP $\widetilde{V}_1^{\widetilde{\pi}}((\overline{s}, \mathcal{B}))$ over the available action sets $\mathcal{B} \in \mathcal{P}(\mathcal{A})$:

$$V_1^{\widetilde{\pi}}(\overline{s}) := \mathbb{E}_{\mathcal{B} \sim C^{\mathsf{Markov}}(\cdot|\overline{s})}[\widetilde{V}_1^{\widetilde{\pi}}((\overline{s}, \mathcal{B}))]. \tag{4}$$

Similarly, for the optimal value function, we have: $V_1^*(\overline{s}) := \mathbb{E}_{\mathcal{B} \sim C^{\mathsf{Markov}}(\cdot|\overline{s})}[\widetilde{V}_1^*((\overline{s}, \mathcal{B}))]$.

**Regret.** We evaluate the performance of an algorithm in terms of cumulative regret over the $K \in \mathbb{N}$ episodes against

the optimal policy $\widetilde{\pi}^*$ *constant* throughout the episodes.

**Definition 4.2** (Per-stage Disclosure Regret). *Let* $\mathfrak{A}$ *be an algorithm playing a sequence of policies* $(\widetilde{\pi}_k)_{k=1}^K \in \widetilde{\Pi}^K$, *we define the* per-stage disclosure (PS) *policy regret as:*

$$R_{PS}(\mathfrak{A}, T) = KV_1^*(\overline{s}) - \sum_{k \in [\![K]\!]} V_1^{\widetilde{\pi}_k}(\overline{s}).$$

Differently from the per-episode case (Definition 3.1) where comparator $\pi_k^*$ changes over episodes, in the per-stage case (Definition 4.2), we consider a constant comparator $\widetilde{\pi}^*$.

**Algorithm.** Given the mapping to the augmented MDP, we can resort to the original UCBVI (Azar et al., 2017) to learn in the SleMDP with the following regret guarantees.

**Theorem 4.1** (Upper Bound – Per-stage Disclosure: Markovian). *For any* $\delta \in (0, 1)$, *with probability* $1 - \delta$, *the per-stage disclosure regret of* UCBVI *with Bernstein-Freedman bonuses (Azar et al., 2017,* bonus_2*) on the augmented MDP is bounded by:*

$$R_{PS}(\text{UCBVI}, T) \leqslant 34H\widetilde{L}\sqrt{SAT2^A} + 2500H^4S^2A2^{2A}\widetilde{L}^2,$$

*where* $\widetilde{L} = \log(5H^2S2^AAT/\delta)$. *In particular, for* $T \geqslant \Omega(H^6S^3A^32^{3A})$ *and selecting* $\delta = 2^A/T$, *we have:*

$$\mathbb{E}[R_{PS}(\text{UCBVI}, T)] \leqslant \widetilde{\mathcal{O}}\left(H\sqrt{SAT2^A}\right).$$

*Proof.* The proof is an application of (Azar et al., 2017, Theorem 2), where we consider the state space cardinality $S2^A$ of the augmented MDP and the stage-dependent transition probabilities (equivalent to increasing the state space cardinality by a multiplicative factor of $H$). $\square$

The regret guarantees of Theorem 4.1 are clearly unsatisfactory due to the presence of the exponential dependence on the cardinality of the action space $\mathcal{A}$. In the following, we show that such an exponential dependence is unavoidable when the action availability follows a Markovian process.

**Lower Bound.** The following theorem presents the lower bound on the Markovian per-stage disclosure regret.

**Theorem 4.2** (Lower Bound – Per-stage Disclosure: Markovian). *For any algorithm* $\mathfrak{A}$, *there exists an instance of Sleeping MDP with per-stage disclosure and Markovian action availability such that, for* $T \geqslant \Omega(HSA2^A)$ *and* $H \geqslant \Omega(A)$, *the per-stage disclosure policy regret satisfies:*

$$\mathbb{E}[R_{PS}(\mathfrak{A}, T)] \geqslant \Omega\left(H\sqrt{SAT2^{A/2}}\right).$$

*Proof Sketch.* The instances are made of two parts (see Figure 3). First, every instance has an $A$-ary tree MDP as in (Domingues et al., 2021), where all actions are available. Then, we attach a partial *sleeping lattice* to every leaf, in which the state $s_i$ does not change, and whenever an action

---

[6] Note the fundamental difference w.r.t. the per-episode case, where the policy space was different for every episode $k$, while here in the per-stage case, the policy space is the same for all episodes, but the policy is conditioned to the actual available actions set $\mathcal{B}$.

[7] As usual, we are in an MDP and, therefore, there exists a policy $\widetilde{\pi}^*$ optimal from every state $\widetilde{s}$.

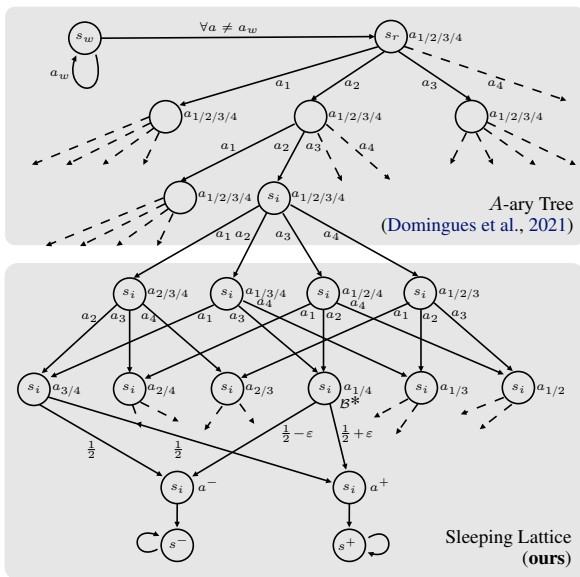

*Figure 3.* Instances used in the proof of Theorem 4.2. Available actions are reported next to the nodes.

---

**Algorithm 5:** `Sleeping VI` for $C = C^{\text{Markov}}$.

**Input:** Sleeping MDP $\mathcal{M} = \left(\mathcal{S}, \mathcal{A}, C^{\text{Markov}}, P, R, H, \overline{s}\right)$

1   $\widetilde{V}_{H+1}^*((s, \mathcal{B})) = 0, \ \forall s \in \mathcal{S}, \mathcal{B} \in \mathcal{P}(\mathcal{A})$
2   **for** $h \in \{H, H-1, \dots, 1\}$ **do**
3     $\widetilde{Q}_h^*((s, \mathcal{B}), a) = R_h(s, a) +$
     $\mathbb{E}_{s' \sim P_h(\cdot|s,a)} \left[ \mathbb{E}_{\mathcal{B}' \sim C_h^{\text{Markov}}(\cdot|s's,a,\mathcal{B})} \left[ \widetilde{V}_{h+1}^*((s', \mathcal{B}')) \right] \right],$
       $\forall s \in \mathcal{S}, \mathcal{B} \in \mathcal{P}(\mathcal{A}), a \in \mathcal{B}$
4     $\widetilde{V}_h^*((s, \mathcal{B})) = \max_{a \in \mathcal{B}} \widetilde{Q}_h^*((s, \mathcal{B}), a),$
       $\forall s \in \mathcal{S}, \mathcal{B} \in \mathcal{P}(\mathcal{A})$
5   **end**
6   **return** $\widetilde{\pi}_h^*((s, \mathcal{B})) \in \arg \max_{a \in \mathcal{B}} \widetilde{Q}_{k,h}^*((s, \mathcal{B}), a),$
      $\forall s \in \mathcal{S}, \ \mathcal{B} \in \mathcal{P}(\mathcal{A}), \ h \in [\![H]\!]$

---

**Algorithm 6:** `Sleeping VI` for $C = C^{\text{ind}}$.

**Input:** Sleeping MDP $\mathcal{M} = \left(\mathcal{S}, \mathcal{A}, C^{\text{ind}}, P, R, H, \overline{s}\right)$

1   $V_{H+1}^*(s) = 0, \ \forall s \in \mathcal{S}$
2   **for** $h \in \{H, H-1, \dots, 1\}$ **do**
3     $Q_h^*(s, a) = R_h(s, a) + \mathbb{E}_{s' \sim P_h(\cdot|s,a)} \left[ V_{h+1}^*(s') \right],$
      $\forall a \in \mathcal{A}, s \in \mathcal{S}$
4     $V_h^*(s) = \mathbb{E}_{\mathcal{B} \sim C_h^{\text{ind}}(\cdot|s)} \left[ \max_{a \in \mathcal{B}} Q_h^*(s, a) \right], \ \forall s \in \mathcal{S}$
5   **end**
6   **return** $\widetilde{\pi}_h^*(s, \mathcal{B}) \in \arg \max_{a \in \mathcal{B}} Q_{k,h}^*(s, a),$
      $\forall s \in \mathcal{S}, \ \mathcal{B} \in \mathcal{P}(\mathcal{A}), \ h \in [\![H]\!]$

---

$a_h$ is played, it gets removed from the next set of available actions, i.e., $\mathcal{A}_{h+1}(s_i) = \mathcal{A}_h(s_i) \backslash \{a_h\}$. This happens for $A/2$ times, until we reach the largest layer of the lattice, i.e., the one with $\binom{A}{A/2} \approx 2^{A/2}$ nodes. At this point, whatever action is played, the next action availability set will be a singleton either containing $a^+$ or $a^-$ with equal probability. $a^+$ (resp. $a^-$) leads to an absorbing state with reward $+1$ (resp. $-1$). Hard instances are constructed by electing a leaf state $s^*$, an action $a^*$, a stage $h^*$, and an availability set $\mathcal{B}^*$ (with $|\mathcal{B}^*| = A/2$) that increase by $\varepsilon > 0$ the probability of getting singleton $\{a^+\}$ in the next stage. $\qquad\square$

This lower bound, using a novel construction (i.e., the *sleeping lattice*, Figure 3), shows that an exponential dependence on $A$ is unavoidable when there is temporal correlation among the sets of available actions. Given this statistical barrier, we next discuss a tractable scenario in which the action availability is sampled independently at each stage.

## 5. Per-Stage Disclosure: Independent Case

We now discuss the scenario in which, at every stage, the available action set $\mathcal{A}_{k,h}$ is sampled independently from the past, i.e., $\mathcal{A}_{k,h} \sim C_h^{\text{ind}}(\cdot|s_{k,h})$. The lack of a temporal structure allows removing the dependence of the value functions on the available action sets, differently from the augmented MDP of Section 4. Indeed, looking at the Bellman's equation in the augmented MDP (line 3 of Algorithm 5), we realize that $C_h^{\text{ind}}(\cdot|s)$ does not depend on $\mathcal{B}$, removing such a dependence from the state-action value function $\widetilde{Q}_h^*((s, \mathcal{B}), a)$, that, from now on, we will denote as $Q_h^*(s, a)$. Moreover, similarly to what has been done in

Equation (4), it is convenient to introduce the value function $V_h^*(s) := \mathbb{E}_{\mathcal{B} \sim C_h^{\text{ind}}(\cdot|s)}[\widetilde{V}_h^*(s, \mathcal{B})]$. Given these quantities, to solve a SleMDP with independent availabilities, we resort to a simpler value iteration approach (Algorithm 6).

**Lower Bound.** The following theorem provides a regret lower bound, by reducing the considered scenario to an MDP in which all actions are always available.

**Theorem 5.1** (Lower Bound – Per-stage Disclosure: Independent). *For any algorithm $\mathfrak{A}$, there exists an instance of SleMDPs with independent action availability with per-stage disclosure such that, for $T \geqslant \Omega(H^2 SA)$, the per-stage disclosure regret satisfies:*

$$\mathbb{E}\left[R_{PS}(\mathfrak{A}, T)\right] \geqslant \Omega\left(H\sqrt{SAT}\right).$$

*Proof Sketch.* The formal proof is provided in Appendix C. Similarly to Theorem 3.1, the proof directly follows from that of (Domingues et al., 2021, Theorem 9). We have to lower bound the regret on the worst instance of SleMDPs with independent per-stage disclosure action availability. Since the SleMDPs with independent per-stage action availability are a generalization of MDPs (an MDP is a SleMDP where $C_h^{\text{ind}}(\mathcal{A}|s) = 1, \ \forall s \in \mathcal{S}, h \in [\![H]\!]$), the regret lower bound for MDPs holds for SleMDPs too. $\qquad\square$

**Algorithm.** To efficiently learn in this setting, we propose an algorithm that extends UCBVI (Azar et al., 2017) with the

---

**Algorithm 7:** `Sleeping UCBVI (S-UCBVI).`

1 **Initialize**: $N_{0,h}(s,a,s') = 0$, $N_{0,h}(s,a) = 0$,
2          $N_{0,h}(s) = 0$, $N_{0,h}(s,\mathcal{B}) = 0$,
3          $\forall (s,a,s',h,\mathcal{B}) \in \mathcal{S} \times \mathcal{A} \times \mathcal{S} \times \llbracket H \rrbracket \times \mathcal{P}(\mathcal{A})$
4 $\widehat{Q}_{0,h}(s,a) = H - h + 1$, $\forall (s,a,h) \in \mathcal{S} \times \mathcal{A} \times \llbracket H \rrbracket$
5 **for** $k \in \llbracket K \rrbracket$ **do**
6      //Compute $\widehat{V}_{k,\cdot}(\cdot), \widehat{Q}_{k,\cdot}(\cdot,\cdot)$ for episode $k$
7      Estimate $\widehat{P}_{k,h}(s'|s,a)$ as in Eq. (2)
8      Estimate $\widehat{C}^{\text{ind}}_{k,h}(\mathcal{B}|s)$ as in Eq. (5)
9      Initialize $\widehat{V}_{k,H+1}(s) = 0, \forall s \in \mathcal{S}$
10      **for** $h = \{H, H-1, \ldots, 1\}$ **do**
11         **for** $s \in \mathcal{S}$ **do**
12            Compute $\widehat{Q}_{k,h}(s,a), \forall a \in \mathcal{A}$, as in Eq. (6)
13            Compute $\widehat{V}_{k,h}(s)$ as in Eq. (7)
14         **end**
15      **end**
16      //Play optimistic for episode $k$
17      Agent observes state $s_{k,1}$
18      **for** $h \in \llbracket H \rrbracket$ **do**
19         Agents observes action set $\mathcal{A}_{k,h}(s_{k,h})$
20         Agent plays $a_{k,h} \in \underset{a \in \mathcal{A}_{k,h}(s_{k,h})}{\arg\max} \widehat{Q}_{k,h}(s_{k,h},a)$
21         Environment returns $r_{k,h}$ and $s_{k,h+1}$
22         Increment counters
23      **end**
24 **end**

---

estimation of the action availabilities $C^{\text{ind}}_h(\cdot|s)$. `Sleeping UCBVI` (`S-UCBVI`, Algorithm 7) is an optimistic algorithm where the key innovation is using *two bonuses*, one for the state-action value functions $\widehat{Q}_{k,h}(s,a)$ (as for `UCBVI`, to handle uncertainty on $\widehat{P}$) and one for the state value functions $\widehat{V}_{k,h}(s)$ (to handle uncertainty on $\widehat{C}$). We estimate the transition model as in Equation (2) (line 7) and we keep an estimate of the action availability as follows (line 8):

$$\widehat{C}^{\text{ind}}_{k,h}(\mathcal{B}|s) = \frac{N_{k,h}(s,\mathcal{B})}{N_{k,h}(s)}, \quad (5)$$

where $N_{k,h}(s)$ is the number of times state $s$ is visited at stage $h$ and $N_{k,h}(s,\mathcal{B})$ is the number of times we observe action availability $\mathcal{B}$. Then, we perform an optimistic value iteration (lines 9-15) to obtain the optimistic estimate $\widehat{V}_{k,h}(s)$ and $\widehat{Q}_{k,h}(s,a)$ by means of *two additive optimistic bonuses*. Going backward from $h = H$, we compute $\widehat{Q}_{k,h}(s,a)$ as:

$$\widehat{Q}_{k,h}(s,a) = \min\Big\{\widehat{Q}_{k-1,h}(s,a), \quad (6)$$
$$R_h(s,a) + b^Q_{k,h}(s,a) + \sum_{s' \in \mathcal{S}} \widehat{P}_{k,h}(s'|s,a)\widehat{V}_{k,h+1}(s')\Big\},$$

where $b^Q_{k,h}(s,a)$ is the exploration bonus to account for the uncertainty on the transition model estimate $\widehat{P}$:

$$b^Q_{k,h}(s,a) := \sqrt{\frac{4L\widehat{\mathbb{V}}_{k,h}(s,a)}{N_{k,h}(s,a)}} + \frac{7HL}{3(N_{k,h}(s,a)-1)}$$

$$+ \sqrt{\frac{4\mathbb{E}_{s' \sim \widehat{P}_{k,h}(\cdot|s,a)}[\bar{b}^Q_{k,h+1}(s')]}{N_{k,h}(s,a)}},$$

where $\widehat{\mathbb{V}}_{k,h} = \mathbb{V}\text{ar}_{s' \sim \widehat{P}_{k,h}(\cdot|s,a)}[\widehat{V}_{k,h+1}(s')]$ is the empirical variance of the next-state estimated value function, $\bar{b}^Q_{k,h+1}(s') = \min\{2900^2 H^3 S^3 A 2^A L^3/N_{k,h+1}(s'), H^2\}$ is the additional bonus term, and $L = \log(80HS^2A2^AT/\delta)$. Then, we compute the optimistic value function $\widehat{V}_{k,h}(s)$ as:

$$\widehat{V}_{k,h}(s) = \sum_{\mathcal{B} \in \mathcal{P}(\mathcal{A})} \widehat{C}^{\text{ind}}_{k,h}(\mathcal{B}|s)\max_{a \in \mathcal{B}} \widehat{Q}_{k,h}(s,a) + b^V_{k,h}(s), \quad (7)$$

where $b^V_{k,h}$ is a bonus accounting for the uncertainty on the action set availability defined as:

$$b^V_{k,h}(s) := \sqrt{\frac{4L\widehat{\mathbb{Q}}_{k,h}(s)}{N_{k,h}(s)}} + \frac{7HL}{3(N_{k,h}(s)-1)}$$

$$+ \sqrt{\frac{4\mathbb{E}_{\mathcal{B} \sim \widehat{C}^{\text{ind}}_{k,h}(\cdot|s)}[\bar{b}^V_{k,h}(s, \pi_{k,h}(s,\mathcal{B}_{k,h}))]}{N_{k,h}(s)}},$$

where $\widehat{\mathbb{Q}}_{k,h}(s) = \mathbb{V}\text{ar}_{\mathcal{B} \sim \widehat{C}^{\text{ind}}_{k,h}(\cdot|s)}[\widehat{Q}_{k,h}(s, \pi_{k,h}(s,\mathcal{B}))]$, and $\bar{b}^V_{k,h}(s,a) = \min\{1350^2 H^3 S^3 A 2^A L^3/N_{k,h}(s,a), H^2\}$ is an additional bonus. Finally, the algorithm plays an action greedily w.r.t. $\widehat{Q}_{k,h}(s,a)$ (lines 17-23).

We provide the regret upper bound of `S-UCBVI` for per-stage disclosure and independent availabilities.

**Theorem 5.2** (Regret Upper Bound `S-UCBVI` with independent availability and per-stage disclosure)**.** *For any $\delta \in (0,1)$, with probability $1-\delta$, the per-stage disclosure regret of* `S-UCBVI` *on any SleMDP with per-stage disclosure independent action availabilities is bounded by:*

$$R_{PS}(\text{S-UCBVI}, T) \leq 512H\sqrt{SATLG}$$
$$+ 498^2 H^6 S^3 A 2^A L^2 G,$$

*where $L = \log(80HS^2A2^AT/\delta)$ and $G = \log(HSAT)$. In particular, for $T \geq \Omega(H^{10}S^5A^42^{2A})$ and selecting $\delta = 2^A/T$, we have:*

$$\mathbb{E}\left[R_{PS}(\text{S-UCBVI}, T)\right] \leq \widetilde{\mathcal{O}}\left(H\sqrt{SAT}\right).$$

*Proof Sketch.* The proof is provided in Appendix D and extends (Azar et al., 2017, Theorem 2). The key difference is the use of two bonus terms to ensure the optimism of both $\widehat{V}_{k,h}$ and $\widehat{Q}_{k,h}$ due to the estimation of the action availabilities $\widehat{C}^{\text{ind}}_{k,h}$ and of the transition model $\widehat{P}_{k,h}$. Given $\widetilde{\Delta}^V_{k,h} = \widehat{V}_{k,h} - V^{\pi_k}_h$ and $\widetilde{\Delta}^Q_{k,h} = \widehat{Q}_{k,h} - Q^{\pi_k}_h$, we notice a recursive dependence, that we unfold since $\widetilde{\Delta}^V$ can be derived from $\widetilde{\Delta}^Q$ and $\widetilde{\Delta}^Q$ is obtained by upper-bounding $\widetilde{\Delta}^V$ as:

$$\widetilde{\Delta}^V_{k,h} \leq b^V_{k,h} + \varepsilon^Q_{k,h} + 4H^2 2^A L/N_{k,h} + \widetilde{\Delta}^Q_{k,h},$$

| | Per-stage Disclosure | | Per-episode Disclosure | |
| --- | --- | --- | --- | --- |
| | Lower Bound | Upper Bound | Lower Bound | Upper Bound |
| Independent | $\Omega\big(H\sqrt{SAT}\big)$ $T \geqslant \Omega(H^2SA)$ Theorem 5.1 | $\tilde{\mathcal{O}}\big(H\sqrt{SAT}\big)$ $T \geqslant \Omega(H^{10}S^5A^42^{2A})$ Theorem 5.2 | $\Omega\big(H\sqrt{SAT}\big)$ $T \geqslant \Omega(H^2SA)$ Theorem 3.1 | $\tilde{\mathcal{O}}\big(H\sqrt{SAT}\big)$ $T \geqslant \Omega(H^6S^3A)$ Theorem 3.2 |
| Markovian | $\Omega\big(H\sqrt{SAT2^{A/2}}\big)$ $T \geqslant \Omega(HSA2^A)$ Theorem 4.2 | $\tilde{\mathcal{O}}\big(H\sqrt{SAT2^A}\big)$ $T \geqslant \Omega(H^6S^3A^32^{3A})$ Theorem 4.1 | | |

*Table 1.* Summary of the results.

where $\varepsilon_{k,h}^Q$ is a martingale difference sequence that leads to a lower-order term. As such, we upper bound the two quantities as done in Lemma 3 of (Azar et al., 2017) at the cost of a multiplicative $e$ constant, avoiding any exponential dependency on $A$ in the higher-order terms. Then, extending the rationale of (Azar et al., 2017), via backwards induction, we show that the empirical variance is small enough that the additional bonuses $\bar{b}_{k,h}^V$ and $\bar{b}_{k,h}^Q$ guarantee that $\hat{V}_{k,h}$ and $\hat{Q}_{k,h}$ are indeed optimistic. This is done by deriving an upper bound on $\hat{Q}_{k,h}(s,a) - Q_h^*(s,a)$ in the order of:

$$\hat{Q}_{k,h}(s,a) - Q_h^*(s,a) \leqslant \min\{\mathcal{O}(H^3S^3A2^AL^3/N_{k,h}(s,a)), H\}.$$

Then, we use the latter result to derive an upper bound to $\hat{V}_{k,h}(s) - V_h^*(s)$ in the order of:

$$\hat{V}_{k,h}(s) - V_h^*(s) \leqslant \min\{\mathcal{O}(H^3S^3A2^AL^3/N_{k,h}(s)), H\}.$$

Subsequently, we use these inequalities to show that $\hat{Q}_{k,h} \geqslant Q_h^*$ and $\hat{V}_{k,h} \geqslant V_h^*$ hold, thus, demonstrating the optimism. Finally, we derive the regret bound by combining the terms in the upper bound of $\tilde{\Delta}_{k,h}^V$, observing that, when applying the pigeonhole principle, we consider all the actions in $\mathcal{A}$ as available, as this provides the worst-case allocation. □

This result shows that for a large enough $T$, the regret suffered by S-UCBVI is of the same order as that of UCBVI-BF (Azar et al., 2017, Theorem 2), matching the lower bound for standard RL, up to logarithmic terms. Thus, the need for estimating the action availability $C^{\text{ind}}$, that is a distribution over $\mathcal{P}(\mathcal{A})$, results in a minimum value of $T$ that scales exponentially with $A$. Nevertheless, no exponential dependence on $A$ is present in the leading term.

## 6. Discussion and Conclusions

We summarize the results presented in this paper in Table 1. We motivated the introduction of approaches aware of the action availability in opposition to an LLC since they allow learning better-performing behaviors. We illustrated this in Example 1. Now, we formally prove that this is the case.

**Per-episode $\geqslant$ Per-stage.** We restrict to stochastic availabilities sampled independently at every stage, i.e., via $C^{\text{ind}}$. Indeed, in this case, we can imagine availabilities $\mathcal{A}_{k,h}(s)$ to be pre-sampled for every $k \in [\![K]\!]$ and $(s,h) \in \mathcal{S} \times [\![H]\!]$. In the per-episode disclosure, $\mathcal{A}_{k,h}(s)$ is revealed at the beginning of episode $k$, while in the per-stage disclosure $\mathcal{A}_{k,h}(s)$ is revealed only when, at episode $k$, we reach state $s$ at stage $h$. This observation allows concluding that the expected optimal performance in the per-episode disclosure is superior to that of the per-stage disclosure:

$$\underbrace{\mathbb{E}[V_k^*] = \mathbb{E}\left[\max_{\pi_k \in \Pi_k} V_1^{\pi_k}(\bar{s})\right]}_{\text{Per-episode}} \geqslant \underbrace{\max_{\pi \in \Pi} V_1^{\pi}(\bar{s}) = V_1^*(\bar{s})}_{\text{Per-stage}},$$

where the expectation is taken w.r.t. the randomness of $\mathcal{A}_{k,h}(s)$, and where $\Pi_k$ and $\Pi$ are defined in Sections 3 and 4, respectively, and the inequality follows from Jensen's.

**Per-stage $\geqslant$ LLC.** The LLC can be regarded as a (possibility stochastic) function that, given an unavailable action $a_{k,h} \notin \mathcal{A}_{k,h}$ sampled from policy $\pi_h(\cdot|s_{k,h})$, overrides it with an available action $a'_{k,h} \in \mathcal{A}_{k,h}$ sampled according to some strategy $\rho_h^{\text{LLC}}(\cdot|s_{k,h}, a_{k,h}, \mathcal{A}_{k,h})$. Thus, the overall effect of the LLC is equivalent to playing a policy $\tilde{\pi}_h^{\text{LLC}}(\cdot|s_{k,h}, \mathcal{A}_{k,h}) = \sum_{a \in \mathcal{A}} \rho_h^{\text{LLC}}(\cdot|s_{k,h}, a, \mathcal{A}_{k,h})\pi_h(a|s_{k,h})$ that belongs to the policy space $\Pi$ on which we optimize in the per-stage disclosure case. Thus, the performance of the LLC cannot be larger than that of the optimal policy of the *per-stage* case (i.e., $V_1^{\tilde{\pi}^{\text{LLC}}}(\bar{s}) \leqslant \max_{\pi \in \Pi} V_1^{\pi}(\bar{s}) = V_1^*(\bar{s})$).

**Future Works.** Interesting future research directions include investigating action availability structures that are more general than the independent case while preserving statistical tractability. Moreover, it is of interest to devise *instance-dependent* features to characterize the complexity of an instance in the regret bounds based on the characteristics of action availability.

## Acknowledgments

Funded by the European Union – Next Generation EU within the project NRPP M4C2, Investment 1.3 DD. 341 – 15 March 2022 – FAIR – Future Artificial Intelligence Research – Spoke 4 – PE00000013 – D53C22002380006.

## Impact Statement

This paper presents work whose goal is to advance the field of Machine Learning. There are many potential societal consequences of our work, none which we feel must be specifically highlighted here.

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

## A. Example

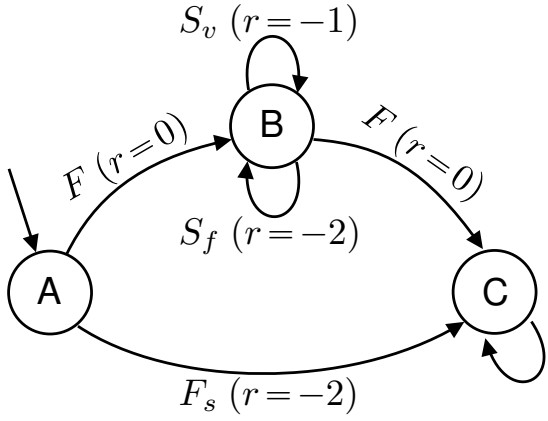

*Figure 4.* Illustrative Sleeping MDP.

In this appendix, we discuss an example of Sleeping MDP, analyze the two disclosure scenarios described in the paper (i.e., *per-episode* and *per-stage disclosure*, indicated with PE and PS, respectively), and the case in which we do not have information about the available actions, making use of a *low-level controller* (LLC) to correct taken actions that are not available, generating an equivalent MDP as depicted in Figure 1a.

Consider the Sleeping MDP depicted in Figure 4, where the goal is to go from initial state A to the absorbing state C. To reach this goal, we have two paths, the first (the one below) is safe but costly, the second one is unsafe and may be not always available, but has no cost. More formally, we consider an undiscounted finite-horizon Sleeping MDP with initial state A, and absorbing state is C (we assume infinite horizon here since there is probability 1 of reaching state C). To reach the absorbing state, we have two possible paths, which can be chosen using deterministic actions. The first one is the one following action $F_s$ (i.e., forward safe), which deterministically leads to the final state with total reward $-2$. The second path, instead, leads to the final state through state B. We assume we can always take action $F$ (i.e., forward) in state A and deterministically reach B without costs ($r = 0$). Then, when we are in B, the next forward action may be available or not, and we assume it is available with probability $p$ (formally $\Pr(F \in \mathcal{A}_h(B)) = p$, for every stage $h$). If $F$ is not available, we can voluntarily stay in B and wait (i.e., play action $S_v$), obtaining a reward of $-1$, or try to go anyway. In state B, there is also another action $S_f$ (i.e., stay forced), that makes the agent remain in state B and receive a reward of $-2$. This latter action will be employed by the LLC to override forbidden actions (i.e., the attempt to play $F$ in state B when it is not available).

We now compute the value functions of this Sleeping MDP for the three scenarios under analysis.

**Sleeping MDP with per-episode disclosure.** We first analyze the optimal value function $V^*_{\mathsf{PE}}(A)$ in state A for the per-episode disclosure scenario. We recall that in this case, we knew the action was available at the beginning of the episode. It is easy to observe that, considering $h = 1$ as the moment in which the first decision is made, the path through B will be convenient if at $h = 2$ or $h = 3$ the action $F$ will be available in state B (if action $F$ will be available for the first time in $h = 4$ or later, we can choose the safe path).[8] Knowing that such action will be available with probability $p$, the probability that it will be available at time $h = 2$ is indeed $p$, while the probability that will be not available in $h = 2$ but will be available in $h = 3$ is $(1 - p)p$. Given that, the value function for state A is:

$$V^*_{\mathsf{PE},1}(A) = 0 \cdot \underbrace{p}_{\substack{f \text{ available} \\ \text{in } h=2}} - 1 \cdot \underbrace{(p(1-p))}_{\substack{f \text{ available in } h=3 \\ \text{and not in } h=2}} - 2 \cdot \underbrace{(1 - (p + p(1-p)))}_{\text{Otherwise}}.$$

**Sleeping MDP with per-stage disclosure.** We now analyze the optimal value function $V^*_{\mathsf{PS},1}(A)$ in state A for the per-stage disclosure scenario, where we do not know the actual action availability in advance. Given the simplicity of the problem, we

---

[8]For the sake of precision, it is not relevant if action $F$ is available in state B when $h = 1$, because we are not in such a state and we cannot play it.

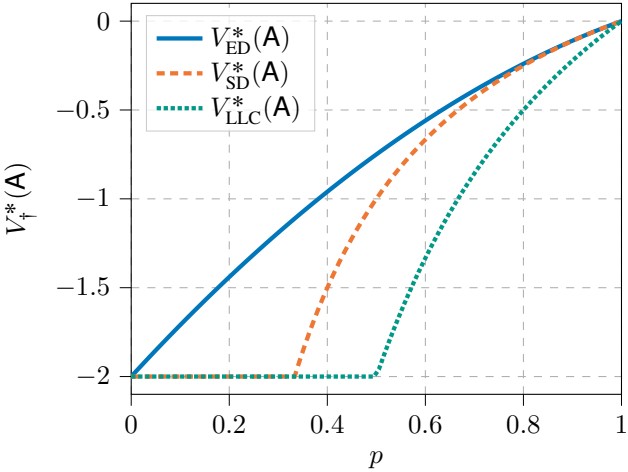

*Figure 5.* Optimal value functions $V_\dagger^*(\mathsf{A})$ for $\dagger \in \{\mathsf{PE}, \mathsf{PS}, \mathsf{LLC}\}$.

analyze value functions from the final state backward. We first observe how, for absorbing state $\mathsf{C}$, we have $V_{\mathsf{PS},h}^*(\mathsf{C}) = 0$ for every $h \in \mathbb{N}$. Then, we move to state $\mathsf{B}$, and we trivially observe that the optimal policy is to go to state $\mathsf{C}$ if possible (i.e., if action $F$ is available), and use action $S_v$ otherwise. The value function for state $\mathsf{B}$ is:

$$V_{\mathsf{PS},h}^*(\mathsf{B}) = p(0 + V_{\mathsf{PS},h+1}^*(\mathsf{C})) + (1-p)(-1 + V_{\mathsf{PS},h+1}^*(\mathsf{B})).$$

Since $V_{\mathsf{PS},h}^*(\mathsf{B}) = V_{\mathsf{PS},h+1}^*(\mathsf{B})$, by solving for $V_{\mathsf{PS},h}^*(\mathsf{B})$, we get:

$$V_{\mathsf{PS}}^*(\mathsf{B}) = -\frac{1-p}{p}.$$

Given that, in state $\mathsf{A}$ the optimal policy will choose the best path in expectation, its value function will be:

$$V_{\mathsf{PS},1}^*(\mathsf{A}) = \max\left\{-2, -\frac{1-p}{p}\right\}.$$

**Sleeping MDP as and MDP with LLC.** We finally suppose that we want to handle this Sleeping MDP as if it were a standard MDP with stochastic actions and rewards depending also on the landing state. We start as before by observing that $V_{\mathsf{LLC}}^*(\mathsf{C}) = 0$. Then, we can reason about state $\mathsf{B}$. It is clearly visible that we have to try to go to state $\mathsf{C}$; otherwise, we will continue to pay the cost of staying in $\mathsf{B}$ (for a sufficiently large horizon). Given that, the expected value of this state is $0$ if we play action $F$ and it is available, $-2$ if we play it when it is not available. Formally:

$$V_{\mathsf{LLC},h}^*(\mathsf{B}) = p(0 + V_{\mathsf{LLC},h+1}^*(\mathsf{C})) + (1-p)(-2 + V_{\mathsf{LLC},h+1}^*(\mathsf{B})).$$

Since, $V_{\mathsf{LLC},h+1}^*(\mathsf{B}) = V_{\mathsf{LLC},h}^*(\mathsf{B})$ we get:

$$V_{\mathsf{LLC},h}^*(\mathsf{B}) = -2\frac{1-p}{p}.$$

As before, in state $\mathsf{A}$, we can select the best action, leading to:

$$V_{\mathsf{LLC},1}^*(\mathsf{A}) = \max\left\{-2, -2\frac{(1-p)}{p}\right\}.$$

**Comparison of the Results.** Figure 5 shows the value functions for the three cases for all the values of $p \in [0,1]$. We can observe how, as supported by the intuition, for every $p \in [0,1]$, it is always better to know the action availability in advance ($V_{\mathsf{PE}}^*(\mathsf{A}) \geqslant V_{\mathsf{PS}}^*(\mathsf{A})$), as we have more information and we can take better decisions. This implies that the two notions of regret must differ (given that the two optimal value functions will be different), in particular in terms of the comparator we use, which should be appropriate and reachable. Finally, we observe that the performance of the equivalent MDP integrating the low-level controller is the worst, as expected.

## B. Related Works

In this appendix, we summarize the relevant literature for this work. We start by presenting an overview of the fundamental works in the sleeping MAB literature, as this problem has never been faced in the RL scenario. Then, we introduce works on invalid action masking in RL. Finally, we summarize the main results on regret bounds for standard RL.

**Sleeping MAB.** (Kleinberg et al., 2008; 2010) are the seminal works for the Sleeping MAB setting. In their work, the authors study both the full and partial information settings, considering both the stochastic and adversarial reward models and adversarially chosen action sets. (Kanade et al., 2009) presents the first polynomial time algorithm for MAB with adversarial rewards and stochastic action sets. In the same scenario, (Saha et al., 2020) improves the performance, keeping the computational complexity polynomial. (Kanade & Steinke, 2014) presents the first polynomial time algorithm bandits with both adversarial rewards and action sets, and (Nguyen & Mehta, 2024) achieves near-optimal regret bounds in this scenario. (Chatterjee et al., 2017) studies the setting with adversarial action sets and Bernoulli rewards. (Cortes et al., 2019) extends the Sleeping framework to consider graph feedback. (Gaillard et al., 2023) studies different notions of regret in the sleeping MAB setting, and discusses their relation.

**Invalid Action Masking for RL.** Several works consider the possibility of having not all actions available in all the states (Vinyals et al., 2017). This deterministic masking operation can be done over several types of algorithms such as policy gradient solutions (Huang & Ontañón, 2022) and state-of-the-art deep RL algorithms such as DQN (Mnih et al., 2013). However, in all these works, given a state $s$, we have a deterministic mapping to the action availability (i.e., $C : \mathcal{S} \rightarrow \mathcal{P}(\mathcal{A})$). In this work, instead, we consider way more challenging scenarios, as we have, given a state, probability distribution over the available action sets (i.e., $C : \mathcal{S} \rightarrow \Delta(\mathcal{P}(\mathcal{A}))$), also in the per-episode disclosure scenario, that can be seen as a stochastic generalization of action masking.

**Minimax Regret Bounds for RL.** (Auer et al., 2008; Jaksch et al., 2010) present the first minimax lower bound in the order of $\Omega(\sqrt{DSAT})$ for average reward MDPs with stationary transition probabilities where $D$ is the diameter of the MDP.[9][10] (Domingues et al., 2021) generalize this result by providing a standard proof framework for episodic MDPs and demonstrate a lower bound in the order of $\Omega(H\sqrt{SAT})$ with stage-dependent transitions and $\Omega(\sqrt{HSAT})$ with stage-independent ones. From the algorithmic perspective, (Jaksch et al., 2010) propose UCRL2, which enjoys $\widetilde{\mathcal{O}}(DS\sqrt{AT})$ regret with stage-independent transition. (Azar et al., 2017) propose UCBVI, which enjoys $\widetilde{\mathcal{O}}(\sqrt{HSAT})$ regret with stage-independent transitions and $\widetilde{\mathcal{O}}(H\sqrt{SAT})$ with stage-dependent ones.[11] (Zhang et al., 2024) theoretically improves the result of (Azar et al., 2017), even if preserving the same (optimal, up to logarithmic factors) rate, by reducing the requirement for the minimum $T$ needed in order to match the lower bound.

## C. Omitted Proofs of the Lower Bounds

### C.1. Proof of Theorem 5.1

**Theorem 5.1** (Lower Bound – Per-stage Disclosure: Independent). *For any algorithm $\mathfrak{A}$, there exists an instance of SleMDPs with independent action availability with per-stage disclosure such that, for $T \geqslant \Omega(H^2 SA)$, the per-stage disclosure regret satisfies:*

$$\mathbb{E}\left[R_{PS}(\mathfrak{A}, T)\right] \geqslant \Omega\left(H\sqrt{SAT}\right).$$

*Proof.* This proof closely follows the one of (Domingues et al., 2021, Theorem 9). In order to demonstrate the lower bound for Sleeping MDPs with per-stage disclosure in the case of independent action availability, we first modify the class of hard MDP instances provided in (Domingues et al., 2021, Section 3.1), and then we follow similar derivations to the original proof, thus reporting only the relevant modifications.

**Definition of the Sleeping MDPs class.** We start by modifying the class $\mathcal{C}_{\overline{H}, \varepsilon'}$ provided in (Domingues et al., 2021) to transform it into a specific class of Sleeping MDPs. As in the original proof, the SleMDPs all have three special states: a

---

[9]This result considers a different setting w.r.t. the one of finite-horizon MDPs considered in this work. However, we can generalize this result by observing that $H = \mathcal{O}(D)$, see (Domingues et al., 2021).

[10](Bartlett & Tewari, 2009) present a variant of the lower bound which does not hold in general, see (Jaksch et al., 2010; Osband & Van Roy, 2016) for a detailed discussion.

[11]The result of (Azar et al., 2017) is derived with stage-independent transition. However, we can derive the result by considering a fictitious MDP with augmented state space $\mathcal{S} \times [\![H]\!]$.

*waiting* state $s_w$, a *good* state $s_g$, and a *bad* state $s_b$. The remaining $S - 3$ states are arranged in a $A$-ary tree of depth $d - 1$. The agent starts each episode in $s_w$, and can select to remain in $s_w$ by playing an action $a_w$ up to stage $\overline{H}$, after which it is forced to transition to state $s_{root}$, which is the root of the $A$-ary tree. For every triplet:

$$(h^*, l^*, a^*) \in \{1 + d, \ldots, \overline{H} + d\} \times \mathcal{L} \times \mathcal{A} \backslash \{a_w\},$$

we define a Sleeping MDP $\mathcal{M}_{S,(h^*,l^*,a^*)}$ as follows. The action availabilities in $s_w$ are defined as:

$$C^{\text{ind}}(\mathcal{A}|s_w) = 1, \ \forall h \in [\![H]\!]$$

meaning that all actions are available in the waiting state, and the transition probabilities from $s_w$ are defined as:

$$P_h(s_w|s = s_w, a_{k,h} = a_w) = \mathbb{1}\{h \leqslant \overline{H}\},$$
$$P_h(s_{root}|s = s_w, a_{k,h} = a_w) = 1 - P_h(s_w|s = s_w, a_{k,h} = a_w),$$
$$P_h(s_{root}|s = s_w, a_{k,h} = a) = 1, \forall a \neq a_w, h \in [\![H]\!].$$

Let $\mathcal{I} = \mathcal{S} \backslash \{\{s_w\} \cup \mathcal{L} \cup \{s_g, s_b\}\}$ be the set of internal nodes of the $A$-ary tree, then the action availabilities inside the tree are defined as:

$$C^{\text{ind}}(\mathcal{A}|s) = 1, \forall s \in \mathcal{I}.$$

The transition probabilities for any state in the tree are deterministic: playing the $a$-th action leads deterministically to the $a$-th child node of the current node. For every leaf node $s_l \in \mathcal{L}$, we define the action availabilities such that:

$$\Pr(a_w \in \mathcal{A}_{k,h}(s)) = 1, \ \forall s \in \mathcal{L}, k \in [\![K]\!], h \in [\![H]\!],$$
$$\Pr(a \in \mathcal{A}_{k,h}(s)) = \sum_{\mathcal{B} \in \mathcal{P}(\mathcal{A}) \text{ s.t. } a \in \mathcal{B}} C^{\text{ind}}(\mathcal{B}|s) = c \in [0,1], \ \forall s \in \mathcal{L}, a \in \mathcal{A} \backslash \{a_w\}, k \in [\![K]\!], h \in [\![H]\!].$$

As such, at least one action is guaranteed to be available at every stage of every episode, and every action (except $a_w$) is available with probability $c$. The leaf nodes are the only nodes which can transition to $s_g$ and $s_b$, and they do so according to the following transition probabilities:

$$P_h(s_g|s_i, a) = \frac{1}{2} + \Delta_{(h^*,l^*,a^*)}(h, s_i a),$$
$$P_h(s_b|s_i, a) = 1 - P_h(s_g|s_i, a),$$

where $\Delta_{(h^*,l^*,a^*)}(h, s_i, a) = \varepsilon' \mathbb{1}\{(h, i, a) = (h^*, l^*, a^*)\}$. Finally, we also define a reference Seeping MDP $\mathcal{M}_{S,0}$ which has the same structure as the SleMDPs defined above, but with $\Delta_0(h, s_i, a) = 0$. The rewards are defined as:

$$R(s, a, h) = \mathbb{1}\{s = s_g, h \geqslant \overline{H} + d + 1\}, \forall a \in \mathcal{A}.$$

**Regret of an algorithm $\mathfrak{A}$ in $\mathcal{M}_{S,(h^*,l^*,a^*)}$.** Following the same reasoning as in (Domingues et al., 2021), it is clear to see that the learner is required to learn the optimal trajectory, which enables it to play action $a^*$ in leaf node $s_{l*}$ at stage $h^*$. However, this trajectory is only achievable with probability $c$, due to the availability of action $a^*$ in node $s_{l*}$. Notice that, if the optimal trajectory is not available in an episode, then the learner does not incur in any regret, as neither the agent nor the optimal policy can achieve it. We now follow the proof of Th. 9 of (Domingues et al., 2021), adapting to the Sleeping MDP setting. We report only the relevant modifications. Let $S_{k,h}$ and $A_{k,h}$ be the random variables that represent, respectively, the state occupied and the action selected at stage $h$ of episode $k$. We start by observing that the average reward gathered by an algorithm $\mathfrak{A}$ is again defined as:

$$\mathbb{E}_{(h^*,l^*,a^*)}\left[\sum_{k=1}^{K}\sum_{h=1}^{H} R(S_{k,h}, A_{k,h}, h)\right] = (H - \overline{H} - d)\sum_{k=1}^{K} \Pr(S_{k,\overline{H}+d+1} = s_g).$$

For any stage $h \in [\![1 + d, \overline{H} + d]\!]$, we can rewrite Eq. (7) of (Domingues et al., 2021) as:

$$\Pr_{(h^*,l^*,a^*)}(S_{k,h+1} = s_g) = \Pr_{(h^*,l^*,a^*)}(s_{k,h} = s_g) + \frac{1}{2}\Pr_{(h^*,l^*,a^*)}(S_{k,h} \in \mathcal{L}) +$$
$$+ \mathbb{1}\{h = h^*\}\Pr_{(h^*,l^*,a^*)}(S_{k,h} = s_{l*}, A_{k,h} = a^*|a^* \in \mathcal{A}_{k,h})$$
$$= \Pr_{(h^*,l^*,a^*)}(s_{k,h} = s_g) + \frac{1}{2}\Pr_{(h^*,l^*,a^*)}(S_{k,h} \in \mathcal{L}) +$$

$$+ \frac{1}{c} \mathbb{1}\{h = h^*\} \Pr_{(h*,l*,a*)} (S_{k,h} = s_{l*}, A_{k,h} = a^*), \tag{8}$$

where (8) is obtained by applying Bayes' rule and observing that $\Pr_{(h*,l*,a*)}(a^* \in \mathcal{A}_{k,h} | S_{k,h} = s_{l*}, A_{k,h} = a^*) = 1$. Following the proof, we obtain that the optimal value in any of the SleMDPs is $\rho^* = (H - \overline{H} - d(1/2 + \varepsilon))/c$. We can then rewrite the suffered regret as:

$$R_T(\mathfrak{A}) = K(H - \overline{H} - d)\varepsilon \left( 1 - \frac{1}{K} \mathbb{E}_{(h*,l*,a*)}[Z_{K,(h*,l*,a*)}] \right),$$

where $Z_{K,(h*,l*,a*)} = \sum_{k=1}^{K} \mathbb{1}\{S_{k,h*} = s_{l*}, A_{k,h*} = a^* | a^* \in \mathcal{A}_{k,h*}\}$. Observe that:

$$\frac{1}{K} \mathbb{E}_{(h*,l*,a*)}[Z_{K,(h*,l*,a*)}] = \frac{1}{K} \sum_{k=1}^{K} \mathbb{E}_{(h*,l*,a*)}[\mathbb{1}\{S_{k,h*} = s_{l*}, A_{k,h*} = a^* | a^* \in \mathcal{A}_{k,h*}\}]$$

$$= \frac{1}{K} \sum_{k=1}^{K} \Pr_{(h*,l*,a*)} (S_{k,h*} = s_{l*}, A_{k,h*} = a^* | a^* \in \mathcal{A}_{k,h*})$$

$$= \frac{1}{K} \sum_{k=1}^{K} \frac{1}{c} \Pr_{(h*,l*,a*)} (S_{k,h*} = s_{l*}, A_{k,h*} = a^*)$$

$$= \frac{1}{cK} \mathbb{E}_{(h*,l*,a*)}[N_{K,(h*,l*,a*)}],$$

where $N_{K,(h*,l*,a*)}$ is defined as in (Domingues et al., 2021). We can then bound the maximum regret over all possible instances as:

$$\max_{(h*,l*,a*)} R_T(\mathfrak{A}) \geqslant K(H - \overline{H} - d)\, \varepsilon \left( 1 - \frac{1}{\overline{H}LAK} \sum_{(h*,l*,a*)} \mathbb{E}_{(h*,l*,a*)}[Z_{K,(h*,l*,a*)}] \right)$$

$$\geqslant K(H - \overline{H} - d)\, \varepsilon \left( 1 - \frac{1}{\overline{H}LAKc} \sum_{(h*,l*,a*)} \mathbb{E}_{(h*,l*,a*)}[N_{K,(h*,l*,a*)}] \right).$$

Following the derivation, we finally obtain:

$$\max_{(h*,l*,a*)} R_T(\mathfrak{A}) \geqslant K(H - \overline{H} - d)\varepsilon \left( 1 - \frac{1}{\overline{H}LAc} - \frac{\sqrt{2}\varepsilon\sqrt{\overline{H}LAK}}{\overline{H}LAc} \right).$$

Clearly, as $c$ appears only at the denominator of negative terms, and $c \in [0, 1]$ by definition, the value of $c$ that maximizes the regret is $c = 1$. Intuitively, given a finite number of episodes, and given that the agent does not pay any regret if the optimal trajectory is not available, the case in which the agent can pay the maximum regret is that in which the optimal trajectory is available in every episode. Finally, we conclude the proof by plugging in the same optimal values for $\varepsilon$ and $\overline{H}$, obtaining a lower bound of $\Omega(H\sqrt{SAT})$. $\qquad\square$

### C.2. Proof of Theorem 4.2

**Theorem 4.2** (Lower Bound – Per-stage Disclosure: Markovian). *For any algorithm $\mathfrak{A}$, there exists an instance of Sleeping MDP with per-stage disclosure and Markovian action availability such that, for $T \geqslant \Omega(HSA2^A)$ and $H \geqslant \Omega(A)$, the per-stage disclosure policy regret satisfies:*

$$\mathbb{E}\left[R_{PS}(\mathfrak{A}, T)\right] \geqslant \Omega \left( H\sqrt{SAT2^{A/2}} \right).$$

*Proof.* We start the proof by describing the instance. The goal of this proof is to show that an exponential component in the regret is not avoidable if we consider Markovian action availability and per-stage disclosure.

### Construction of the Instances

To build the "hard instances" of SleMDPs we need in order to get the lower bound exponential in the number of actions, we consider two building blocks, as depicted in Figure 6. In these instances, we consider $H > \frac{A}{2} + 2 + \log_A S$, we consider

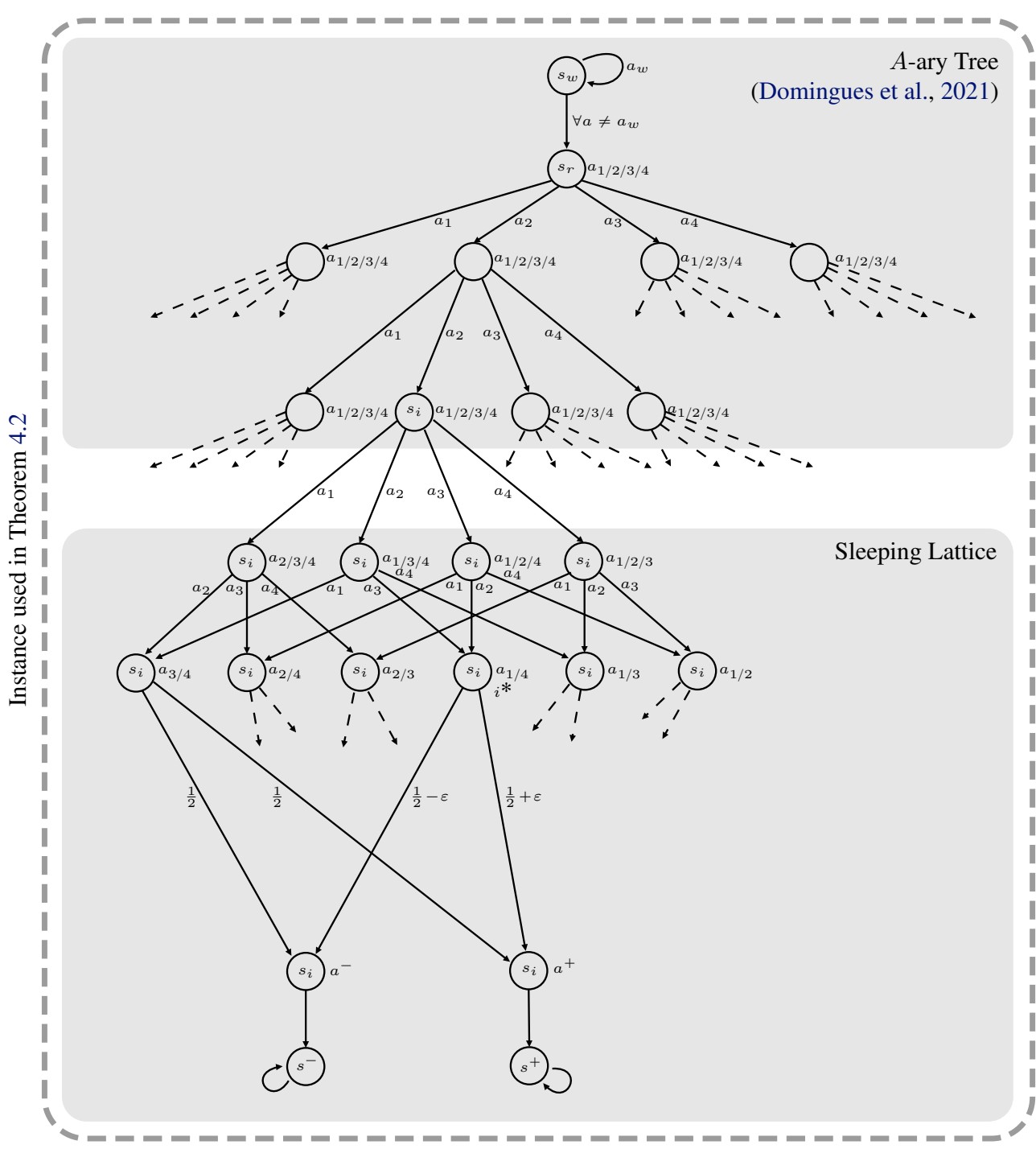

Figure 6. Instance used in the lower bound construction for $C = C^{\text{Markov}}$ with per-stage disclosure.

the state set to be composed of $S$ states and 3 additional states: $s_w$, $s^+$, and $s^-$ which we will define later. Moreover, we consider the action set to be composed by $A + 2$ action, with $A$ even, where we have 2 special actions $a^+$ and $a^-$ we will define later.

$A$-**ary Tree.** The first building block is the tree structure instance proposed by (Domingues et al., 2021), depicted in the grey area in Figure 6. In this structure, we have a starting state $s_w$ from which we have to exit at the proper moment, which is designed to get the proper dependency on the horizon $H$. Then, we have a $A$-ary tree structure that we consider in the same way as (Domingues et al., 2021). For this part, we consider that all the actions are available in all the states. Moreover, we consider deterministic transitions. We refer the reader to (Domingues et al., 2021) for further details. This construction gives us a lower bound in the order of $\Omega\left(H\sqrt{SAT}\right)$ and, given that we consider full action availability, we are in the same scenario as for the original paper, so we avoid to report all the derivation, and we refer the reader to (Domingues et al., 2021, Theorem 9) for the analysis.

**Sleeping MDP Lattice.** The second building block is a *lattice* structure to generate the hard instance in terms of Markovian action availability sets. In every leaf of the $A$-ary tree, we add a lattice of depth $A/2$ designed as follows. We start, as depicted in Figure 6, in a generic leaf state $s_i$ where we will have all the actions available to build the lattice. In the example, the available actions are reported next to the state. When we are in the lattice and we make an action $a_i$, such an action will be no more available in the next stage, while we remain in the same $s_i \in \mathcal{S}$, so the only evolution we trigger is a deterministic evolution of the available action set $\mathcal{B}$ which will become $\mathcal{B}\backslash\{a_i\}$. We stop the construction of the lattice when we are in the layer of the lattice with the maximum extension. It is simple to verify that in such a layer, we have $n = \binom{A}{A/2}$ different availability combinations. Then, at step $\frac{A}{2}+1$ in the lattice, we have that the different instances become distinguishable, and we will have an action set $i^* \in [\![n]\!]$ which is the optimal one. We call $m_{i*}$ the instance in which $i^*$ is the optimum. Now, if we are in the optimal action availability set, we have probability $\frac{1}{2} + \varepsilon$ to remain in $s_i$ and have the chance of playing action $a^+$, and we have probability $\frac{1}{2} - \varepsilon$ to remain in $s_i$ and having available only action $a^-$. Then, if we play $a^+$ in $s_i$ in the proper stage $h \in [\![H]\!]$, we get reward 1; otherwise we get 0. If $i \in [\![n]\!], i \neq i^*$, we have the same probability of going to good and bad action set, i.e., $\frac{1}{2}$.

Given that, an instance is characterized by a tuple $(s^*, h^*, a^*, \mathcal{B}^*)$. While the first 3 terms are the same of (Domingues et al., 2021), the last term is the optimal available action set, i.e., the one allowing us, if properly triggered, to play $a^+$ and get reward 1. $\mathcal{B}^*$ is the set corresponding to $i^* \in [\![n]\!]$.

**Analysis**

Given that we already know the result of the first building block (i.e., the $A$-ary tree), we dedicate our efforts to getting the exponential dependency on $A$, then combining the results together just requires some straightforward algebra.

Fix $i^* \in [\![n]\!]$. We call $N_i(K)$ the number of times we observed $i$ over $K$ episodes. We define $i^- \in [\![n]\!]$ the ones such that:

$$i^- \in \underset{i \in [\![n]\!]\backslash\{i*\}}{\arg\min} \mathbb{E}_{m_{i*}}[N_i(K)].$$

Given that, since we are in a loop-free structure, we know that:

$$\sum_{i \in [\![n]\!]\backslash\{i*\}} \mathbb{E}_{m_{i*}}[N_i(K)] \leqslant K,$$

for the so-called "averaging hammer" we have:

$$\mathbb{E}_{m_{i*}}[N_{i-}(K)] \leqslant \frac{K}{n-1}.$$

We now consider instance $m_{i^-}$, which is defined similarly to $i^*$ but this time the best available action set, i.e., the one with a probability of activating $a^+$ equal to $\frac{1}{2} + 2\varepsilon$, is $i^-$. We highlight that in this instance we have a multiplicative term 2 before $\varepsilon$.

We can now compute the regret and optimize the value of $\varepsilon$. We use the notation $R_T(m)$ to indicate the regret after $T$ interactions on instance $m$:

$$\max\{R_T(m_{i*}), R_T(m_{i-})\} \geqslant \frac{1}{2}\left(R_T(m_{i*}) + R_T(m_{i-})\right)$$

$$\geqslant \frac{\Delta K}{4}\left(\Pr_{m_{i*}}\left(N_{i*}(K) \leqslant \frac{K}{2}\right) + \Pr_{m_{i-}}\left(N_{i*}(K) \geqslant \frac{K}{2}\right)\right)$$

$$\geqslant \frac{\Delta K}{8} \exp\left( \mathbb{E}_{m_{i*}} [N_{i^-}(K)] \; D_{\mathrm{KL}}\left(\frac{1}{2} + 2\varepsilon, \frac{1}{2} + \varepsilon\right) \right) \tag{9}$$

$$\geqslant \frac{\Delta K}{8} \exp\left( -\frac{K\varepsilon^2}{n-1} \right), \tag{10}$$

where $\Delta$ is the value function we will compute later, line (9) is the Bretagnolle-Huber inequality (see Lattimore & Szepesvári, 2020, Theorem 14.2), and line (10) holds for sufficiently small $\varepsilon$.

We can now compute $\Delta$:

$$V^*_{m_{i*}} = \left(\frac{1}{2} + \varepsilon\right) \cdot 1 + \left(\frac{1}{2} - \varepsilon\right) \cdot 0 = \varepsilon$$

$$V^*_{m_{i^-}} = \left(\frac{1}{2} + 2\varepsilon\right) \cdot 1 + \left(\frac{1}{2} - 2\varepsilon\right) \cdot 0 = 2\varepsilon$$

so we have $\Delta = \varepsilon$.

Now we have to choose $\varepsilon$ and we choose $\varepsilon = \sqrt{\frac{n-1}{K}}$ and we get:

$$\max\{R_T(m_{i*}), R_T(m_{i^-})\} \geqslant \frac{\varepsilon K}{8} \exp\left( -\frac{K\varepsilon^2}{n-1} \right)$$

$$\geqslant \frac{e^{-1}}{8} \sqrt{K(n-1)}$$

$$\geqslant \Omega\left( \sqrt{K 2^{A/2}} \right),$$

where the last inequality is derived after having observed that:

$$n = \binom{A}{A/2} \geqslant \left(\frac{A}{A/2}\right)^{A/2} \geqslant 2^{A/2}$$

Now, applying the same reasoning as (Domingues et al., 2021) we can retrieve the multiplicative factor $\Omega\left(H^{3/2}\sqrt{SA}\right)$, leading to a bound in the order of $\Omega\left(H\sqrt{SAT}2^{A/2}\right)$. This concludes the proof.

$\square$

# D. Proof of Theorem 5.2

In this appendix, we provide the formal proof of Theorem 5.2. The proof follows the one of (Azar et al., 2017, Theorem 2) and as such, some of the original lemmas which are used as is are reported to increase the readability of the proof.

## D.1. Notation

We now collect the notation necessary for the understanding of the proof of Theorem 5.2.

| Symbol | Meaning |
| --- | --- |
| $\mathcal{S}$ | State space |
| $\mathcal{A}$ | Action space |
| $P$ | Transition distribution |
| $C$ | Action set availability distribution |
| $R$ | Reward function |
| $H$ | Length of the episode |
| $K$ | Total number of episodes |
| $T$ | Total number of steps |
| $T_k$ | Total number of steps up to episode $k$ |
| $S$ | Cardinality of the state space |
| $A$ | Cardinality of the action space |

| | |
|---|---|
| $s_{k,h}$ | State occupied at stage $h$ of episode $k$ |
| $a_{k,h}^{\pi_k(\mathcal{B})}$ | Action played at stage $h$ of episode $k$ under policy $\pi_k$ with action set $\mathcal{B}$ available |
| $N_k(s,a)$ | Number of visits to state-action pair $(s,a)$ up to episode $k$ |
| $N_k(s,a,s')$ | Number of transitions to state $s'$ from state $s$ after playing action $a$, up to episode $k$ |
| $N_{k,h}(s)$ | Number of visits to state $s$ at stage $h$ up to episode $k$ |
| $N_{k,h}(s,a)$ | Number of visits to state-action pair $(s,a)$ at stage $h$ up to episode $k$ |
| $\widehat{P}_k$ | Estimated transition distribution |
| $\widehat{C}$ | Estimated action set availability distribution |
| $b^Q$ | State-action value function exploration bonus |
| $\overline{b}_{k,h}^Q(s)$ | $\min\{\frac{2900^2 H^3 S^3 A 2^A L^3}{N_{k,h}(s)}, H^2\}$ |
| $b^V$ | State value function exploration bonus |
| $\overline{b}_{k,h}^V(s,\mathcal{B})$ | $\min\{\frac{1350^2 H^3 S^3 A 2^A L^3}{N'_{i,j}(s_{i,j},a_{i,j}^{\pi_i(\mathcal{B})})}, H^2\}$ |
| $\pi_k$ | Policy played during episode $k$ |
| $\pi^*$ | Optimal policy |
| $Q_h^*$ | State-action value function of the optimal policy |
| $Q_h^\pi$ | State-action value function following policy $\pi$ |
| $\widehat{Q}_{k,h}$ | Optimistic state-action value function |
| $V_h^*$ | Value function of the optimal policy at stage $h$ |
| $V_h^\pi$ | Value function under policy $\pi$ at stage $h$ |
| $\widehat{V}_{k,h}$ | Optimistic estimator of the optimal value function at stage $h$ of episode $k$ |
| $\Delta_{k,h}^V(s)$ | Regret in state $s$, at stage $h$ of episode $k$, following policy $\pi_k$ |
| $\widetilde{\Delta}_{k,h}^V(s)$ | Pseudo-regret in state $s$, at stage $h$ of episode $k$, following policy $\pi_k$ |
| $\Delta_{k,h}^Q(s,a)$ | $Q_h^*(s,a) - Q_h^{\pi_k}(s,a)$ |
| $\widetilde{\Delta}_{k,h}^Q(s,a)$ | $\widehat{Q}_{k,h}(s,a) - Q_h^{\pi_k}(s,a)$ |
| $\mathcal{E}$ | Concentration inequalities event |
| $\Omega_{k,h}$ | Optimism event |
| $\varepsilon^V, \overline{\varepsilon}^V, \xi^V, \overline{\xi}^V, \varepsilon^Q, \overline{\varepsilon}^Q$ | Martingale differences sequences |
| $[k]_{\text{typ}}, [k]_{\text{typ},s}, [k]_{\text{typ},s,a}$ | Sets of typical episodes |
| $\mathcal{H}_{k,h}$ | History of the interactions up to, and including, stage $h$ of episode $k$, *not including* the observed available action set $\mathcal{B}_{k,h}$ |
| $\mathcal{H}_{k,h,\mathcal{B}}$ | History of the interactions up to, and including, stage $h$ of episode $k$, *including* the observed available action set $\mathcal{B}_{k,h}$ |
| $L$ | Logarithmic term $\log(80 H S^2 A 2^A T/\delta)$ |
| $G$ | Logarithmic term $\log(HSAT)$ |
| $\mathbb{V}_h^{\pi_k}$ | Next-state variance of $V^{\pi_k}$ |
| $\mathbb{V}_h^*$ | Next-state variance of $V^*$ |
| $\widehat{\mathbb{V}}_{k,h}$ | Empirical next-state variance of $\widehat{V}_{k,h}$ |
| $\widehat{\mathbb{V}}_{k,h}^*$ | Empirical next-state variance of $V^*$ |
| $\mathbb{Q}_h^{\pi_k}$ | Variance of $Q^{\pi_k}$ |
| $\mathbb{Q}_h^*$ | Variance of $Q^*$ |
| $\widehat{\mathbb{Q}}_{k,h}$ | Empirical variance of $\widehat{Q}_{k,h}$ |
| $\widehat{\mathbb{Q}}_{k,h}^*$ | Empirical variance of $Q^*$ |

Table 2: Table of notation

We now restate the definitions of the Martingale Difference Sequences:

$$\varepsilon_{k,h}^V := \left( P\widetilde{\Delta}_{k,h+1}^V \right)(s_{k,h}, a_{k,h}^{\pi_k(\mathcal{B}_{k,h})}) - \widetilde{\Delta}_{k,h+1}^V(s_{k,h+1}),$$

$$\overline{\varepsilon}^V_{k,h} := \sum_{s'\in\mathcal{S}} P(s'|s_{k,h},a^{\pi_k(\mathcal{B}_{k,h})}_{k,h})\sqrt{\frac{\mathbb{I}\{s'\in[s]_{k,h}\}}{N_k(s_{k,h},a^{\pi_k(\mathcal{B}_{k,h})}_{k,h})P(s'|s_{k,h},a^{\pi_k(\mathcal{B}_{k,h})}_{k,h})}}\widetilde{\Delta}^V_{k,h+1}(s')$$

$$-\sqrt{\frac{\mathbb{I}\{s_{k,h+1}\in[s]_{k,h}\}}{N_k(s_{k,h},a^{\pi_k(\mathcal{B}_{k,h})}_{k,h})P(s'|s_{k,h},a^{\pi_k(\mathcal{B}_{k,h})}_{k,h})}}\widetilde{\Delta}^V_{k,h+1}(s_{k,h+1}),$$

$$\xi^V_{k,h} := \sum_{\mathcal{B}\in\mathcal{P}(\mathcal{A})} C^{\mathsf{ind}}(\mathcal{B}|s_{k,h})\left(\sum_{s'\in\mathcal{S}} P(s'|s,a^{\pi_k(\mathcal{B})}_{k,h})\widetilde{\Delta}^V_{k,h+1}(s')\right) - \widetilde{\Delta}^V_{k,h+1}(s_{k,h+1}),$$

$$\overline{\xi}^V_{k,h} := \sum_{\mathcal{B}\in\mathcal{P}(\mathcal{A})} C^{\mathsf{ind}}(\mathcal{B}|s_{k,h})\left(\sum_{s'\in\mathcal{S}} P(s'|s,a^{\pi_k(\mathcal{B})}_{k,h})\sqrt{\frac{\mathbb{I}\{s'\in[s]_{k,h}\}}{N_k(s_{k,h},a^{\pi_k(\mathcal{B})}_{k,h})P(s'|s_{k,h},a^{\pi_k(\mathcal{B})}_{k,h})}}\widetilde{\Delta}^V_{k,h+1}(s')\right)$$

$$-\sqrt{\frac{\mathbb{I}\{s_{k,h+1}\in[s]_{k,h}\}}{N_k(s_{k,h},a^{\pi_k(\mathcal{B})}_{k,h})P(s'|s_{k,h},a^{\pi_k(\mathcal{B})}_{k,h})}}\widetilde{\Delta}^V_{k,h+1}(s_{k,h+1}),$$

$$\varepsilon^Q_{k,h} := \left(C^{\mathsf{ind}}\widetilde{\Delta}^Q_{k,h}\right)(s_{k,h}) - \widetilde{\Delta}^Q_{k,h}(s_{k,h},a^{\pi_k(\mathcal{B}_{k,h})}_{k,h})$$

$$\overline{\varepsilon}^Q_{k,h} := \sum_{\mathcal{B}\in\mathcal{P}(\mathcal{A})} C^{\mathsf{ind}}(\mathcal{B}|s_{k,h})\sqrt{\frac{\mathbb{I}\{\mathcal{B}\in[\mathcal{B}]_{k,h}\}}{N_k(s_{k,h})C^{\mathsf{ind}}(\mathcal{B}|s_{k,h})}}\widetilde{\Delta}^Q_{k,h}(s_{k,h},a^{\pi_k(\mathcal{B})}_{k,h})$$

$$-\sqrt{\frac{\mathbb{I}\{\mathcal{B}_{k,h}\in[\mathcal{B}]_{k,h}\}}{N_k(s_{k,h})C^{\mathsf{ind}}(\mathcal{B}_{k,h}|s_{k,h})}}\widetilde{\Delta}^Q_{k,h}(s_{k,h},a^{\pi_k(\mathcal{B}_{k,h})}_{k,h}),$$

and of the variance terms:

$$\mathbb{V}^{\pi_k}_h(s,a) := \operatorname*{\mathbb{V}ar}_{s'\sim P(\cdot|s,a)}[V^{\pi_k}_h(s')],$$

$$\mathbb{V}^*_h(s,a) := \operatorname*{\mathbb{V}ar}_{s'\sim P(\cdot|s,a)}[V^*_h(s')],$$

$$\widehat{\mathbb{V}}_{k,h}(s,a) := \operatorname*{\mathbb{V}ar}_{s'\sim\widehat{P}_k(\cdot|s,a)}[\widehat{V}_{k,h}(s')],$$

$$\widehat{\mathbb{V}}^*_{k,h}(s,a) := \operatorname*{\mathbb{V}ar}_{s'\sim\widehat{P}_k(\cdot|s,a)}[V^*_h(s')],$$

$$\mathbb{Q}^{\pi_k}_h(s) := \operatorname*{\mathbb{V}ar}_{\mathcal{B}\sim C^{\mathsf{ind}}(\cdot|s)}[Q^{\pi_k}_h(s,\pi_{k,h}(s,\mathcal{B})],$$

$$\mathbb{Q}^*_h(s) := \operatorname*{\mathbb{V}ar}_{\mathcal{B}\sim C^{\mathsf{ind}}(\cdot|s)}[Q^*_h(s,\pi^*_h(s,\mathcal{B}))],$$

$$\widehat{\mathbb{Q}}_{k,h}(s) := \operatorname*{\mathbb{V}ar}_{\mathcal{B}\sim\widehat{C}^{\mathsf{ind}}_k(\cdot|s)}[\widehat{Q}_{k,h}(s,\pi_{k,h}(s,\mathcal{B}))],$$

$$\widehat{\mathbb{Q}}^*_{k,h}(s) := \operatorname*{\mathbb{V}ar}_{\mathcal{B}\sim\widehat{C}^{\mathsf{ind}}_k(\cdot|s)}[Q^*_h(s,\pi^*_h(s,\mathcal{B}))].$$

For ease of notation, we will employ the following notation throughout the appendix. Let $F : X \to \Delta(Y)$ be a probability distribution over a set $Y$ conditioned to a set $X$. Let $\widehat{F} : X \to \Delta(Y)$ be an estimator of $F$, and let $G : X \times Y \to \mathbb{R}$ be a real-valued function. Then the define the following notations:

$$(FG)(x) := \sum_{y\in Y} F(y|x)G(x,y),$$

$$\left((\widehat{F}-F)G\right)(x) := \sum_{y\in Y}\left(\widehat{F}(y|x) - F(y|x)\right)G(x,y).$$

Similarly, let $F : X \times Y \to \Delta(Z)$, $\widehat{F} : X \times Y \to \Delta(Z)$, and $G : Z \to \mathbb{R}$ be defined with the same meaning as above, then

we define the following notations:

$$(FG)(x,y) := \sum_{z \in Z} F(z|x,y)G(z),$$

$$\left((\widehat{F} - F)G\right)(x,y) := \sum_{z \in Z} \left(\widehat{F}(z|x,y) - F(y|x,y)\right)G(z).$$

### D.2. High Probability Events

In this section, we state the high probability events $\Omega_{k,h}$ and $\mathcal{E}$. For ease of reference, we employ the same notation as (Azar et al., 2017).

Let $\Omega_{k,h}$ be the set of events:

$$\Omega_{k,h} := \left\{ \widehat{V}_{i,j}(s) \geqslant V_j^*(s) \wedge \widehat{Q}_{i,j}(s,a) \geqslant Q_j^*(s,a), \forall (i,j) \in [k,h]_{\text{hist}}, s \in \mathcal{S}, a \in \mathcal{A} \right\},$$

for $k \in [\![K]\!]$ and $h \in [\![H]\!]$, where:

$$[k,h]_{\text{hist}} := \{(i,j) : i \in [\![K]\!], j \in [\![H]\!], (i < k) \vee (i = k, j \geqslant h)\},$$

under which optimism holds.

Event $\mathcal{E}$ is the event defined as:

$$\mathcal{E} := \mathcal{E}_{\widehat{P}} \bigcap \mathcal{E}_{\widehat{C}^{\text{ind}}} \bigcap \bigcap_{\substack{k \in [\![K]\!] \\ h \in [\![H]\!] \\ s \in \mathcal{S} \\ a \in \mathcal{A} \\ \mathcal{B} \in \mathcal{P}(\mathcal{A})}} \left\{ \mathcal{E}_{\text{az}}(\mathcal{F}_{\widetilde{\Delta}^V,k,h}, H, L) \bigcap \mathcal{E}_{\text{az}}(\mathcal{F}_{\widetilde{\Delta}^V,k,h,s}, H, L) \bigcap \mathcal{E}_{\text{az}}(\mathcal{F}_{\widetilde{\Delta}^V,k,h,s,a}, H, L) \right.$$

$$\bigcap \mathcal{E}_{\text{az}}(\mathcal{F}'_{\widetilde{\Delta}^V,k,h}, \frac{1}{\sqrt{L}}, L) \bigcap \mathcal{E}_{\text{az}}(\mathcal{F}'_{\widetilde{\Delta}^V,k,h,s}, \frac{1}{\sqrt{L}}, L) \bigcap \mathcal{E}_{\text{az}}(\mathcal{F}'_{\widetilde{\Delta}^V,k,h,s,a}, \frac{1}{\sqrt{L}}, L)$$

$$\bigcap \mathcal{E}_{\text{az}}(\mathcal{F}_{\widetilde{\Delta}^V,k,h,\mathcal{B}}, H, L) \bigcap \mathcal{E}_{\text{az}}(\mathcal{F}_{\widetilde{\Delta}^V,k,h,\mathcal{B},s}, H, L) \bigcap \mathcal{E}_{\text{az}}(\mathcal{F}_{\widetilde{\Delta}^V,k,h,\mathcal{B},s,a}, H, L)$$

$$\bigcap \mathcal{E}_{\text{az}}(\mathcal{F}'_{\widetilde{\Delta}^V,k,h,\mathcal{B}}, \frac{1}{\sqrt{L}}, L) \bigcap \mathcal{E}_{\text{az}}(\mathcal{F}'_{\widetilde{\Delta}^V,k,h,\mathcal{B},s}, \frac{1}{\sqrt{L}}, L) \bigcap \mathcal{E}_{\text{az}}(\mathcal{F}'_{\widetilde{\Delta}^V,k,h,\mathcal{B},s,a}, \frac{1}{\sqrt{L}}, L)$$

$$\bigcap \mathcal{E}_{\text{az}}(\mathcal{F}_{\widetilde{\Delta}^Q,k,h}, H, L) \bigcap \mathcal{E}_{\text{az}}(\mathcal{F}_{\widetilde{\Delta}^Q,k,h,s}, H, L) \bigcap \mathcal{E}_{\text{az}}(\mathcal{F}_{\widetilde{\Delta}^Q,k,h,s,a}, H, L)$$

$$\bigcap \mathcal{E}_{\text{az}}(\mathcal{F}'_{\widetilde{\Delta}^Q,k,h}, \frac{1}{\sqrt{L}}, L) \bigcap \mathcal{E}_{\text{az}}(\mathcal{F}'_{\widetilde{\Delta}^Q,k,h,s}, \frac{1}{\sqrt{L}}, L) \bigcap \mathcal{E}_{\text{az}}(\mathcal{F}'_{\widetilde{\Delta}^Q,k,h,s,a}, \frac{1}{\sqrt{L}}, L)$$

$$\bigcap \mathcal{E}_{\text{fr}}(\mathcal{G}_{\mathbb{V},k,h}, H^4 T_k, H^3, L) \bigcap \mathcal{E}_{\text{fr}}(\mathcal{G}_{\mathbb{V},k,h,s}, H^5 N_{k,h}, H^3, L)$$

$$\bigcap \mathcal{E}_{\text{fr}}(\mathcal{G}_{\mathbb{V},k,h,s,a}, H^5 N_{k,h}, H^3, L) \bigcap \mathcal{E}_{\text{fr}}(\mathcal{G}_{\mathbb{Q},k,h}, H^4 T_k, H^3, L)$$

$$\bigcap \mathcal{E}_{\text{fr}}(\mathcal{G}_{\mathbb{Q},k,h,s}, H^5 N_{k,h}, H^3, L) \bigcap \mathcal{E}_{\text{fr}}(\mathcal{G}_{\mathbb{Q},k,h,s,a}, H^5 N_{k,h}, H^3, L)$$

$$\bigcap \mathcal{E}_{\text{az}}\left(\mathcal{F}_{\overline{b}^V,k,h}, H^2, L\right) \bigcap \mathcal{E}_{\text{az}}\left(\mathcal{F}_{\overline{b}^V,k,h,s}, H^2, L\right)$$

$$\bigcap \mathcal{E}_{\text{az}}\left(\mathcal{F}_{\overline{b}^V,k,h,s,a}, H^2, L\right) \bigcap \mathcal{E}_{\text{az}}\left(\mathcal{F}_{\overline{b}^Q,k,h}, H^2, L\right)$$

$$\left. \bigcap \mathcal{E}_{\text{az}}\left(\mathcal{F}_{\overline{b}^Q,k,h,s}, H^2, L\right) \bigcap \mathcal{E}_{\text{az}}\left(\mathcal{F}_{\overline{b}^Q,k,h,s,a}, H^2, L\right) \right\}.$$

The proof that event $\mathcal{E}$ holds with probability at least $1 - \delta$ directly follows from Lemma 1 of (Azar et al., 2017).

We now state the definitions of the events that compose $\mathcal{E}$. Events $\mathcal{E}_{\widehat{P}}$ and $\mathcal{E}_{\widehat{C}^{\mathrm{ind}}}$ concern the estimation of the transition probability and of the action availability distributions:

$$\mathcal{E}_{\widehat{P}} := \{\widehat{P}_k(s'|s,a) \in \mathfrak{P}(k,h,N_k(s,a),s,a,s'), \forall k \in [\![K]\!], h \in [\![H]\!], (s,a,s') \in \mathcal{S} \times \mathcal{A} \times \mathcal{S}\},$$

$$\mathcal{E}_{\widehat{C}^{\mathrm{ind}}} := \{\widehat{C}_k^{\mathrm{ind}}(\mathcal{B}|s) \in \mathfrak{C}(k,h,N_k(s),s,\mathcal{B}), \forall k \in [\![K]\!], h \in [\![H]\!], s \in \mathcal{S}, \mathcal{B} \in \mathcal{P}(\mathcal{A})\}.$$

$\mathfrak{P}(k,h,n,s,a,s')$ is the subset of the set of all transition probability distributions $\mathfrak{P}$ such that:

$$\mathfrak{P}(k,h,n,s,a,s') := \left\{ \widetilde{P}(\cdot|s,a) \in \mathfrak{P} : \|\widetilde{P}(\cdot|s,a) - P(\cdot|s,a)\|_1 \leqslant 2\sqrt{\frac{SL}{n}}, \right. \tag{11}$$

$$\left| \sum_{s' \in \mathcal{S}} (\widetilde{P}(s'|s,a) - P(s'|s,a))V_h^*(s') \right|$$

$$\leqslant \min \left\{ \sqrt{\frac{2\widehat{\mathbb{V}}_{k,h+1}^*(s,a)L}{n}} + \frac{7HL}{3(n-1)}, \sqrt{\frac{2\mathbb{V}_{h+1}^*(s,a)L}{n}} + \frac{2HL}{3n} \right\}, \tag{12}$$

$$\left. \left| \widetilde{P}(s'|s,a) - P(s'|s,a) \right| \leqslant \sqrt{\frac{2P(s'|s,a)(1 - P(s'|s,a))L}{n}} + \frac{2L}{3n} \right\}, \tag{13}$$

and $\mathfrak{C}(k,h,N_k(s),s,\mathcal{B})$ is the subset of all the action availability distributions $\mathfrak{C}$ such that:

$$\mathfrak{C}(k,h,N_k(s),s,\mathcal{B}) := \left\{ \widetilde{C}^{\mathrm{ind}}(\cdot|s) \in \mathfrak{C} : \|\widetilde{C}^{\mathrm{ind}}(\cdot|s) - C^{\mathrm{ind}}(\cdot|s)\|_1 \leqslant 2\sqrt{\frac{2^A L}{n}}, \right. \tag{14}$$

$$\left| \sum_{\mathcal{B} \in \mathcal{P}(\mathcal{A})} (\widetilde{C}^{\mathrm{ind}}(\mathcal{B}|s) - C^{\mathrm{ind}}(\mathcal{B}|s))Q_h^*(s, a_{k,h}^{\pi^*(\mathcal{B})}) \right|$$

$$\leqslant \min \left\{ \sqrt{\frac{2\widehat{\mathbb{Q}}_{k,h}^*(s)L}{n}} + \frac{7HL}{3(n-1)}, \sqrt{\frac{2\mathbb{Q}_h^*(s)L}{n}} + \frac{2HL}{3n} \right\}, \tag{15}$$

$$\left. \left| \widetilde{C}^{\mathrm{ind}}(\mathcal{B}|s) - C^{\mathrm{ind}}(\mathcal{B}|s) \right| \leqslant \sqrt{\frac{2C^{\mathrm{ind}}(\mathcal{B}|s)(1 - C^{\mathrm{ind}}(\mathcal{B}|s))L}{n}} + \frac{2L}{3n} \right\}, \tag{16}$$

where Eq. (11) and Eq. (14) follows by applying Lemma 2.1 of (Weissman et al., 2003), Eq. (12) and Eq. (15) follows by applying both the Bernstein inequality (see, e.g., Cesa-Bianchi & Lugosi, 2006) and the Empirical Bernstein inequality (Maurer & Pontil, 2009), and Eq. (13) and Eq. (16) derive from the application of the Bernstein inequality for Bernoulli random variables.

The remaining events concern the summation of Martingale difference sequences. For ease of reading, let us introduce the following shorthand notation:

$$\mathbb{I}_s := \mathbb{I}\{s_{i,h} = s\},$$

$$\mathbb{I}_{s,a} := \mathbb{I}\{s_{i,h} = s, a_{i,h}^{\pi_i(\mathcal{B}_{i,h})} = a\},$$

where $\mathbb{I}$ represents the indicator function.

$$\mathcal{E}_{\mathrm{az}}(\mathcal{F}_{\widetilde{\Delta}^V,k,h}, H, L) := \Bigg\{ \sum_{i=1}^{k} \sum_{j=h}^{H-1} (P\widetilde{\Delta}_{i,j+1}^V)(s_{i,j}, a_{i,j}^{\pi_i(\mathcal{B}_{i,j})}) - \sum_{i=1}^{k} \sum_{j=h}^{H-1} \widetilde{\Delta}_{i,j+1}^V(s_{i,j+1})$$

$$\leqslant 2\sqrt{k(H-h)H^2 L} \Bigg\},$$

$$\mathcal{E}_{\mathrm{az}}(\mathcal{F}_{\widetilde{\Delta}^V,k,h,s}, H, L) := \Bigg\{ \sum_{i=1}^{k} \mathbb{I}_s \sum_{j=h}^{H-1} (P\widetilde{\Delta}_{i,j+1}^V)(s_{i,j}, a_{i,j}^{\pi_i(\mathcal{B}_{i,j})}) - \sum_{i=1}^{k} \mathbb{I}_s \sum_{j=h}^{H-1} \widetilde{\Delta}_{i,j+1}^V(s_{i,j+1})$$

$$\leqslant 2\sqrt{N_{k,h}(s)(H-h)H^2 L} \Bigg\},$$

$$\mathcal{E}_{\mathrm{az}}(\mathcal{F}_{\widetilde{\Delta}^V,k,h,s,a}, H, L) := \Bigg\{ \sum_{i=1}^{k} \mathbb{I}_{s,a} \sum_{j=h}^{H-1} (P\widetilde{\Delta}_{i,j+1}^V)(s_{i,j}, a_{i,j}^{\pi_i(\mathcal{B}_{i,j})}) - \sum_{i=1}^{k} \mathbb{I}_{s,a} \sum_{j=h}^{H-1} \widetilde{\Delta}_{i,j+1}^V(s_{i,j+1})$$

$$\leqslant 2\sqrt{N_{k,h}(s,a)(H-h)H^2 L} \Bigg\},$$

$$\mathcal{E}_{\mathrm{az}}\left(\mathcal{F}_{\widetilde{\Delta}^V,k,h}', \frac{1}{\sqrt{L}}, L\right) := \Bigg\{ \sum_{i=1}^{k} \sum_{j=h}^{H-1} \sum_{s'\in\mathcal{S}} P(s'|s_{i,j}, a_{i,j}^{\pi_i(\mathcal{B}_{i,j})}) \sqrt{\frac{\mathbb{I}\{s'\in[s]_{i,j}\}}{N_i(s_{i,j}, a_{i,j}^{\pi_i(\mathcal{B}_{i,j})})P(s'|s_{i,j}, a_{i,j}^{\pi_i(\mathcal{B}_{i,j})})}} \widetilde{\Delta}_{i,j+1}^V(s')$$

$$- \sum_{i=1}^{k} \sum_{j=h}^{H-1} \sqrt{\frac{\mathbb{I}\{s_{i,j+1}\in[s]_{i,j}\}}{N_i(s_{i,j}, a_{i,j}^{\pi_i(\mathcal{B}_{i,j})})P(s_{i,j+1}|s_{i,j}, a_{i,j}^{\pi_i(\mathcal{B}_{i,j})})}} \widetilde{\Delta}_{i,j+1}^V(s_{i,j+1})$$

$$\leqslant 2\sqrt{k(H-h)} \Bigg\},$$

$$\mathcal{E}_{\mathrm{az}}\left(\mathcal{F}_{\widetilde{\Delta}^V,k,h,s}', \frac{1}{\sqrt{L}}, L\right) := \Bigg\{ \sum_{i=1}^{k} \mathbb{I}_s \sum_{j=h}^{H-1} \sum_{s'\in\mathcal{S}} P(s'|s_{i,j}, a_{i,j}^{\pi_i(\mathcal{B}_{i,j})}) \sqrt{\frac{\mathbb{I}\{s'\in[s]_{i,j}\}}{N_i(s_{i,j}, a_{i,j}^{\pi_i(\mathcal{B}_{i,j})})P(s'|s_{i,j}, a_{i,j}^{\pi_i(\mathcal{B}_{i,j})})}} \widetilde{\Delta}_{i,j+1}^V(s')$$

$$- \sum_{i=1}^{k} \mathbb{I}_s \sum_{j=h}^{H-1} \sqrt{\frac{\mathbb{I}\{s_{i,j+1}\in[s]_{i,j}\}}{N_i(s_{i,j}, a_{i,j}^{\pi_i(\mathcal{B}_{i,j})})P(s_{i,j+1}|s_{i,j}, a_{i,j}^{\pi_i(\mathcal{B}_{i,j})})}} \widetilde{\Delta}_{i,j+1}^V(s_{i,j+1})$$

$$\leqslant 2\sqrt{N_{k,h}(s)(H-h)} \Bigg\},$$

$$\mathcal{E}_{\mathrm{az}}\left(\mathcal{F}_{\widetilde{\Delta}^V,k,h,s,a}', \frac{1}{\sqrt{L}}, L\right) := \Bigg\{ \sum_{i=1}^{k} \mathbb{I}_{s,a} \sum_{j=h}^{H-1} \sum_{s'\in\mathcal{S}} P(s'|s_{i,j}, a_{i,j}^{\pi_i(\mathcal{B}_{i,j})}) \sqrt{\frac{\mathbb{I}\{s'\in[s]_{i,j}\}}{N_i(s_{i,j}, a_{i,j}^{\pi_i(\mathcal{B}_{i,j})})P(s'|s_{i,j}, a_{i,j}^{\pi_i(\mathcal{B}_{i,j})})}} \widetilde{\Delta}_{i,j+1}^V(s')$$

$$- \sum_{i=1}^{k} \mathbb{I}_{s,a} \sum_{j=h}^{H-1} \sqrt{\frac{\mathbb{I}\{s_{i,j+1}\in[s]_{i,j}\}}{N_i(s_{i,j}, a_{i,j}^{\pi_i(\mathcal{B}_{i,j})})P(s_{i,j+1}|s_{i,j}, a_{i,j}^{\pi_i(\mathcal{B}_{i,j})})}} \widetilde{\Delta}_{i,j+1}^V(s_{i,j+1})$$

$$\leqslant 2\sqrt{N_{k,h}(s,a)(H-h)} \Bigg\},$$

$$\mathcal{E}_{\mathrm{az}}(\mathcal{F}_{\widetilde{\Delta}^V,k,h,\mathcal{B}}, H, L) := \Bigg\{ \sum_{i=1}^{k} \sum_{j=h}^{H-1} \sum_{\mathcal{B}\in\mathcal{P}(\mathcal{A})} C^{\mathrm{ind}}(\mathcal{B}|s_{i,j}) \sum_{s'\in\mathcal{S}} P(s'|s_{i,j}, a_{i,j}^{\pi_i(\mathcal{B})}) \widetilde{\Delta}_{i,j+1}^V(s')$$

$$- \sum_{i=1}^{k} \sum_{j=h}^{H-1} \widetilde{\Delta}_{i,j+1}^V(s_{i,j+1})$$

$$\leqslant 2\sqrt{k(H-h)H^2 L} \Bigg\},$$

$$\mathcal{E}_{\text{az}}(\mathcal{F}_{\widetilde{\Delta}^V,k,h,\mathcal{B},s}, H, L) := \left\{ \sum_{i=1}^{k} \mathbb{I}_s \sum_{j=h}^{H-1} \sum_{\mathcal{B}\in\mathcal{P}(\mathcal{A})} C^{\text{ind}}(\mathcal{B}|s_{i,j}) \sum_{s'\in\mathcal{S}} P(s'|s_{i,j}, a_{i,j}^{\pi_i(\mathcal{B})}) \widetilde{\Delta}_{i,j+1}^V(s') \right.$$

$$- \sum_{i=1}^{k} \mathbb{I}_s \sum_{j=h}^{H-1} \widetilde{\Delta}_{i,j+1}^V(s_{i,j+1})$$

$$\left. \leqslant 2\sqrt{N_{k,h}(s)(H-h)H^2 L} \right\},$$

$$\mathcal{E}_{\text{az}}(\mathcal{F}_{\widetilde{\Delta}^V,k,h,\mathcal{B},s,a}, H, L) := \left\{ \sum_{i=1}^{k} \mathbb{I}_{s,a} \sum_{j=h}^{H-1} \sum_{\mathcal{B}\in\mathcal{P}(\mathcal{A})} C^{\text{ind}}(\mathcal{B}|s_{i,j}) \sum_{s'\in\mathcal{S}} P(s'|s_{i,j}, a_{i,j}^{\pi_i(\mathcal{B})}) \widetilde{\Delta}_{i,j+1}^V(s') \right.$$

$$- \sum_{i=1}^{k} \mathbb{I}_{s,a} \sum_{j=h}^{H-1} \widetilde{\Delta}_{i,j+1}^V(s_{i,j+1})$$

$$\left. \leqslant 2\sqrt{N_{k,h}(s,a)(H-h)H^2 L} \right\},$$

$$\mathcal{E}_{\text{az}}\left(\mathcal{F}'_{\widetilde{\Delta}^V,k,h,\mathcal{B}}, \frac{1}{\sqrt{L}}, L\right) := \left\{ \sum_{i=1}^{k} \sum_{j=h}^{H-1} \mathbb{E}_{\substack{\mathcal{B}\sim C^{\text{ind}}(\cdot|s_{i,j}) \\ s'\sim P(\cdot|s_{i,j}, a_{i,j}^{\pi_i(\mathcal{B})})}} \left[ \sqrt{\frac{\mathbb{I}\{s'\in[s]_{i,j}\}}{N_i(s_{i,j}, a_{i,j}^{\pi_i(\mathcal{B})})P(s'|s_{i,j}, a_{i,j}^{\pi_i(\mathcal{B})})}} \widetilde{\Delta}_{i,j+1}^V(s') \right] \right.$$

$$- \sum_{i=1}^{k} \sum_{j=h}^{H-1} \sqrt{\frac{\mathbb{I}\{s_{i,j+1}\in[s]_{i,j}\}}{N_i(s_{i,j}, a_{i,j}^{\pi_i(\mathcal{B})})P(s'|s_{i,j}, a_{i,j}^{\pi_i(\mathcal{B})})}} \widetilde{\Delta}_{i,j+1}^V(s_{i,j+1})$$

$$\left. \leqslant 2\sqrt{k(H-h)} \right\},$$

$$\mathcal{E}_{\text{az}}\left(\mathcal{F}'_{\widetilde{\Delta}^V,k,h,\mathcal{B},s}, \frac{1}{\sqrt{L}}, L\right) := \left\{ \sum_{i=1}^{k} \mathbb{I}_s \sum_{j=h}^{H-1} \mathbb{E}_{\substack{\mathcal{B}\sim C^{\text{ind}}(\cdot|s_{i,j}) \\ s'\sim P(\cdot|s_{i,j}, a_{i,j}^{\pi_i(\mathcal{B})})}} \left[ \sqrt{\frac{\mathbb{I}\{s'\in[s]_{i,j}\}}{N_i(s_{i,j}, a_{i,j}^{\pi_i(\mathcal{B})})P(s'|s_{i,j}, a_{i,j}^{\pi_i(\mathcal{B})})}} \widetilde{\Delta}_{i,j+1}^V(s') \right] \right.$$

$$- \sum_{i=1}^{k} \mathbb{I}_s \sum_{j=h}^{H-1} \sqrt{\frac{\mathbb{I}\{s_{i,j+1}\in[s]_{i,j}\}}{N_i(s_{i,j}, a_{i,j}^{\pi_i(\mathcal{B})})P(s'|s_{i,j}, a_{i,j}^{\pi_i(\mathcal{B})})}} \widetilde{\Delta}_{i,j+1}^V(s_{i,j+1})$$

$$\left. \leqslant 2\sqrt{N_{k,h}(s)(H-h)} \right\},$$

$$\mathcal{E}_{\text{az}}\left(\mathcal{F}'_{\widetilde{\Delta}^V,k,h,\mathcal{B},s,a}, \frac{1}{\sqrt{L}}, L\right) := \left\{ \sum_{i=1}^{k} \mathbb{I}_{s,a} \sum_{j=h}^{H-1} \mathbb{E}_{\substack{\mathcal{B}\sim C^{\text{ind}}(\cdot|s_{i,j}) \\ s'\sim P(\cdot|s_{i,j}, a_{i,j}^{\pi_i(\mathcal{B})})}} \left[ \sqrt{\frac{\mathbb{I}\{s'\in[s]_{i,j}\}}{N_i(s_{i,j}, a_{i,j}^{\pi_i(\mathcal{B})})P(s'|s_{i,j}, a_{i,j}^{\pi_i(\mathcal{B})})}} \widetilde{\Delta}_{i,j+1}^V(s') \right] \right.$$

$$- \sum_{i=1}^{k} \mathbb{I}_{s,a} \sum_{j=h}^{H-1} \sqrt{\frac{\mathbb{I}\{s_{i,j+1}\in[s]_{i,j}\}}{N_i(s_{i,j}, a_{i,j}^{\pi_i(\mathcal{B})})P(s'|s_{i,j}, a_{i,j}^{\pi_i(\mathcal{B})})}} \widetilde{\Delta}_{i,j+1}^V(s_{i,j+1})$$

$$\left. \leqslant 2\sqrt{N_{k,h}(s,a)(H-h)} \right\},$$

$$\mathcal{E}_{\text{az}}(\mathcal{F}_{\widetilde{\Delta}^Q,k,h}, H, L) := \left\{ \sum_{i=1}^{k} \sum_{j=h}^{H-1} (C^{\text{ind}}\widetilde{\Delta}_{i,j}^Q)(s_{i,j}) - \sum_{i=1}^{k} \sum_{j=h}^{H-1} \widetilde{\Delta}_{i,j}^Q(s_{i,j}, a_{i,j}^{\pi_i(\mathcal{B}_{i,j})}) \right.$$

$$\left. \leqslant 2\sqrt{k(H-h)H^2 L} \right\},$$

$$\mathcal{E}_{\text{az}}(\mathcal{F}_{\widetilde{\Delta}^Q,k,h,s}, H, L) := \left\{ \sum_{i=1}^{k} \mathbb{I}_s \sum_{j=h}^{H-1} (C^{\text{ind}}\widetilde{\Delta}_{i,j}^Q)(s_{i,j}) - \sum_{i=1}^{k} \mathbb{I}_s \sum_{j=h}^{H-1} \widetilde{\Delta}_{i,j}^Q(s_{i,j}, a_{i,j}^{\pi_i(\mathcal{B}_{i,j})}) \right.$$

$$\leqslant 2\sqrt{N_{k,h}(s)(H-h)H^2L}\bigg\},$$

$$\mathcal{E}_{\mathrm{az}}(\mathcal{F}_{\widetilde{\Delta}^Q,k,h,s,a}, H, L) := \Bigg\{ \sum_{i=1}^{k} \mathbb{I}_{s,a} \sum_{j=h}^{H-1} (C^{\mathsf{ind}}\widetilde{\Delta}_{i,j}^Q)(s_{i,j}) - \sum_{i=1}^{k} \mathbb{I}_{s,a} \sum_{j=h}^{H-1} \widetilde{\Delta}_{i,j}^Q(s_{i,j}, a_{i,j}^{\pi_i(\mathcal{B}_{i,j})})$$

$$\leqslant 2\sqrt{N_{k,h}(s,a)(H-h)H^2L}\Bigg\},$$

$$\mathcal{E}_{\mathrm{az}}(\mathcal{F}'_{\widetilde{\Delta}^Q,k,h}, \frac{1}{\sqrt{L}}, L) := \Bigg\{ \sum_{i=1}^{k} \sum_{j=h}^{H-1} \sum_{\mathcal{B}\in\mathcal{P}(\mathcal{A})} C^{\mathsf{ind}}(\mathcal{B}|s_{i,j})\sqrt{\frac{\mathbb{I}\{\mathcal{B}\in[\mathcal{B}]_{i,j}\}}{N_i(s_{i,j})C^{\mathsf{ind}}(\mathcal{B}|s_{i,j})}}\widetilde{\Delta}_{i,j}^Q(s_{i,j}, a_{i,j}^{\pi_i(\mathcal{B})})$$

$$- \sum_{i=1}^{k} \sum_{j=h}^{H-1} \sqrt{\frac{\mathbb{I}\{\mathcal{B}_{i,j}\in[\mathcal{B}]_{i,j}\}}{N_i(s_{i,j})C^{\mathsf{ind}}(\mathcal{B}|s_{i,j})}}\widetilde{\Delta}_{i,j}^Q(s_{i,j}, a_{i,j}^{\pi_i(\mathcal{B}_{i,j})})$$

$$\leqslant 2\sqrt{k(H-h)}\Bigg\},$$

$$\mathcal{E}_{\mathrm{az}}(\mathcal{F}'_{\widetilde{\Delta}^Q,k,h,s}, \frac{1}{\sqrt{L}}, L) := \Bigg\{ \sum_{i=1}^{k} \mathbb{I}_s \sum_{j=h}^{H-1} \sum_{\mathcal{B}\in\mathcal{P}(\mathcal{A})} C^{\mathsf{ind}}(\mathcal{B}|s_{i,j})\sqrt{\frac{\mathbb{I}\{\mathcal{B}\in[\mathcal{B}]_{i,j}\}}{N_i(s_{i,j})C^{\mathsf{ind}}(\mathcal{B}|s_{i,j})}}\widetilde{\Delta}_{i,j}^Q(s_{i,j}, a_{i,j}^{\pi_i(\mathcal{B})})$$

$$- \sum_{i=1}^{k} \mathbb{I}_s \sum_{j=h}^{H-1} \sqrt{\frac{\mathbb{I}\{\mathcal{B}_{i,j}\in[\mathcal{B}]_{i,j}\}}{N_i(s_{i,j})C^{\mathsf{ind}}(\mathcal{B}|s_{i,j})}}\widetilde{\Delta}_{i,j}^Q(s_{i,j}, a_{i,j}^{\pi_i(\mathcal{B}_{i,j})})$$

$$\leqslant 2\sqrt{N_{k,h}(s)(H-h)}\Bigg\},$$

$$\mathcal{E}_{\mathrm{az}}(\mathcal{F}'_{\widetilde{\Delta}^Q,k,h,s,a}, \frac{1}{\sqrt{L}}, L) := \Bigg\{ \sum_{i=1}^{k} \mathbb{I}_{s,a} \sum_{j=h}^{H-1} \sum_{\mathcal{B}\in\mathcal{P}(\mathcal{A})} C^{\mathsf{ind}}(\mathcal{B}|s_{i,j})\sqrt{\frac{\mathbb{I}\{\mathcal{B}\in[\mathcal{B}]_{i,j}\}}{N_i(s_{i,j})C^{\mathsf{ind}}(\mathcal{B}|s_{i,j})}}\widetilde{\Delta}_{i,j}^Q(s_{i,j}, a_{i,j}^{\pi_i(\mathcal{B})})$$

$$- \sum_{i=1}^{k} \mathbb{I}_{s,a} \sum_{j=h}^{H-1} \sqrt{\frac{\mathbb{I}\{\mathcal{B}_{i,j}\in[\mathcal{B}]_{i,j}\}}{N_i(s_{i,j})C^{\mathsf{ind}}(\mathcal{B}|s_{i,j})}}\widetilde{\Delta}_{i,j}^Q(s_{i,j}, a_{i,j}^{\pi_i(\mathcal{B}_{i,j})})$$

$$\leqslant 2\sqrt{N_{k,h}(s,a)(H-h)}\Bigg\},$$

$$\mathcal{E}_{\mathrm{fr}}(\mathcal{G}_{\mathbb{V},k,h}, H^4T_k, H^3, L) := \Bigg\{ \sum_{i=1}^{k} \mathbb{E}\left[\sum_{j=h}^{H-1} \mathbb{V}_{j+1}^{\pi_i}(s_{i,j}, a_{i,j}^{\pi_i(\mathcal{B}_{i,j})})|\mathcal{H}_{i,h}\right] - \sum_{i=1}^{k} \sum_{j=h}^{H-1} \mathbb{V}_{j+1}^{\pi_i}(s_{i,j}, a_{i,j}^{\pi_i(\mathcal{B}_{i,j})})$$

$$\leqslant 2\sqrt{H^4T_kL} + \frac{4}{3}H^3L\Bigg\},$$

$$\mathcal{E}_{\mathrm{fr}}(\mathcal{G}_{\mathbb{V},k,h,s}, H^5N_{k,h}, H^3, L) := \Bigg\{ \sum_{i=1}^{k} \mathbb{I}_s\mathbb{E}\left[\sum_{j=h}^{H-1} \mathbb{V}_{j+1}^{\pi_i}(s_{i,j}, a_{i,j}^{\pi_i(\mathcal{B}_{i,j})})|\mathcal{H}_{i,h}\right] - \sum_{i=1}^{k} \mathbb{I}_s \sum_{j=h}^{H-1} \mathbb{V}_{j+1}^{\pi_i}(s_{i,j}, a_{i,j}^{\pi_i(\mathcal{B}_{i,j})})$$

$$\leqslant 2\sqrt{H^5N_{k,h}(s)L} + \frac{4}{3}H^3L\Bigg\},$$

$$\mathcal{E}_{\mathrm{fr}}(\mathcal{G}_{\mathbb{V},k,h,s,a}, H^5N_{k,h}, H^3, L) := \Bigg\{ \sum_{i=1}^{k} \mathbb{I}_{s,a}\mathbb{E}\left[\sum_{j=h}^{H-1} \mathbb{V}_{j+1}^{\pi_i}(s_{i,j}, a_{i,j}^{\pi_i(\mathcal{B}_{i,j})})|\mathcal{H}_{i,h}\right] - \sum_{i=1}^{k} \mathbb{I}_{s,a} \sum_{j=h}^{H-1} \mathbb{V}_{j+1}^{\pi_i}(s_{i,j}, a_{i,j}^{\pi_i(\mathcal{B}_{i,j})})$$

$$\leqslant 2\sqrt{H^5N_{k,h}(s,a)L} + \frac{4}{3}H^3L\Bigg\},$$

$$\mathcal{E}_{\mathrm{fr}}(\mathcal{G}_{\mathbb{Q},k,h}, H^4 T_k, H^3, L) := \left\{ \sum_{i=1}^{k} \mathbb{E}\left[ \sum_{j=h}^{H-1} \mathbb{Q}_j^{\pi_i}(s_{i,j}) | \mathcal{H}_{i,h} \right] - \sum_{i=1}^{k} \sum_{j=h}^{H-1} \mathbb{Q}_j^{\pi_i}(s_{i,j}) \right.$$

$$\left. \leqslant 2\sqrt{H^4 T_k L} + \frac{4}{3} H^3 L \right\},$$

$$\mathcal{E}_{\mathrm{fr}}(\mathcal{G}_{\mathbb{Q},k,h,s}, H^5 N_{k,h}, H^3, L) := \left\{ \sum_{i=1}^{k} \mathbb{I}_s \mathbb{E}\left[ \sum_{j=h}^{H-1} \mathbb{Q}_j^{\pi_i}(s_{i,j}) | \mathcal{H}_{i,h} \right] - \sum_{i=1}^{k} \mathbb{I}_s \sum_{j=h}^{H-1} \mathbb{Q}_j^{\pi_i}(s_{i,j}) \right.$$

$$\left. \leqslant 2\sqrt{H^5 N_{k,h}(s) L} + \frac{4}{3} H^3 L \right\},$$

$$\mathcal{E}_{\mathrm{fr}}(\mathcal{G}_{\mathbb{Q},k,h,s,a}, H^5 N_{k,h}, H^3, L) := \left\{ \sum_{i=1}^{k} \mathbb{I}_{s,a} \mathbb{E}\left[ \sum_{j=h}^{H-1} \mathbb{Q}_j^{\pi_i}(s_{i,j}) | \mathcal{H}_{i,h} \right] - \sum_{i=1}^{k} \mathbb{I}_{s,a} \sum_{j=h}^{H-1} \mathbb{Q}_j^{\pi_i}(s_{i,j}) \right.$$

$$\left. \leqslant 2\sqrt{H^5 N_{k,h}(s,a) L} + \frac{4}{3} H^3 L \right\},$$

$$\mathcal{E}_{\mathrm{az}}\left( \mathcal{F}_{\overline{b}^Q,k,h}, H^2, L \right) := \left\{ \sum_{i=1}^{k} \sum_{j=h}^{H-1} (P\overline{b}^Q)(s_{i,j}, a_{i,j}^{\pi_i(\mathcal{B}_{i,j})}) - \sum_{i=1}^{k} \sum_{j=h}^{H-1} \overline{b}^Q(s_{i,j+1}) \right.$$

$$\left. \leqslant 2\sqrt{k(H-h)H^4 L} \right\},$$

$$\mathcal{E}_{\mathrm{az}}\left( \mathcal{F}_{\overline{b}^Q,k,h,s}, H^2, L \right) := \left\{ \sum_{i=1}^{k} \mathbb{I}_s \sum_{j=h}^{H-1} (P\overline{b}^Q)(s_{i,j}, a_{i,j}^{\pi_i(\mathcal{B}_{i,j})}) - \sum_{i=1}^{k} \mathbb{I}_s \sum_{j=h}^{H-1} \overline{b}^Q(s_{i,j+1}) \right.$$

$$\left. \leqslant 2\sqrt{N_{k,h}(s)(H-h)H^4 L} \right\},$$

$$\mathcal{E}_{\mathrm{az}}\left( \mathcal{F}_{\overline{b}^Q,k,h,s,a}, H^2, L \right) := \left\{ \sum_{i=1}^{k} \mathbb{I}_{s,a} \sum_{j=h}^{H-1} (P\overline{b}^Q)(s_{i,j}, a_{i,j}^{\pi_i(\mathcal{B}_{i,j})}) - \sum_{i=1}^{k} \mathbb{I}_{s,a} \sum_{j=h}^{H-1} \overline{b}^Q(s_{i,j+1}) \right.$$

$$\left. \leqslant 2\sqrt{N_{k,h}(s,a)(H-h)H^4 L} \right\},$$

$$\mathcal{E}_{\mathrm{az}}\left( \mathcal{F}_{\overline{b}^V,k,h}, H^2, L \right) := \left\{ \sum_{i=1}^{k} \sum_{j=h}^{H-1} (C\overline{b}^V)(s_{i,j}) - \sum_{i=1}^{k} \sum_{j=h}^{H-1} \overline{b}^V(s_{i,j}, a_{i,j}^{\pi_i(\mathcal{B}_{i,j})}) \right.$$

$$\left. \leqslant 2\sqrt{k(H-h)H^4 L} \right\},$$

$$\mathcal{E}_{\mathrm{az}}\left( \mathcal{F}_{\overline{b}^V,k,h,s}, H^2, L \right) := \left\{ \sum_{i=1}^{k} \mathbb{I}_s \sum_{j=h}^{H-1} (C\overline{b}^V)(s_{i,j}) - \sum_{i=1}^{k} \mathbb{I}_s \sum_{j=h}^{H-1} \overline{b}^V(s_{i,j}, a_{i,j}^{\pi_i(\mathcal{B}_{i,j})}) \right.$$

$$\left. \leqslant 2\sqrt{N_{k,h}(s)(H-h)H^4 L} \right\},$$

$$\mathcal{E}_{\mathrm{az}}\left( \mathcal{F}_{\overline{b}^V,k,h,s,a}, H^2, L \right) := \left\{ \sum_{i=1}^{k} \mathbb{I}_{s,a} \sum_{j=h}^{H-1} (C\overline{b}^V)(s_{i,j}) - \sum_{i=1}^{k} \mathbb{I}_{s,a} \sum_{j=h}^{H-1} \overline{b}^V(s_{i,j}, a_{i,j}^{\pi_i(\mathcal{B}_{i,j})}) \right.$$

$$\left. \leqslant 2\sqrt{N_{k,h}(s,a)(H-h)H^4 L} \right\}.$$

The choice of $L$ derives from a union bound over the 40 events, for every episode $k \in [\![K]\!]$, stage $h \in [\![H]\!]$, state $s \in \mathcal{S}$, action $a \in \mathcal{A}$, and action subset $\mathcal{B} \in \mathcal{P}(\mathcal{A})$, noting that each event holds with at least probability $1 - \delta$.

## D.3. Technical Lemmas

**Lemma D.1** (Regret decomposition upper bound $\Delta_{k,h}^V$). *Let $k \in [\![K]\!]$ and $h \in [\![H]\!]$. Assume events $\mathcal{E}$ and $\Omega_{k,h}$ hold. Then, the regret from stage $h$ onward of all episodes up to $k$, in terms of state value function, can be upper bounded as follows:*

$$
\sum_{i=1}^k \Delta_{i,h}^V(s_{i,h}) \leqslant \sum_{i=1}^k \widetilde{\Delta}_{i,h}^V(s_{i,h})
$$

$$
\leqslant e^2 \sum_{i=1}^k \sum_{j=h}^{H-1} \Bigg[ b_{i,j}^V(s_{i,j}) + \mathbb{E}_{\mathcal{B} \sim C^{\mathrm{ind}}(\cdot|s_{i,j})}[b_{i,j}^Q(s_{i,j}, a_{i,j}^{\pi_i(\mathcal{B})})] + \frac{1}{H} b_{i,j}^Q(s_{i,j}, a_{i,j}^{\pi_i(\mathcal{B}_{i,j})})
$$

$$
+ \xi_{i,j}^V + \frac{1}{H}\varepsilon_{i,j}^V + \sqrt{2L}\bar{\xi}_{i,j}^V + \frac{\sqrt{2L}}{H}\bar{\varepsilon}_{i,j}^V + \sqrt{2L}\bar{\varepsilon}_{i,j}^Q
$$

$$
+ \mathbb{E}_{\mathcal{B} \sim C^{\mathrm{ind}}(\cdot|s_{i,j})}\left[\left((\widehat{P}_i - P)V_{j+1}^*\right)(s_{i,j}, a_{i,j}^{\pi_i(\mathcal{B})})\right] + \frac{1}{H}\left((\widehat{P}_i - P)V_{j+1}^*\right)(s_{i,j}, a_{i,j}^{\pi_i(\mathcal{B}_{i,j})})
$$

$$
+ \mathbb{E}_{\mathcal{B} \sim C^{\mathrm{ind}}(\cdot|s_{i,j})}\left[\frac{8H^2SL}{3N_i(s_{i,j}, a_{i,j}^{\pi_i(B)})}\right] + \frac{8HSL}{3N_i(s_{i,j}, a_{i,j}^{\pi_i(\mathcal{B}_{i,j})})}
$$

$$
+ \frac{2H^2 2^A L}{N_i(s_{i,j})} + \frac{2H 2^A L}{3N_i(s_{i,j})} + \sqrt{\frac{2\mathbb{Q}_j^{\pi_i}(s_{i,j})L}{N_i(s_{i,j})}} + \frac{2HL}{3N_i(s_{i,j})}\Bigg].
$$

*Proof.* Considering a single value of $k \in [\![K]\!]$, we first observe that, under $\Omega_{k,h}$:

$$
\Delta_{k,h}^V(s_{k,h}) = V_h^*(s_{k,h}) - V_h^{\pi_k}(s_{k,h})
$$

$$
\leqslant \widehat{V}_{k,h}(s_{k,h}) - V_h^{\pi_k}(s_{k,h})
$$

$$
= \widetilde{\Delta}_{k,h}^V(s_{k,h}).
$$

As such, we can then bound the pseudo-regret $\widetilde{\Delta}_{k,h}^V(s_{k,h})$:

$$
\widetilde{\Delta}_{k,h}^V(s_{k,h}) = \widehat{V}_{k,h}(s_{k,h}) - V_h^{\pi_k}(s_{k,h})
$$

$$
= b_{k,h}^V(s_{k,h}) + \left(\widehat{C}_k^{\mathrm{ind}}\widehat{Q}_{k,h}\right)(s_{k,h}) - \left(C^{\mathrm{ind}}Q_h^{\pi_k}\right)(s_{k,h})
$$

$$
= b_{k,h}^V(s_{k,h}) + \underbrace{\left((\widehat{C}_k^{\mathrm{ind}} - C^{\mathrm{ind}})\widehat{Q}_{k,h}\right)(s_{k,h})}_{(a)} + \mathbb{E}_{\mathcal{B} \sim C^{\mathrm{ind}}(\cdot|s_{k,h})}[b_{k,h}^Q(s_{k,h}, a_{k,h}^{\pi_k(\mathcal{B})})]
$$

$$
+ \mathbb{E}_{\mathcal{B} \sim C^{\mathrm{ind}}(\cdot|s_{k,h})}\left[\left(\widehat{P}_k\widehat{V}_{k,h+1}\right)(s_{k,h}, a_{k,h}^{\pi_k(\mathcal{B})}) - \left(PV_{h+1}^{\pi_k}\right)(s_{k,h}, a_{k,h}^{\pi_k(\mathcal{B})})\right]
$$

$$
= (a) + b_{k,h}^V(s_{k,h}) + \mathbb{E}_{\mathcal{B} \sim C^{\mathrm{ind}}(\cdot|s_{k,h})}[b_{k,h}^Q(s_{k,h}, a_{k,h}^{\pi_k(\mathcal{B})})]
$$

$$
+ \mathbb{E}_{\mathcal{B} \sim C^{\mathrm{ind}}(\cdot|s_{k,h})}\left[\left((\widehat{P}_k - P)\widehat{V}_{k,h+1}\right)(s_{k,h}, a_{k,h}^{\pi_k(\mathcal{B})})\right] + \mathbb{E}_{\substack{\mathcal{B} \sim C^{\mathrm{ind}}(\cdot|s_{k,h}) \\ s' \sim P(\cdot|s_{k,h}, a_{k,h}^{\pi_k(\mathcal{B})})}}\left[\widetilde{\Delta}_{k,h+1}^V(s')\right]
$$

$$
= (a) + b_{k,h}^V(s_{k,h}) + \mathbb{E}_{\mathcal{B} \sim C^{\mathrm{ind}}(\cdot|s_{k,h})}[b_{k,h}^Q(s_{k,h}, a_{k,h}^{\pi_k(\mathcal{B})})] + \mathbb{E}_{\substack{\mathcal{B} \sim C^{\mathrm{ind}}(\cdot|s_{k,h}) \\ s' \sim P(\cdot|s_{k,h}, a_{k,h}^{\pi_k(\mathcal{B})})}}\left[\widetilde{\Delta}_{k,h+1}^V(s')\right]
$$

$$
+ \mathbb{E}_{\mathcal{B} \sim C^{\mathrm{ind}}(\cdot|s_{k,h})}\left[\left((\widehat{P}_k - P)V_{h+1}^*\right)(s_{k,h}, a_{k,h}^{\pi_k(\mathcal{B})})\right]
$$

$$+ \, \mathbb{E}_{\mathcal{B} \sim C^{\text{ind}}(\cdot|s_{k,h})} \left[ \left( (\widehat{P}_k - P)(\widehat{V}_{k,h+1} - V_{h+1}^*) \right)(s_{k,h}, a_{k,h}^{\pi_k(\mathcal{B})}) \right]$$

$$= (a) + b_{k,h}^V(s_{k,h}) + \mathbb{E}_{\mathcal{B} \sim C^{\text{ind}}(\cdot|s_{k,h})}[b_{k,h}^Q(s_{k,h}, a_{k,h}^{\pi_k(\mathcal{B})})] + \widetilde{\Delta}_{k,h+1}^V(s_{k,h+1})$$

$$+ \, \xi_{k,h}^V + \mathbb{E}_{\mathcal{B} \sim C^{\text{ind}}(\cdot|s_{k,h})} \left[ \left( (\widehat{P}_k - P)V_{h+1}^* \right)(s_{k,h}, a_{k,h}^{\pi_k(\mathcal{B})}) \right]$$

$$+ \, \underbrace{\mathbb{E}_{\mathcal{B} \sim C^{\text{ind}}(\cdot|s_{k,h})}[\left( (\widehat{P}_k - P)(\widehat{V}_{k,h+1} - V_{h+1}^*) \right)(s_{k,h}, a_{k,h}^{\pi_k(\mathcal{B})})]}_{(b)} \tag{17}$$

We now bound term $(b)$ following the procedure of (Azar et al., 2017, Lemma 3), obtaining the following bound:

$$(b) \leqslant \sqrt{2L}\bar{\xi}_{k,h}^V + \frac{1}{H}\widetilde{\Delta}_{k,h+1}^V(s_{k,h+1}) + \mathbb{E}_{\mathcal{B} \sim C^{\text{ind}}(\cdot|s_{k,h})} \left[ \frac{8H^2 SL}{3N_k(s_{k,h}, a_{k,h}^{\pi_k(B)})} \right].$$

To bound term $(a)$ we first derive that:

$$(a) = \left( (\widehat{C}_k^{\text{ind}} - C^{\text{ind}})\widehat{Q}_{k,h} \right)(s_{k,h})$$

$$= \underbrace{\left( (\widehat{C}_k^{\text{ind}} - C^{\text{ind}})(\widehat{Q}_{k,h} - Q_h^{\pi_k}) \right)(s_{k,h})}_{(c)} + \left( (\widehat{C}_k^{\text{ind}} - C^{\text{ind}})Q_h^{\pi_k} \right)(s_{k,h})$$

$$\leqslant (c) + \sqrt{\frac{2\mathbb{Q}_h^{\pi_k}(s_{k,h})L}{N_k(s_{k,h})}} + \frac{2HL}{3N_k(s_{k,h})} \tag{18}$$

where Eq. (18) is obtained by applying Bernstein's inequality. We now bound term $(c)$:

$$(c) = \sum_{\mathcal{B} \in \mathcal{P}(\mathcal{A})} \left( \widehat{C}_k^{\text{ind}}(\mathcal{B}|s_{k,h}) - C^{\text{ind}}(\mathcal{B}|s_{k,h}) \right) \widetilde{\Delta}_{k,h}^Q(s_{k,h}, a_{k,h}^{\pi_k(\mathcal{B})})$$

$$\leqslant \sum_{\mathcal{B} \in \mathcal{P}(\mathcal{A})} \left( \sqrt{\frac{2C^{\text{ind}}(\mathcal{B}|s_{k,h})(1 - C^{\text{ind}}(\mathcal{B}|s_{k,h}))L}{N_k(s_{k,h})}} + \frac{2L}{3N_k(s_{k,h})} \right) \widetilde{\Delta}_{k,h}^Q(s_{k,h}, a_{k,h}^{\pi_k(\mathcal{B})}) \tag{19}$$

$$\leqslant \sum_{\mathcal{B} \in \mathcal{P}(\mathcal{A})} \left( \sqrt{\frac{2C^{\text{ind}}(\mathcal{B}|s_{k,h})L}{N_k(s_{k,h})}} + \frac{2L}{3N_k(s_{k,h})} \right) \widetilde{\Delta}_{k,h}^Q(s_{k,h}, a_{k,h}^{\pi_k(\mathcal{B})})$$

$$\leqslant \sqrt{2L} \underbrace{\sum_{\mathcal{B} \in \mathcal{P}(\mathcal{A})} \sqrt{\frac{C^{\text{ind}}(\mathcal{B}|s_{k,h})}{N_k(s_{k,h})}} \widetilde{\Delta}_{k,h}^Q(s_{k,h}, a_{k,h}^{\pi_k(\mathcal{B})})}_{(d)} + \frac{2H2^A L}{3N_k(s_{k,h})} \tag{20}$$

where Eq. (19) is obtained by applying Bernstein's inequality for Bernoulli Random Variables, and Eq. (20) follows by upper bounding $\widetilde{\Delta}_{k,h}^Q(s_{k,h}, a_{k,h}^{\pi_k(\mathcal{B})})$ with $H$. Let $[\mathcal{B}]_{k,h} = \{\mathcal{B} \in \mathcal{P}(\mathcal{A}) : N_k(s_{k,h})C^{\text{ind}}(\mathcal{B}|s_{k,h}) \geqslant 2H^2 L\}$. To bound term $(d)$, we first rewrite it as:

$$(d) = \underbrace{\sum_{\mathcal{B} \in [\mathcal{B}]_{k,h}} \sqrt{\frac{C^{\text{ind}}(\mathcal{B}|s_{k,h})}{N_k(s_{k,h})}} \widetilde{\Delta}_{k,h}^Q(s_{k,h}, a_{k,h}^{\pi_k(\mathcal{B})})}_{(e)} + \underbrace{\sum_{\mathcal{B} \notin [\mathcal{B}]_{k,h}} \sqrt{\frac{C^{\text{ind}}(\mathcal{B}|s_{k,h})}{N_k(s_{k,h})}} \widetilde{\Delta}_{k,h}^Q(s_{k,h}, a_{k,h}^{\pi_k(\mathcal{B})})}_{(f)}, \tag{21}$$

we can then bound term $(f)$ as:

$$(f) = \sum_{\mathcal{B} \notin [\mathcal{B}]_{k,h}} \frac{\sqrt{N_k(s_{k,h})C^{\mathsf{ind}}(\mathcal{B}|s_{k,h})}}{N_k(s_{k,h})} \widetilde{\Delta}_{k,h}^Q(s_{k,h}, a_{k,h}^{\pi_k(\mathcal{B})})$$

$$\leqslant \frac{H^2 2^A \sqrt{2L}}{N_k(s_{k,h})},$$

by applying the condition of $[\mathcal{B}]_{k,h}$, and term $(e)$ as:

$$(e) = \sum_{\mathcal{B} \in [\mathcal{B}]_{k,h}} \sqrt{\frac{C^{\mathsf{ind}}(\mathcal{B}|s_{k,h})}{N_k(s_{k,h})}} \widetilde{\Delta}_{k,h}^Q(s_{k,h}, a_{k,h}^{\pi_k(\mathcal{B})})$$

$$= \bar{\varepsilon}_{k,h}^Q + \sqrt{\frac{\mathbb{I}\{\mathcal{B}_{k,h} \in [\mathcal{B}]_{k,h}\}}{N_k(s_{k,h})C^{\mathsf{ind}}(\mathcal{B}_{k,h}|s_{k,h})}} \widetilde{\Delta}_{k,h}^Q(s_{k,h}, a_{k,h}^{\pi_k(\mathcal{B}_{k,h})}) \tag{22}$$

$$\leqslant \bar{\varepsilon}_{k,h}^Q + \frac{1}{\sqrt{2H^2L}} \widetilde{\Delta}_{k,h}^Q(s_{k,h}, a_{k,h}^{\pi_k(\mathcal{B}_{k,h})}), \tag{23}$$

where Eq. (22) is obtained by applying the definition of $\bar{\varepsilon}_{k,h}^Q$, Eq. (23) follows from the definition of $[\mathcal{B}]_{k,h}$. Plugging the bounds of terms $(e)$ and $(f)$ into Eq. (21), we get:

$$(d) \leqslant \bar{\varepsilon}_{k,h}^Q + \frac{1}{\sqrt{2H^2L}} \widetilde{\Delta}_{k,h}^Q(s_{k,h}, a_{k,h}^{\pi_k(\mathcal{B}_{k,h})}) + \frac{H^2 2^A \sqrt{2L}}{N_k(s_{k,h})}.$$

We can then bound $\widetilde{\Delta}_{k,h}^Q(s_{k,h}, a_{k,h}^{\pi_k(\mathcal{B}_{k,h})})$ as in Lemma 3 of (Azar et al., 2017) and plug the bound of term $(d)$ into Eq. (20), thus obtaining:

$$(c) \leqslant \sqrt{2L}\bar{\varepsilon}_{k,h}^Q + \frac{1}{H}b_{k,h}^Q(s_{k,h}, a_{k,h}^{\pi_k(\mathcal{B}_{k,h})}) + \left(\frac{1}{H} + \frac{1}{H^2}\right) \widetilde{\Delta}_{k,h+1}^V(s_{k,h+1})$$

$$+ \frac{1}{H}\left((\widehat{P}_k - P)V_{h+1}^*\right)(s_{k,h}, a_{k,h}^{\pi_k(\mathcal{B}_{k,h})}) + \frac{1}{H}\varepsilon_{k,h}^V + \frac{\sqrt{2L}}{H}\bar{\varepsilon}_{k,h}^V$$

$$+ \frac{8HSL}{3N_k(s_{k,h}, a_{k,h}^{\pi_k(\mathcal{B}_{k,h})})} + \frac{2H^2 2^A L}{N_k(s_{k,h})} + \frac{2H 2^A L}{3N_k(s_{k,h})}.$$

Finally, putting together, the bounds of terms $(a)$, $(b)$, and $(c)$, and plugging them into Eq. (17), we obtain:

$$\widetilde{\Delta}_{k,h}^V(s_{k,h}) \leqslant b_{k,h}^V(s_{k,h}) + \mathbb{E}_{\mathcal{B}\sim C^{\mathsf{ind}}(\cdot|s_{k,h})}[b_{k,h}^Q(s_{k,h}, a_{k,h}^{\pi_k(\mathcal{B})})] + \frac{1}{H}b_{k,h}^Q(s_{k,h}, a_{k,h}^{\pi_k(\mathcal{B}_{k,h})})$$

$$+ \left(1 + \frac{2}{H} + \frac{1}{H^2}\right) \widetilde{\Delta}_{k,h+1}^V(s_{k,h+1}) + \xi_{k,h}^V + \frac{1}{H}\varepsilon_{k,h}^V$$

$$+ \sqrt{2L}\bar{\xi}_{k,h}^V + \frac{\sqrt{2L}}{H}\bar{\varepsilon}_{k,h}^V + \sqrt{2L}\bar{\varepsilon}_{k,h}^Q$$

$$+ \mathbb{E}_{\mathcal{B}\sim C^{\mathsf{ind}}(\cdot|s_{k,h})}\left[\left((\widehat{P}_k - P)V_{h+1}^*\right)(s_{k,h}, a_{k,h}^{\pi_k(\mathcal{B})})\right] + \frac{1}{H}\left((\widehat{P}_k - P)V_{h+1}^*\right)(s_{k,h}, a_{k,h}^{\pi_k(\mathcal{B}_{k,h})})$$

$$+ \mathbb{E}_{\mathcal{B}\sim C^{\mathsf{ind}}(\cdot|s_{k,h})}\left[\frac{8H^2 SL}{3N_k(s_{k,h}, a_{k,h}^{\pi_k(\mathcal{B})})}\right] + \frac{8HSL}{3N_k(s_{k,h}, a_{k,h}^{\pi_k(\mathcal{B}_{k,h})})}$$

$$+ \frac{2H^2 2^A L}{N_k(s_{k,h})} + \frac{2H 2^A L}{3N_k(s_{k,h})} + \sqrt{\frac{2\mathbb{Q}_h^{\pi_k}(s_{k,h})L}{N_k(s_{k,h})}} + \frac{2HL}{3N_k(s_{k,h})}.$$

By applying the same inductive argument of (Azar et al., 2017, Lemma 3), we can isolate the term $\widetilde{\Delta}_{k,h}^V$ and rewrite:

$$
\begin{aligned}
\widetilde{\Delta}_{k,h}^V(s_{k,h}) \leqslant e^2 \sum_{j=h}^{H-1} \Bigg[ & b_{k,j}^V(s_{k,j}) + \mathbb{E}_{\mathcal{B} \sim C^{\mathrm{ind}}(\cdot|s_{k,j})} [b_{k,j}^Q(s_{k,j}, a_{k,j}^{\pi_k(\mathcal{B})})] + \frac{1}{H} b_{k,j}^Q(s_{k,j}, a_{k,j}^{\pi_k(\mathcal{B}_{k,j})}) \\
& + \xi_{k,h}^V + \frac{1}{H} \varepsilon_{k,j}^V + \sqrt{2L}\bar{\xi}_{k,h}^V + \frac{\sqrt{2L}}{H}\bar{\varepsilon}_{k,j}^V + \sqrt{2L}\bar{\varepsilon}_{k,j}^Q \\
& + \mathbb{E}_{\mathcal{B} \sim C^{\mathrm{ind}}(\cdot|s_{k,j})} \left[ \left( (\widehat{P}_k - P)V_{j+1}^* \right)(s_{k,j}, a_{k,j}^{\pi_k(\mathcal{B})}) \right] + \frac{1}{H} \left( (\widehat{P}_k - P)V_{j+1}^* \right)(s_{k,j}, a_{k,j}^{\pi_k(\mathcal{B}_{k,j})}) \\
& + \mathbb{E}_{\mathcal{B} \sim C^{\mathrm{ind}}(\cdot|s_{k,j})} \left[ \frac{8H^2 SL}{3N_k(s_{k,j}, a_{k,j}^{\pi_k(\mathcal{B})})} \right] + \frac{8HSL}{3N_k(s_{k,j}, a_{k,j}^{\pi_k(\mathcal{B}_{k,j})})} \\
& + \frac{2H^2 2^A L}{N_k(s_{k,j})} + \frac{2H 2^A L}{3N_k(s_{k,j})} + \sqrt{\frac{2\mathbb{Q}_j^{\pi_k}(s_{k,j})L}{N_k(s_{k,j})}} + \frac{2HL}{3N_k(s_{k,j})} \Bigg],
\end{aligned}
\tag{24}
$$

observing that our multiplicative constant is $e^2$ instead of $e$ due to the $(1 + 2/H + 1/H^2)$ coefficient of the recursive term. Finally, recalling the definition of $\Omega_{k,h}$:

$$\Omega_{k,h} := \left\{ \widehat{V}_{i,j}(s) \geqslant V_j^*(s) \wedge \widehat{Q}_{i,j}(s,a) \geqslant Q_j^*(s,a), \forall (i,j) \in [k,h]_{\mathrm{hist}}, s \in \mathcal{S}, a \in \mathcal{A} \right\},$$

where $[k,h]_{\mathrm{hist}} := \{(i,j) : i \in [\![K]\!], j \in [\![H]\!], (i < k) \vee (i = k, j \geqslant h)\}$, we observe that if $\Omega_{k,h}$ holds then also the events $\Omega_{i,j}$ for $(i,j) \in [k,h]_{\mathrm{hist}}$ hold. As such, we can sum up the bound of Eq. (24) over all the episodes $i \in [\![k]\!]$, thus concluding the proof. □

**Lemma D.2** (Regret decomposition upper bound $\Delta_{k,h}^Q$). *Let $k \in [\![K]\!]$ and $h \in [\![H]\!]$. Assume events $\mathcal{E}$ and $\Omega_{k,h}$ hold. Then, the regret from stage $h$ onward of all episodes up to $k$, in terms of state-action value function, can be upper bounded as follows:*

$$
\begin{aligned}
\sum_{i=1}^{k} \Delta_{k,h}^Q(s_{i,h}, a_{i,h}^{\pi_i(\mathcal{B}_{i,h})}) &\leqslant \sum_{i=1}^{k} \widetilde{\Delta}_{k,h}^Q(s_{i,h}, a_{i,h}^{\pi_i(\mathcal{B}_{i,h})}) \\
&\leqslant e \sum_{i=1}^{k} \sum_{j=h}^{H-1} \Bigg[ b_{i,j}^Q(s_{i,j}, a_{i,j}^{\pi_i(\mathcal{B}_{i,j})}) + \left( 1 + \frac{1}{H} \right) b_{i,j+1}^V(s_{i,j+1}) \\
&\quad + \sqrt{2L}\bar{\varepsilon}_{i,j}^V + \varepsilon_{i,j}^V + \left( 1 + \frac{1}{H} \right) \varepsilon_{i,j+1}^Q \\
&\quad + \left( (\widehat{P}_i - P)V_{j+1}^* \right)(s_{i,j}, a_{i,j}^{\pi_i(\mathcal{B}_{i,j})}) \\
&\quad + \frac{8H^2 SL}{N_i(s_{i,j}, a_{i,j}^{\pi_i(\mathcal{B}_{i,j})})} + \left( 1 + \frac{1}{H} \right) 2H\sqrt{\frac{2^A L}{N_i(s_{i,j+1})}} \Bigg].
\end{aligned}
$$

*Proof.* Considering a single value of $k \in [\![K]\!]$, we first observe that, under $\Omega_{k,h}$:

$$
\begin{aligned}
\Delta_{k,h}^Q(s_{k,h}, a_{k,h}^{\pi_k(\mathcal{B}_{k,h})}) &= Q_h^*(s_{k,h}, a_{k,h}^{\pi_k(\mathcal{B}_{k,h})}) - Q_h^{\pi_k}(s_{k,h}, a_{k,h}^{\pi_k(\mathcal{B}_{k,h})}) \\
&\leqslant \widehat{Q}_{k,h}(s_{k,h}, a_{k,h}^{\pi_k(\mathcal{B}_{k,h})}) - Q_h^{\pi_k}(s_{k,h}, a_{k,h}^{\pi_k(\mathcal{B}_{k,h})})
\end{aligned}
$$

$$= \widetilde{\Delta}_{k,h}^{Q}(s_{k,h}, a_{k,h}^{\pi_k(\mathcal{B}_{k,h})}).$$

As such, we can bound the pseudo-regret $\widetilde{\Delta}_{k,h}^{Q}(s_{k,h}, a_{k,h}^{\pi_k(\mathcal{B}_{k,h})})$:

$$
\begin{aligned}
\widetilde{\Delta}_{k,h}^{Q}(s_{k,h}, a_{k,h}^{\pi_k(\mathcal{B}_{k,h})}) &= \widehat{Q}_{k,h}(s_{k,h}, a_{k,h}^{\pi_k(\mathcal{B}_{k,h})}) - Q_h^{\pi_k}(s_{k,h}, a_{k,h}^{\pi_k(\mathcal{B}_{k,h})}) \\
&= b_{k,h}^{Q}(s_{k,h}, a_{k,h}^{\pi_k(\mathcal{B}_{k,h})}) + \left(\widehat{P}_k \widehat{V}_{k,h+1}\right)(s_{k,h}, a_{k,h}^{\pi_k(\mathcal{B}_{k,h})}) - \left(P V_{h+1}^{\pi_k}\right)(s_{k,h}, a_{k,h}^{\pi_k(\mathcal{B}_{k,h})}) \\
&= b_{k,h}^{Q}(s_{k,h}, a_{k,h}^{\pi_k(\mathcal{B}_{k,h})}) + \left((\widehat{P}_k - P)\widehat{V}_{k,h+1}\right)(s_{k,h}, a_{k,h}^{\pi_k(\mathcal{B}_{k,h})}) \\
&\quad + \left(P(\widehat{V}_{k,h+1} - V_{h+1}^{\pi_k})\right)(s_{k,h}, a_{k,h}^{\pi_k(\mathcal{B}_{k,h})}) \\
&= b_{k,h}^{Q}(s_{k,h}, a_{k,h}^{\pi_k(\mathcal{B}_{k,h})}) + \left((\widehat{P}_k - P)(\widehat{V}_{k,h+1} - V_{h+1}^{*})\right)(s_{k,h}, a_{k,h}^{\pi_k(\mathcal{B}_{k,h})}) \\
&\quad + \left(P(\widehat{V}_{k,h+1} - V_{h+1}^{\pi_k})\right)(s_{k,h}, a_{k,h}^{\pi_k(\mathcal{B}_{k,h})}) + \left((\widehat{P}_k - P)V_{h+1}^{*}\right)(s_{k,h}, a_{k,h}^{\pi_k(\mathcal{B}_{k,h})}) \\
&\leqslant b_{k,h}^{Q}(s_{k,h}, a_{k,h}^{\pi_k(\mathcal{B}_{k,h})}) + \sqrt{2L}\bar{\varepsilon}_{k,h}^{V} + \varepsilon_{k,h}^{V} + \left(1 + \frac{1}{H}\right)\widetilde{\Delta}_{k,h+1}^{V}(s_{k,h+1}) + \\
&\quad + \frac{8H^2 SL}{N_k(s_{k,h}, a_{k,h}^{\pi_k(\mathcal{B}_{k,h})})} + \left((\widehat{P}_k - P)V_{h+1}^{*}\right)(s_{k,h}, a_{k,h}^{\pi_k(\mathcal{B}_{k,h})}),
\end{aligned}
\tag{25}
$$

where Eq. (25) is obtained by bounding $(\widehat{P}_k - P)(\widehat{V}_{k,h+1} - V_{h+1}^{*})$ according to the procedure of (Azar et al., 2017, Lemma 3). Observing that:

$$
\begin{aligned}
\widetilde{\Delta}_{k,h+1}^{V}(s_{k,h+1}) &= b_{k,h+1}^{V}(s_{k,h+1}) + \left(\widehat{C}_k^{\mathrm{ind}} \widehat{Q}_{k,h+1}\right)(s_{k,h+1}) - \left(C^{\mathrm{ind}} Q_{h+1}^{\pi_k}\right)(s_{k,h+1}) \\
&= b_{k,h+1}^{V}(s_{k,h+1}) + \left((\widehat{C}_k^{\mathrm{ind}} - C^{\mathrm{ind}})\widehat{Q}_{k,h+1}\right)(s_{k,h+1}) + \left(C^{\mathrm{ind}}(\widehat{Q}_{k,h+1} - Q_{h+1}^{\pi_k})\right)(s_{k,h+1}) \\
&\leqslant b_{k,h+1}^{V}(s_{k,h+1}) + \varepsilon_{k,h+1}^{Q} + 2H\sqrt{\frac{2^A L}{N_k(s_{k,h+1})}} + \widetilde{\Delta}_{k,h+1}^{Q}(s_{k,h+1}, a_{k,h+1}^{\pi_k(\mathcal{B}_{k,h+1})}),
\end{aligned}
\tag{26}
$$

where Eq. (26) is obtained by applying the definition of $\varepsilon_{k,h}^{Q}$ and by bounding $(\widehat{C}_k^{\mathrm{ind}} - C^{\mathrm{ind}})\widehat{Q}_{k,h+1}$ using Thr. 2.1 of (Weissman et al., 2003), then we can rewrite:

$$
\begin{aligned}
\widetilde{\Delta}_{k,h}^{Q}(s_{k,h}, a_{k,h}^{\pi_k(\mathcal{B}_{k,h})}) &\leqslant b_{k,h}^{Q}(s_{k,h}, a_{k,h}^{\pi_k(\mathcal{B}_{k,h})}) + \sqrt{2L}\bar{\varepsilon}_{k,h}^{V} + \varepsilon_{k,h}^{V} + \frac{8H^2 SL}{N_k(s_{k,h}, a_{k,h}^{\pi_k(\mathcal{B}_{k,h})})} \\
&\quad + \left((\widehat{P}_k - P)V_{h+1}^{*}\right)(s_{k,h}, a_{k,h}^{\pi_k(\mathcal{B}_{k,h})}) + \left(1 + \frac{1}{H}\right)b_{k,h+1}^{V}(s_{k,h+1}) + \left(1 + \frac{1}{H}\right)\varepsilon_{k,h+1}^{Q} \\
&\quad + \left(1 + \frac{1}{H}\right)2H\sqrt{\frac{2^A L}{N_k(s_{k,h+1})}} + \left(1 + \frac{1}{H}\right)\widetilde{\Delta}_{k,h+1}^{Q}(s_{k,h+1}, a_{k,h+1}^{\pi_k(\mathcal{B}_{k,h+1})}).
\end{aligned}
$$

By applying the same inductive argument of (Azar et al., 2017, Lemma 3), we can isolate the term $\widetilde{\Delta}_{k,h}^{Q}$ and rewrite:

$$
\begin{aligned}
\widetilde{\Delta}_{k,h}^{Q}(s_{k,h}, a_{k,h}^{\pi_k(\mathcal{B}_{k,h})}) &\leqslant e \sum_{j=h}^{H-1} \left[ b_{k,j}^{Q}(s_{k,j}, a_{k,j}^{\pi_k(\mathcal{B}_{k,j})}) + \left(1 + \frac{1}{H}\right)b_{k,j+1}^{V}(s_{k,j+1}) \right. \\
&\quad \left. + \sqrt{2L}\bar{\varepsilon}_{k,j}^{V} + \varepsilon_{k,j}^{V} + \left(1 + \frac{1}{H}\right)\varepsilon_{k,j+1}^{Q} \right.
\end{aligned}
$$

$$+ \left( (\widehat{P}_k - P)V^*_{j+1} \right) (s_{k,j}, a^{\pi_k(\mathcal{B}_{k,j})}_{k,j})$$

$$+ \frac{8H^2 SL}{N_k(s_{k,j}, a^{\pi_k(\mathcal{B}_{k,j})}_{k,j})} + \left( 1 + \frac{1}{H} \right) 2H \sqrt{\frac{2^A L}{N_k(s_{k,j+1})}} \Bigg]. \tag{27}$$

Finally, recalling the definition of $\Omega_{k,h}$:

$$\Omega_{k,h} := \left\{ \widehat{V}_{i,j}(s) \geqslant V^*_j(s) \wedge \widehat{Q}_{i,j}(s,a) \geqslant Q^*_j(s,a), \forall(i,j) \in [k,h]_{\mathrm{hist}}, s \in \mathcal{S}, a \in \mathcal{A} \right\},$$

where $[k,h]_{\mathrm{hist}} := \{(i,j) : i \in [\![K]\!], j \in [\![H]\!], (i < k) \vee (i = k, j \geqslant h)\}$, we observe that if $\Omega_{k,h}$ holds then also the events $\Omega_{i,j}$ for $(i,j) \in [k,h]_{\mathrm{hist}}$ hold. As such, we can sum up the bound of Eq. (27) over all the episodes $i \in [\![k]\!]$, thus concluding the proof. □

**Lemma D.3.** *Let $k \in [\![K]\!]$ and $h \in [\![H]\!]$. Let events $\mathcal{E}$ and $\Omega_{k,h}$ hold. Then the following bounds hold:*

$$\sum_{i=1}^{k} \sum_{j=h}^{H-1} \varepsilon^V_{i,j} \leqslant 2\sqrt{H^2 T_k L},$$

$$\sum_{i=1}^{k} \sum_{j=h}^{H-1} \xi^V_{i,j} \leqslant 2\sqrt{H^2 T_k L},$$

$$\sum_{i=1}^{k} \sum_{j=h}^{H-1} \varepsilon^Q_{i,j} \leqslant 2\sqrt{H^2 T_k L},$$

$$\sum_{i=1}^{k} \sum_{j=h}^{H-1} \overline{\varepsilon}^V_{i,j} \leqslant \sqrt{T_k},$$

$$\sum_{i=1}^{k} \sum_{j=h}^{H-1} \overline{\xi}^V_{i,j} \leqslant \sqrt{T_k},$$

$$\sum_{i=1}^{k} \sum_{j=h}^{H-1} \overline{\varepsilon}^Q_{i,j} \leqslant \sqrt{T_k}.$$

$$\tag{28}$$

*Moreover, for every $s \in \mathcal{S}$ and $a \in \mathcal{A}$, the following bounds also hold:*

$$\sum_{i=1}^{k} \mathbb{I}\{s_{i,h} = s\} \sum_{j=h}^{H-1} \varepsilon^V_{i,j} \leqslant 2\sqrt{H^3 N_{k,h}(s) L},$$

$$\sum_{i=1}^{k} \mathbb{I}\{s_{i,h} = s\} \sum_{j=h}^{H-1} \xi^V_{i,j} \leqslant 2\sqrt{H^3 N_{k,h}(s) L},$$

$$\sum_{i=1}^{k} \mathbb{I}\{s_{i,h} = s\} \sum_{j=h}^{H-1} \varepsilon^Q_{i,j} \leqslant 2\sqrt{H^3 N_{k,h}(s) L},$$

$$\sum_{i=1}^{k} \mathbb{I}\{s_{i,h} = s\} \sum_{j=h}^{H-1} \overline{\varepsilon}^V_{i,j} \leqslant \sqrt{H N_{k,h}(s)},$$

$$\sum_{i=1}^{k} \mathbb{I}\{s_{i,h} = s\} \sum_{j=h}^{H-1} \overline{\xi}^V_{i,j} \leqslant \sqrt{H N_{k,h}(s)},$$

$$\sum_{i=1}^{k} \mathbb{I}\{s_{i,h} = s\} \sum_{j=h}^{H-1} \overline{\varepsilon}_{i,j}^{Q} \leqslant \sqrt{HN_{k,h}(s)},$$

$$\sum_{i=1}^{k} \mathbb{I}\{s_{i,h} = s, a_{i,h}^{\pi_i(\mathcal{B}_{i,h})} = a\} \sum_{j=h}^{H-1} \varepsilon_{i,j}^{V} \leqslant 2\sqrt{H^3 N_{k,h}(s,a)L},$$

$$\sum_{i=1}^{k} \mathbb{I}\{s_{i,h} = s, a_{i,h}^{\pi_i(\mathcal{B}_{i,h})} = a\} \sum_{j=h}^{H-1} \xi_{i,j}^{V} \leqslant 2\sqrt{H^3 N_{k,h}(s,a)L},$$

$$\sum_{i=1}^{k} \mathbb{I}\{s_{i,h} = s, a_{i,h}^{\pi_i(\mathcal{B}_{i,h})} = a\} \sum_{j=h}^{H-1} \varepsilon_{i,j}^{Q} \leqslant 2\sqrt{H^3 N_{k,h}(s,a)L},$$

$$\sum_{i=1}^{k} \mathbb{I}\{s_{i,h} = s, a_{i,h}^{\pi_i(\mathcal{B}_{i,h})} = a\} \sum_{j=h}^{H-1} \overline{\varepsilon}_{i,j}^{V} \leqslant \sqrt{HN_{k,h}(s,a)},$$

$$\sum_{i=1}^{k} \mathbb{I}\{s_{i,h} = s, a_{i,h}^{\pi_i(\mathcal{B}_{i,h})} = a\} \sum_{j=h}^{H-1} \overline{\xi}_{i,j}^{V} \leqslant \sqrt{HN_{k,h}(s,a)},$$

$$\sum_{i=1}^{k} \mathbb{I}\{s_{i,h} = s, a_{i,h}^{\pi_i(\mathcal{B}_{i,h})} = a\} \sum_{j=h}^{H-1} \overline{\varepsilon}_{i,j}^{Q} \leqslant \sqrt{HN_{k,h}(s,a)}.$$

*Proof.* The proof follows a similar reasoning as that of Lemma 5 of (Azar et al., 2017), observing that under $\mathcal{E}$ and $\Omega_{k,h}$ the following events hold:

$$\mathcal{E}_{\mathrm{az}}(\mathcal{F}_{\widetilde{\Delta}^V, k, h}, H, L), \mathcal{E}_{\mathrm{az}}(\mathcal{F}_{\widetilde{\Delta}^V, k, h, s}, H, L), \mathcal{E}_{\mathrm{az}}(\mathcal{F}_{\widetilde{\Delta}^V, k, h, s, a}, H, L),$$

$$\mathcal{E}_{\mathrm{az}}(\mathcal{F}_{\widetilde{\Delta}^V, k, h, \mathcal{B}}, H, L), \mathcal{E}_{\mathrm{az}}(\mathcal{F}_{\widetilde{\Delta}^V, k, h, \mathcal{B}, s}, H, L), \mathcal{E}_{\mathrm{az}}(\mathcal{F}_{\widetilde{\Delta}^V, k, h, \mathcal{B}, s, a}, H, L),$$

$$\mathcal{E}_{\mathrm{az}}(\mathcal{F}_{\widetilde{\Delta}^Q, k, h}, H, L), \mathcal{E}_{\mathrm{az}}(\mathcal{F}_{\widetilde{\Delta}^Q, k, h, s}, H, L), \mathcal{E}_{\mathrm{az}}(\mathcal{F}_{\widetilde{\Delta}^Q, k, h, s, a}, H, L),$$

$$\mathcal{E}_{\mathrm{az}}(\mathcal{F}'_{\widetilde{\Delta}^V, k, h}, \frac{1}{\sqrt{L}}, L), \mathcal{E}_{\mathrm{az}}(\mathcal{F}'_{\widetilde{\Delta}^V, k, h, s}, \frac{1}{\sqrt{L}}, L), \mathcal{E}_{\mathrm{az}}(\mathcal{F}'_{\widetilde{\Delta}^V, k, h, s, a}, \frac{1}{\sqrt{L}}, L),$$

$$\mathcal{E}_{\mathrm{az}}(\mathcal{F}'_{\widetilde{\Delta}^V, k, h, \mathcal{B}}, \frac{1}{\sqrt{L}}, L), \mathcal{E}_{\mathrm{az}}(\mathcal{F}'_{\widetilde{\Delta}^V, k, h, \mathcal{B}, s}, \frac{1}{\sqrt{L}}, L), \mathcal{E}_{\mathrm{az}}(\mathcal{F}'_{\widetilde{\Delta}^V, k, h, \mathcal{B}, s, a}, \frac{1}{\sqrt{L}}, L),$$

$$\mathcal{E}_{\mathrm{az}}(\mathcal{F}'_{\widetilde{\Delta}^Q, k, h}, \frac{1}{\sqrt{L}}, L), \mathcal{E}_{\mathrm{az}}(\mathcal{F}'_{\widetilde{\Delta}^Q, k, h, s}, \frac{1}{\sqrt{L}}, L), \mathcal{E}_{\mathrm{az}}(\mathcal{F}'_{\widetilde{\Delta}^Q, k, h, s, a}, \frac{1}{\sqrt{L}}, L).$$

$\square$

**Remark D.1.** *With a slight abuse of notation, but to benefit the ease of reading, we will refer to the following lemmas also for the summations in which the action set is not fixed but is taken in expectation over $C^{\mathrm{ind}}$. As an example, we will refer to Lemma D.4 when upper bounding the summation:*

$$\sum_{i=1}^{k} \sum_{j=h}^{H-1} \mathbb{E}_{\mathcal{B} \sim C^{\mathrm{ind}}(\cdot | s_{i,j})} \left[ \mathbb{V}_{j+1}^{\pi_i}(s_{i,j}, a_{i,j}^{\pi_i(\mathcal{B})}) \right]. \tag{29}$$

*Considering the summations of the random variables that define the events (see Lemma 1 of Azar et al., 2017) and changing the conditioning from*
$mathcal{H}_{k,h,\mathcal{B}}$ *to* $\mathcal{H}_{k,h}$ *(i.e., removing $\mathcal{B}_{k,h}$ from the history) we see that the quantities in play retain the properties that make them Martingale difference sequences. As such, we can apply the same derivations, obtaining the same bounds.*

**Lemma D.4** (Lemma 8 of Azar et al. (2017))**.** *Let $k \in [\![K]\!]$ and $h \in [\![H]\!]$. Let $\pi_k$ be the policy followed during episode $k$. Under the events $\mathcal{E}$ and $\Omega_{k,h}$, the following holds for every $s \in \mathcal{S}$:*

$$\sum_{i=1}^{k}\sum_{j=h}^{H-1}\mathbb{V}_{j+1}^{\pi_i}(s_{i,j},a_{i,j}^{\pi_i(\mathcal{B}_{i,j})}) \leqslant HT_k + 2\sqrt{H^4 T_k L} + \frac{4}{3}H^3 L,$$

$$\sum_{i=1}^{k}\mathbb{I}\{s_{i,h}=s\}\sum_{j=h}^{H-1}\mathbb{V}_{j+1}^{\pi_i}(s_{i,j},a_{i,j}^{\pi_i(\mathcal{B}_{i,j})}) \leqslant H^2 N_{k,h}(s) + 2\sqrt{H^5 N_{k,h}(s) L} + \frac{4}{3}H^3 L,$$

$$\sum_{i=1}^{k}\mathbb{I}\{s_{i,h}=s, a_{i,h}^{\pi_i(\mathcal{B}_{i,h})}=a\}\sum_{j=h}^{H-1}\mathbb{V}_{j+1}^{\pi_i}(s_{i,j},a_{i,j}^{\pi_i(\mathcal{B}_{i,j})}) \leqslant H^2 N_{k,h}(s,a) + 2\sqrt{H^5 N_{k,h}(s,a) L} + \frac{4}{3}H^3 L,$$

*Proof.* The proof of this Lemma directly derives from the one of Lemma 8 of (Azar et al., 2017), observing that under event $\mathcal{E}$, the following events hold:

$$\mathcal{E}_{\mathrm{fr}}(\mathcal{G}_{\mathbb{V},k,h}, H^4 T_k, H^3, L),$$
$$\mathcal{E}_{\mathrm{fr}}(\mathcal{G}_{\mathbb{V},k,h,s}, H^5 N'_{k,h}, H^3, L),$$
$$\mathcal{E}_{\mathrm{fr}}(\mathcal{G}_{\mathbb{V},k,h,s,a}, H^5 N'_{k,h}, H^3, L),$$

and by taking the expectation over both the states and the action availabilities. Eq. (43) and Eq. (44) of (Azar et al., 2017) hold under this modified expectation since the cumulative reward is bounded with $H$. ☐

**Lemma D.5** (Lemma 9 of Azar et al. (2017)). *Let $k \in [\![K]\!]$ and $h \in [\![H]\!]$. Let $\pi_k$ be the policy followed during episode $k$. Under the events $\mathcal{E}$ and $\Omega_{k,h}$, the following holds for every $(s,a) \in \mathcal{S} \times \mathcal{A}$:*

$$\sum_{i=1}^{k}\sum_{j=h}^{H-1}\left(\mathbb{V}_{j+1}^{*}(s_{i,j},a_{i,j}^{\pi_i(\mathcal{B}_{i,j})}) - \mathbb{V}_{j+1}^{\pi_i}(s_{i,j},a_{i,j}^{\pi_i(\mathcal{B}_{i,j})})\right)$$
$$\leqslant 4\sqrt{H^4 T_k L} + 2H\sum_{i=1}^{k}\sum_{j=h}^{H-1}\widetilde{\Delta}_{i,j}^{V}(s_{i,j}),$$

$$\sum_{i=1}^{k}\mathbb{I}\{s_{i,h}=s\}\sum_{j=h}^{H-1}\left(\mathbb{V}_{j+1}^{*}(s_{i,j},a_{i,j}^{\pi_i(\mathcal{B}_{i,j})}) - \mathbb{V}_{j+1}^{\pi_i}(s_{i,j},a_{i,j}^{\pi_i(\mathcal{B}_{i,j})})\right)$$
$$\leqslant 4\sqrt{H^5 N_{k,h}(s) L} + 2H\sum_{i=1}^{k}\mathbb{I}\{s_{i,h}=s\}\sum_{j=h}^{H-1}\widetilde{\Delta}_{i,j}^{V}(s_{i,j}),$$

$$\sum_{i=1}^{k}\mathbb{I}\{s_{i,h}=s, a_{i,h}^{\pi_i(\mathcal{B}_{i,h})}=a\}\sum_{j=h}^{H-1}\left(\mathbb{V}_{j+1}^{*}(s_{i,j},a_{i,j}^{\pi_i(\mathcal{B}_{i,j})}) - \mathbb{V}_{j+1}^{\pi_i}(s_{i,j},a_{i,j}^{\pi_i(\mathcal{B}_{i,j})})\right)$$
$$\leqslant 4\sqrt{H^5 N_{k,h}(s,a) L} + 2H\sum_{i=1}^{k}\mathbb{I}\{s_{i,h}=s, a_{i,h}^{\pi_i(\mathcal{B}_{i,h})}=a\}\sum_{j=h}^{H-1}\widetilde{\Delta}_{i,j}^{V}(s_{i,j}).$$

*Proof.* The proof directly follows from the proof of Lemma 9 of (Azar et al., 2017), observing that events $\mathcal{E}_{\mathrm{az}}(\mathcal{F}_{\widetilde{\Delta}^V,k,h}, H, L), \mathcal{E}_{\mathrm{az}}(\mathcal{F}_{\widetilde{\Delta}^V,k,h,s}, H, L)$, and $\mathcal{E}_{\mathrm{az}}(\mathcal{F}_{\widetilde{\Delta}^V,k,h,s}, H, L)$ hold under $\mathcal{E}$, and $\Omega_{k,h}$ holds. ☐

**Lemma D.6** (Lemma 10 of Azar et al. (2017)). *Let $k \in [\![K]\!]$ and $h \in [\![H]\!]$. Let $\pi_k$ be the policy followed during episode $k$. Under the events $\mathcal{E}$ and $\Omega_{k,h}$, the following holds for every $s \in \mathcal{S}$:*

$$\sum_{i=1}^{k}\sum_{j=h}^{H-1}\left(\widehat{\mathbb{V}}_{i,j+1}(s_{i,j},a_{i,j}^{\pi_i(\mathcal{B}_{i,j})}) - \mathbb{V}_{j+1}^{\pi_i}(s_{i,j},a_{i,j}^{\pi_i(\mathcal{B}_{i,j})})\right)$$

$$\leqslant 7H^2 S\sqrt{AT_k L} + 2H \sum_{i=1}^{k} \sum_{j=h}^{H-1} \widetilde{\Delta}_{i,j+1}^{V}(s_{i,j+1}),$$

$$\sum_{i=1}^{k} \mathbb{I}\{s_{i,h} = s\} \sum_{j=h}^{H-1} \left( \widehat{\mathbb{V}}_{i,j+1}(s_{i,j}, a_{i,j}^{\pi_i(\mathcal{B}_{i,j})}) - \mathbb{V}_{j+1}^{\pi_i}(s_{i,j}, a_{i,j}^{\pi_i(\mathcal{B}_{i,j})}) \right)$$

$$\leqslant 7H^2 S\sqrt{HAN_{k,h}(s)L} + 2H \sum_{i=1}^{k} \mathbb{I}\{s_{i,h} = s\} \sum_{j=h}^{H-1} \widetilde{\Delta}_{i,j+1}^{V}(s_{i,j+1}),$$

$$\sum_{i=1}^{k} \mathbb{I}\{s_{i,h} = s, a_{i,h}^{\pi_i(\mathcal{B}_{i,h})} = a\} \sum_{j=h}^{H-1} \left( \widehat{\mathbb{V}}_{i,j+1}(s_{i,j}, a_{i,j}^{\pi_i(\mathcal{B}_{i,j})}) - \mathbb{V}_{j+1}^{\pi_i}(s_{i,j}, a_{i,j}^{\pi_i(\mathcal{B}_{i,j})}) \right)$$

$$\leqslant 7H^2 S\sqrt{HAN_{k,h}(s,a)L} + 2H \sum_{i=1}^{k} \mathbb{I}\{s_{i,h} = s, a_{i,h}^{\pi_i(\mathcal{B}_{i,h})} = a\} \sum_{j=h}^{H-1} \widetilde{\Delta}_{i,j+1}^{V}(s_{i,j+1}).$$

*Proof.* The proof directly follows that of Lemma 10 of (Azar et al., 2017), observing that under $\Omega_{k,h}$ and $\mathcal{E}$, the following events hold:

$$\mathcal{E}_{\mathrm{az}}(\mathcal{F}_{\widetilde{\Delta}^V, k, h}, H, L), \mathcal{E}_{\mathrm{az}}(\mathcal{F}_{\widetilde{\Delta}^V, k, h, s}, H, L), \mathcal{E}_{\mathrm{az}}(\mathcal{F}_{\widetilde{\Delta}^V, k, h, s, a}, H, L).$$

$\square$

**Lemma D.7.** *Let $k \in [\![K]\!]$ and $h \in [\![H]\!]$. Let $\pi_k$ be the policy followed during episode $k$. Under the events $\mathcal{E}$ and $\Omega_{k,h}$, the following holds for every $s \in \mathcal{S}$:*

$$\sum_{i=1}^{k} \sum_{j=h}^{H-1} \mathbb{Q}_{j}^{\pi_i}(s_{i,j}) \leqslant HT_k + 2\sqrt{H^4 T_k L} + \frac{4}{3}H^3 L,$$

$$\sum_{i=1}^{k} \mathbb{I}\{s_{i,h} = s\} \sum_{j=h}^{H-1} \mathbb{Q}_{j}^{\pi_i}(s_{i,j}) \leqslant H^2 N_{k,h}(s) + 2\sqrt{H^5 N_{k,h}(s)L} + \frac{4}{3}H^3 L,$$

$$\sum_{i=1}^{k} \mathbb{I}\{s_{i,h} = s, a_{i,h}^{\pi_i(\mathcal{B}_{i,h})} = a\} \sum_{j=h}^{H-1} \mathbb{Q}_{j}^{\pi_i}(s_{i,j}) \leqslant H^2 N_{k,h}(s,a) + 2\sqrt{H^5 N_{k,h}(s,a)L} + \frac{4}{3}H^3 L.$$

*Proof.* This proof follows a derivation similar to that of Lemma 8 of (Azar et al., 2017). Let us first recall the definition of $\mathbb{Q}_{j}^{\pi_i}(s_{i,j})$:

$$\mathbb{Q}_{j}^{\pi_i}(s_{i,j}) = \operatorname*{\mathbb{V}ar}_{\mathcal{B} \sim C^{\mathrm{ind}}(\cdot | s_{i,j})} \left[ Q_{j}^{\pi_i}(s_{i,j}, a_{i,j}^{\pi_i(\mathcal{B})}) \right].$$

Under event $\mathcal{E}$, the following events hold:

$$\mathcal{E}_{\mathrm{fr}}(\mathcal{G}_{\mathbb{Q}, k, h}, H^4 T_k, H^3, L), \mathcal{E}_{\mathrm{fr}}(\mathcal{G}_{\mathbb{Q}, k, h, s}, H^5 N_{k,h}, H^3 L), \text{ and } \mathcal{E}_{\mathrm{fr}}(\mathcal{G}_{\mathbb{Q}, k, h, s, a}, H^5 N_{k,h}, H^3 L).$$

Event $\mathcal{E}_{\mathrm{fr}}(\mathcal{G}_{\mathbb{Q}, k, h}, H^4 T_k, H^3, L)$ implies that:

$$\sum_{i=1}^{k} \sum_{j=h}^{H-1} \mathbb{Q}_{j}^{\pi_i}(s_{i,j}) \leqslant \sum_{i=1}^{k} \mathbb{E}\left[ \sum_{j=h}^{H-1} \mathbb{Q}_{j}^{\pi_i}(s_{i,j}) | \mathcal{H}_{k,h} \right] + 2\sqrt{H^4 T_k L} + \frac{4}{3}H^3 L.$$

Observing that, conditioned to $\mathcal{H}_{k,h}$, $\mathbb{Q}_{j}^{\pi_i}(s_{i,j}) = \mathbb{V}_{j+1}^{\pi_i}(s_{i,j}, a_{i,j}^{\pi_i(\mathcal{B}_{i,j})})$ since there is no variance on the reward at stage $j$, we can then recursively apply the *Law of Total Variance* (see e.g., Thr. 9.5.5 of Blitzstein & Hwang, 2019) as done in (Azar et al., 2017, Eq. 26) to obtain the following bound:

$$\sum_{i=1}^{k} \mathbb{E}\left[\sum_{j=h}^{H-1} \mathbb{Q}_j^{\pi_i}(s_{i,j})|\mathcal{H}_{k,h}\right] = \operatorname*{Var}_{(s_{i,j},\mathcal{B}_{i,j})_{j\in[\![h+1,H-1]\!]}} \left[\sum_{j=h+1}^{H-1} R^{\pi_i}(s_{i,j},a_{i,j}^{\pi_i(\mathcal{B}_{i,j})})\right] \leqslant HT_k.$$

Putting everything together, we obtain the following bound:

$$\sum_{i=1}^{k}\sum_{j=h}^{H-1} \mathbb{Q}_j^{\pi_i}(s_{i,j}) \leqslant HT_k + 2\sqrt{H^4 T_k L} + \frac{4}{3}H^3 L.$$

In a similar manner, from events $\mathcal{E}_{\mathrm{az}}(\mathcal{F}_{\widetilde{\Delta}^Q,k,h,s}, H, L)$ and $\mathcal{E}_{\mathrm{az}}(\mathcal{F}_{\widetilde{\Delta}^Q,k,h,s,a}, H, L)$ we can derive the bounds for the two remaining summations, thus concluding the proof. $\qquad\square$

**Lemma D.8.** *Let $k \in [\![K]\!]$ and $h \in [\![H]\!]$. Let $\pi_k$ be the policy followed during episode $k$. Under the events $\mathcal{E}$ and $\Omega_{k,h}$, the following holds for every $s \in \mathcal{S}$:*

$$\sum_{i=1}^{k}\sum_{j=h}^{H-1}\left(\widehat{\mathbb{Q}}_{i,j}(s_{i,j}) - \mathbb{Q}_j^{\pi_i}(s_{i,j})\right) \leqslant H^2\sqrt{S2^A T_k L} + 7H^2 S\sqrt{AT_k L} + 4H^3 SLG$$

$$+ 2H\sum_{i=1}^{k}\sum_{j=h}^{H-1} b_{i,j}^Q(s_{i,j}, a_{i,j}^{\pi_i(\mathcal{B}_{i,j})}) + 2H\sum_{i=1}^{k}\sum_{j=h}^{H-1} \widetilde{\Delta}_{i,j+1}^V(s_{i,j+1}),$$

$$\sum_{i=1}^{k}\mathbb{I}\{s_{i,h} = s\}\sum_{j=h}^{H-1}\left(\widehat{\mathbb{Q}}_{i,j}(s_{i,j}) - \mathbb{Q}_j^{\pi_i}(s_{i,j})\right)$$

$$\leqslant \sqrt{H^5 S2^A N_{k,h}(s)L} + 7\sqrt{H^5 S^2 A N_{k,h}(s)L} + 4H^3 SLG$$

$$+ 2H\sum_{i=1}^{k}\mathbb{I}\{s_{i,h} = s\}\sum_{j=h}^{H-1} b_{i,j}^Q(s_{i,j}, a_{i,j}^{\pi_i(\mathcal{B}_{i,j})})$$

$$+ 2H\sum_{i=1}^{k}\mathbb{I}\{s_{i,h} = s\}\sum_{j=h}^{H-1} \widetilde{\Delta}_{i,j+1}^V(s_{i,j+1}),$$

$$\sum_{i=1}^{k}\mathbb{I}\{s_{i,h} = s, a_{i,h}^{\pi_i(\mathcal{B}_{i,h})} = a\}\sum_{j=h}^{H-1}\left(\widehat{\mathbb{Q}}_{i,j}(s_{i,j}) - \mathbb{Q}_j^{\pi_i}(s_{i,j})\right)$$

$$\leqslant \sqrt{H^5 S2^A N_{k,h}(s,a)L} + 7\sqrt{H^5 S^2 A N_{k,h}(s,a)L} + 4H^3 SLG$$

$$+ 2H\sum_{i=1}^{k}\mathbb{I}\{s_{i,h} = s, a_{i,h}^{\pi_i(\mathcal{B}_{i,h})} = a\}\sum_{j=h}^{H-1} b_{i,j}^Q(s_{i,j}, a_{i,j}^{\pi_i(\mathcal{B}_{i,j})})$$

$$+ 2H\sum_{i=1}^{k}\mathbb{I}\{s_{i,h} = s, a_{i,h}^{\pi_i(\mathcal{B}_{i,h})} = a\}\sum_{j=h}^{H-1} \widetilde{\Delta}_{i,j+1}^V(s_{i,j+1}).$$

*Proof.* We first provide an upper bound to $\widehat{\mathbb{Q}}_{i,j}(s_{i,j}) - \mathbb{Q}_j^{\pi_i}(s_{i,j})$, and we then bound its summation over episodes and stages.

$$\widehat{\mathbb{Q}}_{i,j}(s_{i,j}) - \mathbb{Q}_j^{\pi_i}(s_{i,j}) = \mathbb{E}_{\mathcal{B}\sim\widehat{C}_i^{\mathrm{ind}}(\cdot|s_{i,j})}\left[\widehat{Q}_{i,j}(s_{i,j}, a_{i,j}^{\pi_i(\mathcal{B})})^2\right] - \mathbb{E}_{\mathcal{B}\sim\widehat{C}_i^{\mathrm{ind}}(\cdot|s_{i,j})}\left[\widehat{Q}_{i,j}(s_{i,j}, a_{i,j}^{\pi_i(\mathcal{B})})\right]^2$$

$$- \mathbb{E}_{\mathcal{B}\sim C^{\mathrm{ind}}(\cdot|s_{i,j})}\left[Q_j^{\pi_i}(s_{i,j}, a_{i,j}^{\pi_i(\mathcal{B})})^2\right] + \mathbb{E}_{\mathcal{B}\sim C^{\mathrm{ind}}(\cdot|s_{i,j})}\left[Q_j^{\pi_i}(s_{i,j}, a_{i,j}^{\pi_i(\mathcal{B})})\right]^2$$

$$\leqslant \mathbb{E}_{\mathcal{B}\sim\widehat{C}_i^{\mathrm{ind}}(\cdot|s_{i,j})}\left[\widehat{Q}_{i,j}(s_{i,j},a_{i,j}^{\pi_i(\mathcal{B})})^2\right] - \mathbb{E}_{\mathcal{B}\sim C^{\mathrm{ind}}(\cdot|s_{i,j})}\left[Q_j^{\pi_i}(s_{i,j},a_{i,j}^{\pi_i(\mathcal{B})})^2\right]$$

$$- \mathbb{E}_{\mathcal{B}\sim\widehat{C}_i^{\mathrm{ind}}(\cdot|s_{i,j})}\left[Q_j^*(s_{i,j},a_{i,j}^{\pi^*(\mathcal{B})})\right]^2 + \mathbb{E}_{\mathcal{B}\sim C^{\mathrm{ind}}(\cdot|s_{i,j})}\left[Q_j^*(s_{i,j},a_{i,j}^{\pi^*(\mathcal{B})})\right]^2 \qquad (30)$$

$$\leqslant \mathbb{E}_{\mathcal{B}\sim\widehat{C}_i^{\mathrm{ind}}(\cdot|s_{i,j})}\left[\widehat{Q}_{i,j}(s_{i,j},a_{i,j}^{\pi_i(\mathcal{B})})^2\right] - \mathbb{E}_{\mathcal{B}\sim C^{\mathrm{ind}}(\cdot|s_{i,j})}\left[\widehat{Q}_{i,j}(s_{i,j},a_{i,j}^{\pi_i(\mathcal{B})})^2\right]$$

$$+ \mathbb{E}_{\mathcal{B}\sim C^{\mathrm{ind}}(\cdot|s_{i,j})}\left[\widehat{Q}_{i,j}(s_{i,j},a_{i,j}^{\pi_i(\mathcal{B})})^2\right] - \mathbb{E}_{\mathcal{B}\sim C^{\mathrm{ind}}(\cdot|s_{i,j})}\left[Q_j^{\pi_i}(s_{i,j},a_{i,j}^{\pi_i(\mathcal{B})})^2\right]$$

$$+ 2H\left((\widehat{C}_i^{\mathrm{ind}} - C^{\mathrm{ind}})Q_j^*\right)(s_{i,j})$$

$$\leqslant \mathbb{E}_{\mathcal{B}\sim\widehat{C}_i^{\mathrm{ind}}(\cdot|s_{i,j})}\left[\widehat{Q}_{i,j}(s_{i,j},a_{i,j}^{\pi_i(\mathcal{B})})^2\right] - \mathbb{E}_{\mathcal{B}\sim C^{\mathrm{ind}}(\cdot|s_{i,j})}\left[\widehat{Q}_{i,j}(s_{i,j},a_{i,j}^{\pi_i(\mathcal{B})})^2\right]$$

$$+ \mathbb{E}_{\mathcal{B}\sim C^{\mathrm{ind}}(\cdot|s_{i,j})}\left[\widehat{Q}_{i,j}(s_{i,j},a_{i,j}^{\pi_i(\mathcal{B})})^2\right] - \mathbb{E}_{\mathcal{B}\sim C^{\mathrm{ind}}(\cdot|s_{i,j})}\left[Q_j^{\pi_i}(s_{i,j},a_{i,j}^{\pi_i(\mathcal{B})})^2\right]$$

$$+ 4H\frac{H^2 L}{N_i(s_{i,j})}, \qquad (31)$$

where Eq. (30) follows from observing that, under $\Omega_{k,h}$, $\widehat{Q}_{i,j} \geqslant Q_j^* \geqslant Q_j^{\pi_i}$, $\forall (s,a) \in \mathcal{S} \times \mathcal{A}$, and Eq. (31) is obtained via Hoeffding's inequality. Putting this bound in the double summation, we get:

$$\sum_{i=1}^{k}\sum_{j=h}^{H-1}\left(\widehat{\mathbb{Q}}_{i,j}(s_{i,j}) - \mathbb{Q}_j^{\pi_i}(s_{i,j})\right) \leqslant \underbrace{\sum_{i=1}^{k}\sum_{j=h}^{H-1}\left(\mathbb{E}_{\mathcal{B}\sim\widehat{C}_i^{\mathrm{ind}}(\cdot|s_{i,j})}\left[\widehat{Q}_{i,j}(s_{i,j},a_{i,j}^{\pi_i(\mathcal{B})})^2\right] - \mathbb{E}_{\mathcal{B}\sim C^{\mathrm{ind}}(\cdot|s_{i,j})}\left[\widehat{Q}_{i,j}(s_{i,j},a_{i,j}^{\pi_i(\mathcal{B})})^2\right]\right)}_{(a)}$$

$$+ \underbrace{\sum_{i=1}^{k}\sum_{j=h}^{H-1}\left(\mathbb{E}_{\mathcal{B}\sim C^{\mathrm{ind}}(\cdot|s_{i,j})}\left[\widehat{Q}_{i,j}(s_{i,j},a_{i,j}^{\pi_i(\mathcal{B})})^2 - Q_j^{\pi_i}(s_{i,j},a_{i,j}^{\pi_i(\mathcal{B})})^2\right]\right)}_{(b)}$$

$$+ \underbrace{\sum_{i=1}^{k}\sum_{j=h}^{H-1}\left(4H\frac{H^2 L}{N_i(s_{i,j})}\right)}_{(c)} \qquad (32)$$

We can bound term $(a)$ with $H^2\sqrt{S2^A T_k L}$ by bounding $\widehat{Q}_{i,j}$ with $H$ and applying Thr. 2.1 of (Weissman et al., 2003), and term $(c)$ with $4H^3 SLG$ by applying a pigeonhole argument. We now bound term $(b)$ as follows:

$$(b) = \sum_{i=1}^{k}\sum_{j=h}^{H-1}\sum_{\mathcal{B}\in\mathcal{P}(\mathcal{A})}C^{\mathrm{ind}}(\mathcal{B}|s_{i,j})\left((\widehat{Q}_{i,j}(s_{i,j},a_{i,j}^{\pi_i(\mathcal{B})}) + Q_j^{\pi_i}(s_{i,j},a_{i,j}^{\pi_i(\mathcal{B})}))(\widehat{Q}_{i,j}(s_{i,j},a_{i,j}^{\pi_i(\mathcal{B})}) - Q_j^{\pi_i}(s_{i,j},a_{i,j}^{\pi_i(\mathcal{B})}))\right)$$

$$\leqslant 2H\sum_{i=1}^{k}\sum_{j=h}^{H-1}\sum_{\mathcal{B}\in\mathcal{P}(\mathcal{A})}C^{\mathrm{ind}}(\mathcal{B}|s_{i,j})(\widehat{Q}_{i,j}(s_{i,j},a_{i,j}^{\pi_i(\mathcal{B})}) - Q_j^{\pi_i}(s_{i,j},a_{i,j}^{\pi_i(\mathcal{B})}))$$

$$\leqslant 2H\sum_{i=1}^{k}\sum_{j=h}^{H-1}\varepsilon_{i,j}^Q + 2H\sum_{i=1}^{k}\sum_{j=h}^{H-1}b_{i,j}^Q(s_{i,j},a_{i,j}^{\pi_i(\mathcal{B}_{i,j})})$$

$$+ 2H\sum_{i=1}^{k}\sum_{j=h}^{H-1}\left(\mathbb{E}_{s'\in\widehat{P}_i(\cdot|s_{i,j},a_{i,j}^{\pi_i(\mathcal{B}_{i,j})})}\widehat{V}_{i,j+1}(s') - \mathbb{E}_{s'\sim P(\cdot|s_{i,j},a_{i,j}^{\pi_i(\mathcal{B}_{i,j})})}V_{j+1}^{\pi_i}(s')\right)$$

$$\leqslant 2H\sum_{i=1}^{k}\sum_{j=h}^{H-1}\varepsilon_{i,j}^Q + 2H\sum_{i=1}^{k}\sum_{j=h}^{H-1}b_{i,j}^Q(s_{i,j},a_{i,j}^{\pi_i(\mathcal{B}_{i,j})}) + 2H\sum_{i=1}^{k}\sum_{j=h}^{H-1}\varepsilon_{i,j}^V + 2H\sum_{i=1}^{k}\sum_{j=h}^{H-1}\widetilde{\Delta}_{i,j+1}^V(s_{i,j+1})$$

$$+ 2H^2 \sum_{i=1}^{k} \sum_{j=h}^{H-1} \|\widehat{P}_i(\cdot|s_{i,j}, a_{i,j}^{\pi_i(\mathcal{B}_{i,j})}) - P((\cdot|s_{i,j}, a_{i,j}^{\pi_i(\mathcal{B}_{i,j})})\|_1 \tag{33}$$

$$\leqslant 2H^2\sqrt{T_k L} + 2H \sum_{i=1}^{k} \sum_{j=h}^{H-1} b_{i,j}^{Q}(s_{i,j}, a_{i,j}^{\pi_i(\mathcal{B}_{i,j})}) + H^2 S\sqrt{AT_k L} + 4H^2\sqrt{T_k L} + 2H \sum_{i=1}^{k} \sum_{j=h}^{H-1} \widetilde{\Delta}_{i,j+1}^{V}(s_{i,j+1}), \tag{34}$$

where Eq. (33) is obtained by adding and subtracting $2H \sum_{i=1}^{k} \sum_{j=h}^{H-1} (P\widehat{V}_{i,j+1})(s_{i,j}, a_{i,j}^{\pi_i(\mathcal{B}_{i,j})})$ and $2H \sum_{i=1}^{k} \sum_{j=h}^{H-1} \widetilde{\Delta}_{i,j+1}^{V}(s_{i,j+1})$, and Eq. (34) is obtained since event $\mathcal{E}_{\mathrm{az}}(\mathcal{F}_{\widetilde{\Delta}^{Q},k,h}, H, L)$ holds under $\mathcal{E}$. Plugging the bounds of terms $(a)$, $(b)$, and $(c)$ into Eq. (32), we finally obtain:

$$\sum_{i=1}^{k} \sum_{j=h}^{H-1} \left( \widehat{\mathbb{Q}}_{i,j}(s_{i,j}) - \mathbb{Q}_j^{\pi_i}(s_{i,j}) \right) \leqslant H^2\sqrt{S2^A T_k L} + 2H^2\sqrt{T_k L} + 2H \sum_{i=1}^{k} \sum_{j=h}^{H-1} b_{i,j}^{Q}(s_{i,j}, a_{i,j}^{\pi_i(\mathcal{B}_{i,j})})$$

$$+ H^2 S\sqrt{AT_k L} + 4H^2\sqrt{T_k L} + 2H \sum_{i=1}^{k} \sum_{j=h}^{H-1} \widetilde{\Delta}_{i,j+1}^{V}(s_{i,j+1}) + 4H^3 SLG$$

$$\leqslant H^2\sqrt{S2^A T_k L} + 7H^2 S\sqrt{AT_k L} + 2H \sum_{i=1}^{k} \sum_{j=h}^{H-1} b_{i,j}^{Q}(s_{i,j}, a_{i,j}^{\pi_i(\mathcal{B}_{i,j})})$$

$$+ 2H \sum_{i=1}^{k} \sum_{j=h}^{H-1} \widetilde{\Delta}_{i,j+1}^{V}(s_{i,j+1}) + 4H^3 SLG.$$

In a similar manner, and observing that events $\mathcal{E}_{\mathrm{az}}(\mathcal{F}_{\widetilde{\Delta}^{Q},k,h,s}, H, L)$ and $\mathcal{E}_{\mathrm{az}}(\mathcal{F}_{\widetilde{\Delta}^{Q},k,h,s,a}, H, L)$ hold under $\mathcal{E}$, we can derive the bounds to the remaining summations, thus concluding the proof. $\qquad\square$

**Lemma D.9** (Summation over typical episodes of state-action wise model errors, see Lemma 11 of Azar et al. (2017)). *Let $k \in [\![K]\!]$ and $h \in [\![H]\!]$. Let $\pi_k$ be the policy followed during episode $k$. Under events $\mathcal{E}$ and $\Omega_{k,h}$ the following inequalities hold for every $(s, a) \in \mathcal{S} \times \mathcal{A}$:*

$$\sum_{i=1}^{k} \mathbb{I}\{i \in [k]_{\mathrm{typ}}\} \sum_{j=h}^{H-1} \left( (\widehat{P}_i - P)V_{j+1}^{*} \right)(s_{i,j}, a_{i,j}^{\pi_i(\mathcal{B}_{i,j})})$$

$$\leqslant \sqrt{6HSAT_k LG} + \frac{2}{3}HSALG + 2\sqrt{HSALG \sum_{i=1}^{k} \sum_{j=h}^{H-1} \widetilde{\Delta}_{i,j}^{V}(s_{i,j})}, \tag{35}$$

$$\sum_{i=1}^{k} \mathbb{I}\{i \in [k]_{\mathrm{typ},s}, s_{i,h} = s\} \sum_{j=h}^{H-1} \left( (\widehat{P}_i - P)V_{j+1}^{*} \right)(s_{i,j}, a_{i,j}^{\pi_i(\mathcal{B}_{i,j})})$$

$$\leqslant \sqrt{6H^2 SAN_{k,h}(s)LG} + \frac{2}{3}HSALG$$

$$+ 2\sqrt{HSALG \sum_{i=1}^{k} \mathbb{I}\{s_{i,h} = s\} \sum_{j=h}^{H-1} \widetilde{\Delta}_{i,j}^{V}(s_{i,j})},$$

$$\sum_{i=1}^{k} \mathbb{I}\{i \in [k]_{\mathrm{typ},s,a}, s_{i,h} = s, a_{i,h}^{\pi_i(\mathcal{B}_{i,h})} = a\} \sum_{j=h}^{H-1} \left( (\widehat{P}_i - P)V_{j+1}^{*} \right)(s_{i,j}, a_{i,j}^{\pi_i(\mathcal{B}_{i,j})})$$

$$\leqslant \sqrt{6H^2 SAN_{k,h}(s,a)LG} + \frac{2}{3}HSALG$$

$$+ 2\sqrt{HSALG \sum_{i=1}^{k} \mathbb{I}\{s_{i,h} = s, a_{i,h}^{\pi_i(\mathcal{B}_{i,h})}\} \sum_{j=h}^{H-1} \widetilde{\Delta}_{i,j}^{V}(s_{i,j})},$$

*where:*

$$[k]_{\text{typ}} := \left\{ i \in [\![k]\!] : (s_{i,h}, a_{i,h}^{\pi_i(\mathcal{B}_{i,h})}) \in [(s,a)]_k, i \geqslant 11600 H^3 S^3 A 2^A L^2 G, \forall h \in [\![H]\!] \right\},$$

$$[k]_{\text{typ},s} := \left\{ i \in [\![k]\!] : (s_{i,h}, a_{i,h}^{\pi_i(\mathcal{B}_{i,h})}) \in [(s,a)]_k, N_{k,h}(s) \geqslant 11600 H^3 S^3 A 2^A L^2 G, \forall h \in [\![H]\!] \right\},$$

$$[k]_{\text{typ},s,a} := \left\{ i \in [\![k]\!] : (s_{i,h}, a_{i,h}^{\pi_i(\mathcal{B}_{i,h})}) \in [(s,a)]_k, N_{k,h}(s,a) \geqslant 11600 H^3 S^3 A 2^A L^2 G, \forall h \in [\![H]\!] \right\},$$

$$[(s,a)]_k := \left\{ (s,a) \in \mathcal{S} \times \mathcal{A} : N_k(s,a) \geqslant H, N_{k,h}(s) \geqslant H, \forall h \in [\![H]\!] \right\}.$$

*Proof.* We adapt the proof of (Azar et al., 2017, Lemma 11). We begin by demonstrating the bound of Eq. (35):

$$\sum_{i=1}^{k} \mathbb{I}\{i \in [k]_{\text{typ}}\} \sum_{j=h}^{H-1} \left( (\widehat{P}_i - P) V_{j+1}^* \right) (s_{i,j}, a_{i,j}^{\pi_i(\mathcal{B}_{i,j})})$$

$$\leqslant \sum_{i=1}^{k} \mathbb{I}\{i \in [k]_{\text{typ}}\} \sum_{j=h}^{H-1} \left[ \sqrt{\frac{2 \mathbb{V}_{j+1}^*(s_{i,j}, a_{i,j}^{\pi_i(\mathcal{B}_{i,j})}) L}{N_i(s_{i,j}, a_{i,j}^{\pi_i(\mathcal{B}_{i,j})})}} + \frac{2HL}{3 N_i(s_{i,j}, a_{i,j}^{\pi_i(\mathcal{B}_{i,j})})} \right] \qquad (36)$$

$$\leqslant \sqrt{2L} \underbrace{\sqrt{\sum_{i=1}^{k} \sum_{j=h}^{H-1} \mathbb{V}_{j+1}^*(s_{i,j}, a_{i,j}^{\pi_i(\mathcal{B}_{i,j})})}}_{(a)} \underbrace{\sqrt{\sum_{i=1}^{k} \mathbb{I}\{i \in [k]_{\text{typ}}\} \sum_{j=h}^{H-1} \frac{1}{N_i(s_{i,j}, a_{i,j}^{\pi_i(\mathcal{B}_{i,j})})}}}_{(b)} + \frac{2}{3} HSALG, \quad (37)$$

where Eq. (36) follows from the application of Bernstein's inequality and Eq. (37) follows from the application of Cauchy-Schwarz's inequality together with a pigeonhole argument.

Using another pigeonhole argument, we can bound term $(b)$ with $SAG$. By adding and subtracting $\mathbb{V}_{j+1}^{\pi_i}(s_{i,j}, a_{i,j}^{\pi_i(\mathcal{B}_{i,j})})$ to term $(a)$, we can rewrite it as:

$$(a) = \underbrace{\sum_{i=1}^{k} \sum_{j=h}^{H-1} \mathbb{V}_{j+1}^{\pi_i}(s_{i,j}, a_{i,j}^{\pi_i(\mathcal{B}_{i,j})})}_{(c)} + \underbrace{\sum_{i=1}^{k} \sum_{j=h}^{H-1} (\mathbb{V}_{j+1}^*(s_{i,j}, a_{i,j}^{\pi_i(\mathcal{B}_{i,j})}) - \mathbb{V}_{j+1}^{\pi_i}(s_{i,j}, a_{i,j}^{\pi_i(\mathcal{B}_{i,j})}))}_{(d)}.$$

As events $\mathcal{E}$ and $\Omega_{k,h}$ hold, we can apply Lemma D.4 and Lemma D.5 to bound terms $(c)$ and $(d)$, respectively, thus obtaining:

$$(a) \leqslant H T_k + 6\sqrt{H^4 T_k L} + \frac{4}{3} H^3 L + 2H \sum_{i=1}^{k} \sum_{j=h}^{H-1} \widetilde{\Delta}_{i,j}^{V}(s_{i,j})$$

$$\leqslant 3 H T_k + 2H \sum_{i=1}^{k} \sum_{j=h}^{H-1} \widetilde{\Delta}_{i,j}^{V}(s_{i,j}), \qquad (38)$$

where Eq. (38) holds under the condition of $[k]_{\text{typ}}$. Plugging the bounds of terms $(a)$ and $(b)$ into Eq. (37), rearranging the terms, and applying the subadditivity of the square root, we finally get:

$$\sum_{i=1}^{k} \mathbb{I}\{i \in [k]_{\text{typ}}\} \sum_{j=h}^{H-1} \left( (\widehat{P}_i - P) V_{j+1}^* \right) (s_{i,j}, a_{i,j}^{\pi_i(\mathcal{B}_{i,j})}) \leqslant \sqrt{6HSAT_kLG} + \frac{2}{3}HSALG$$

$$+ 2\sqrt{HSALG \sum_{i=1}^{k} \sum_{j=h}^{H-1} \widetilde{\Delta}_{i,j}^V(s_{i,j})}.$$

In a similar manner, we can demonstrate the bound on the remaining summations, thus concluding the proof. □

**Lemma D.10** (Summation over typical episodes of state-action value function bonus term). *Let $k \in [\![K]\!]$ and $h \in [\![H]\!]$. Let $\pi_k$ be the policy followed during episode $k$. Let the UCB bonus for the state-action value function be defined as:*

$$b_{k,h}^Q(s,a) = \sqrt{\frac{4L \, \mathbb{V}\text{ar}_{s' \sim \widehat{P}_k(\cdot|s,a)}[\widehat{V}_{k,h+1}(s')]}{N_k(s,a)}} + \frac{7HL}{3(N_k(s,a) - 1)}$$

$$+ \sqrt{\frac{4\mathbb{E}_{s' \sim \widehat{P}_k(\cdot|s,a)}[\min\{\frac{2900^2 H^3 S^3 A2^A L^3}{N'_{k,h+1}(s')}, H^2\}]}{N_k(s,a)}}.$$

*Under the events $\mathcal{E}$ and $\Omega_{k,h}$, the following inequalities hold for every $(s,a) \in \mathcal{S} \times \mathcal{A}$:*

$$\sum_{i=1}^{k} \mathbb{I}\{i \in [k]_{\text{typ}}\} \sum_{j=h}^{H-1} b_{i,j}^Q(s_{i,j}, a_{i,j}^{\pi_i(\mathcal{B}_{i,j})})$$

$$\leqslant \sqrt{28HSAT_kLG} + \frac{7}{3}HSALG + 5800\sqrt{H^3 S^5 A2^{2A} L^3 G^2}$$

$$+ \sqrt{8HSALG \sum_{i=1}^{k} \sum_{j=h}^{H-1} \widetilde{\Delta}_{i,j+1}^V(s_{i,j+1})},$$

$$\sum_{i=1}^{k} \mathbb{I}\{i \in [k]_{\text{typ,s}}, s_{i,h} = s\} \sum_{j=h}^{H-1} b_{i,j}^Q(s_{i,j}, a_{i,j}^{\pi_i(\mathcal{B}_{i,j})})$$

$$\leqslant \sqrt{28H^2 SAN_{k,h}(s)LG} + \frac{7}{3}HSALG + 5800\sqrt{H^3 S^5 A2^{2A} L^3 G^2}$$

$$+ \sqrt{8HSALG \sum_{i=1}^{k} \mathbb{I}\{s_{i,h} = s\} \sum_{j=h}^{H-1} \widetilde{\Delta}_{i,j+1}^V(s_{i,j+1})},$$

$$\sum_{i=1}^{k} \mathbb{I}\{i \in [k]_{\text{typ,s,a}}, s_{i,h} = s, a_{i,h}^{\pi_i(\mathcal{B}_{i,h})} = a\} \sum_{j=h}^{H-1} b_{i,j}^Q(s_{i,j}, a_{i,j}^{\pi_i(\mathcal{B}_{i,j})})$$

$$\leqslant \sqrt{28H^2 SAN_{k,h}(s,a)LG} + \frac{7}{3}HSALG + 5800\sqrt{H^3 S^5 A2^{2A} L^3 G^2}$$

$$+ \sqrt{8HSALG \sum_{i=1}^{k} \mathbb{I}\{s_{i,h} = s, a_{i,h}^{\pi_i(\mathcal{B}_{i,h})} = a\} \sum_{j=h}^{H-1} \widetilde{\Delta}_{i,j+1}^V(s_{i,j+1})}.$$

*where:*

$$[k]_{\text{typ}} := \left\{ i \in [\![k]\!] : (s_{i,h}, a_{i,h}^{\pi_i(\mathcal{B}_{i,h})}) \in [(s,a)]_k, i \geqslant 11600 H^3 S^3 A2^A L^2 G, \forall h \in [\![H]\!] \right\},$$

$$[k]_{\text{typ},s} := \left\{ i \in [\![k]\!] : (s_{i,h}, a_{i,h}^{\pi_i(\mathcal{B}_{i,h})}) \in [(s,a)]_k, N_{k,h}(s) \geqslant 11600 H^3 S^3 A 2^A L^2 G, \forall h \in [\![H]\!] \right\},$$

$$[k]_{\text{typ},s,a} := \left\{ i \in [\![k]\!] : (s_{i,h}, a_{i,h}^{\pi_i(\mathcal{B}_{i,h})}) \in [(s,a)]_k, N_{k,h}(s,a) \geqslant 11600 H^3 S^3 A 2^A L^2 G, \forall h \in [\![H]\!] \right\},$$

$$[(s,a)]_k := \left\{ (s,a) \in \mathcal{S} \times \mathcal{A} : N_k(s,a) \geqslant H, N_{k,h}(s) \geqslant H, \forall h \in [\![H]\!] \right\}.$$

*Proof.* The proof of this lemma closely follows that of (Azar et al., 2017, Lem. 12). We can rewrite the summation as:

$$\sum_{i=1}^{k} \mathbb{I}\{i \in [k]_{\text{typ}}\} \sum_{j=h}^{H-1} b_{i,j}^Q(s_{i,j}, a_{i,j}^{\pi_i(\mathcal{B}_{i,j})}) \leqslant \underbrace{\sum_{i=1}^{k} \mathbb{I}\{i \in [k]_{\text{typ}}\} \sum_{j=h}^{H-1} \sqrt{\frac{4L \operatorname{Var}_{s' \sim \widehat{P}_i(\cdot | s_{i,j}, a_{i,j}^{\pi_i(\mathcal{B}_{i,j})})}[\widehat{V}_{i,j+1}(s')]}{N_i(s_{i,j}, a_{i,j}^{\pi_i(\mathcal{B}_{i,j})})}}}_{(a)}$$

$$+ \underbrace{\sum_{i=1}^{k} \mathbb{I}\{i \in [k]_{\text{typ}}\} \sum_{j=h}^{H-1} \frac{7HL}{3(N_i(s_{i,j}, a_{i,j}^{\pi_i(\mathcal{B}_{i,j})}) - 1)}}_{(b)}$$

$$+ \underbrace{\sum_{i=1}^{k} \mathbb{I}\{i \in [k]_{\text{typ}}\} \sum_{j=h}^{H-1} \sqrt{\frac{4\mathbb{E}_{s' \sim \widehat{P}_i(\cdot | s_{i,j}, a_{i,j}^{\pi_i(\mathcal{B}_{i,j})})} \overline{b}_{i,j+1}^Q(s')}{N_i(s_{i,j}, a_{i,j}^{\pi_i(\mathcal{B}_{i,j})})}}}_{(c)}, \tag{39}$$

where $\overline{b}_{i,j+1}^Q(s') = \min\{\frac{2900^2 H^3 S^3 A 2^A L^3}{N'_{i,j+1}(s')}, H^2\}$. First of all, we bound term $(b)$ with $\frac{7}{3} HSALG$ by applying a pigeonhole argument. To bound term $(a)$, we apply Cauchy-Schwarz's inequality to obtain:

$$(a) \leqslant \sqrt{4L} \sqrt{\underbrace{\sum_{i=1}^{k} \sum_{j=h}^{H-1} \widehat{\mathbb{V}}_{i,j+1}(s_{i,j}, a_{i,j}^{\pi_i(\mathcal{B}_{i,j})})}_{(d)}} \sqrt{\underbrace{\sum_{i=1}^{k} \mathbb{I}\{i \in [k]_{\text{typ}}\} \sum_{j=h}^{H-1} \frac{1}{N_i(s_{i,j}, a_{i,j}^{\pi_i(\mathcal{B}_{i,j})})}}_{(e)}}. \tag{40}$$

By applying a pigeonhole argument, we bound term $(e)$ with $SAG$, and we rewrite term $(d)$ as follows:

$$(d) = \underbrace{\sum_{i=1}^{k} \sum_{j=h}^{H-1} \mathbb{V}_{j+1}^{\pi_i}(s_{i,j}, a_{i,j}^{\pi_i(\mathcal{B}_{i,j})})}_{(f)} + \underbrace{\sum_{i=1}^{k} \sum_{j=h}^{H-1} \left( \widehat{\mathbb{V}}_{i,j+1}(s_{i,j}, a_{i,j}^{\pi_i(\mathcal{B}_{i,j})}) - \mathbb{V}_{j+1}^{\pi_i}(s_{i,j}, a_{i,j}^{\pi_i(\mathcal{B}_{i,j})}) \right)}_{(g)}.$$

By applying Lemma D.4 and Lemma D.6 to bound terms $(f)$ and $(g)$, respectively, we obtain the following bound:

$$(d) \leqslant HT_k + 2\sqrt{H^4 T_k L} + \frac{4}{3} H^3 L + 7H^2 S \sqrt{AT_k L} + 2H \sum_{i=1}^{k} \sum_{j=h}^{H-1} \widetilde{\Delta}_{i,j+1}^V(s_{i,j+1})$$

$$\leqslant 4HT_k + 2H \sum_{i=1}^{k} \sum_{j=h}^{H-1} \widetilde{\Delta}_{i,j+1}^V(s_{i,j+1}), \tag{41}$$

where Eq. (41) holds under the condition of $[k]_{\text{typ}}$. Combining the bounds of terms $(d)$ and $(e)$ into Eq. (40) and applying the subadditivity of the square root, we obtain:

$$(a) \leqslant \sqrt{16HSAT_kLG} + \sqrt{8HSALG \sum_{i=1}^{k} \sum_{j=h}^{H-1} \widetilde{\Delta}_{i,j+1}^{V}(s_{i,j+1})}.$$

To bound term $(c)$, we first apply Cauchy-Schwarz's inequality, obtaining:

$$(c) \leqslant 2 \sqrt{\underbrace{\sum_{i=1}^{k} \sum_{j=h}^{H-1} \mathbb{E}_{s' \sim \widehat{P}_i(\cdot|s_{i,j}, a_{i,j}^{\pi_i(\mathcal{B}_{i,j})})} \overline{b}_{i,j+1}^{Q}(s')}_{(h)}} \sqrt{\underbrace{\sum_{i=1}^{k} \mathbb{I}\{i \in [k]_{\mathrm{typ}}\} \sum_{j=h}^{H-1} \frac{1}{N_i(s_{i,j}, a_{i,j}^{\pi_i(\mathcal{B}_{i,j})})}}_{(i)}}. \tag{42}$$

Similar to term $(e)$, we bound term $(i)$ with $SAG$. To bound term $(h)$, we first rewrite it as:

$$\begin{aligned}
(h) &= \sum_{i=1}^{k} \sum_{j=h}^{H-1} \left( \widehat{P}_i \overline{b}_{i,j+1}^{Q} \right) (s_{i,j}, a_{i,j}^{\pi_i(\mathcal{B}_{i,j})}) \\
&= \underbrace{\sum_{i=1}^{k} \sum_{j=h}^{H-1} \left( (\widehat{P}_i - P) \overline{b}_{i,j+1}^{Q} \right) (s_{i,j}, a_{i,j}^{\pi_i(\mathcal{B}_{i,j})})}_{(j)} + \sum_{i=1}^{k} \sum_{j=h}^{H-1} \left( P \overline{b}_{i,j+1}^{Q} \right) (s_{i,j}, a_{i,j}^{\pi_i(\mathcal{B}_{i,j})}) \\
&= (j) + \underbrace{\sum_{i=1}^{k} \sum_{j=h}^{H-1} \left( \left( P \overline{b}_{i,j+1}^{Q} \right) (s_{i,j}, a_{i,j}^{\pi_i(\mathcal{B}_{i,j})}) - \overline{b}_{i,j+1}^{Q}(s_{i,j+1}) \right)}_{(k)} + \underbrace{\sum_{i=1}^{k} \sum_{j=h}^{H-1} \overline{b}_{i,j+1}^{Q}(s_{i,j+1})}_{(l)}. \tag{43}
\end{aligned}$$

By bounding $\overline{b}_{i,j+1}^{Q}$ with $H^2$ and combining the application of (Weissman et al., 2003, Theorem 2.1) and of a pigeonhole argument, we can upper bound term $(j)$ as:

$$H^2 S \sqrt{AT_k L}.$$

To bound term $(k)$, we first observe that it is a Martingale difference sequence, and as such we can bound it via the event $\mathcal{E}_{\mathrm{az}} \left( \mathcal{F}_{\overline{b}^Q, k, h}, H^2, L \right)$, which holds under $\mathcal{E}$, thus obtaining:

$$(k) \leqslant 2H^2 \sqrt{T_k L}.$$

By applying the definition of $\overline{b}_{i,j+1}^{Q}$ together with a pigeonhole argument, we can bound term $(l)$ as:

$$(l) \leqslant 2900^2 H^3 S^4 A 2^A L^3 G.$$

By applying the bounds of terms $(j)$, $(k)$, and $(l)$ into Eq. (43), we obtain:

$$(h) \leqslant H^2 S \sqrt{AT_k L} + 2H^2 \sqrt{T_k L} + 2900^2 H^3 S^4 A 2^A L^3 G.$$

By applying the bounds of terms $(h)$ and $(i)$ to Eq. (42), applying the definition of $[k]_{\mathrm{typ}}$, and applying the subadditivity of the square root, we get:

$$(c) \leqslant 2\sqrt{3HSAT_kL} + 5800\sqrt{H^3S^5A^22^AL^3G^2}.$$

Finally, we can combine the bounds of terms $(a)$, $(b)$, and $(c)$ into Eq. (39), obtaining the following bound:

$$\sum_{i=1}^{k} \mathbb{I}\{i \in [k]_{\text{typ}}\} \sum_{j=h}^{H-1} b_{i,j}^{Q}(s_{i,j}, a_{i,j}^{\pi_i(\mathcal{B}_{i,j})}) \leqslant \sqrt{16HSAT_kLG} + \sqrt{8HSAL^3G \sum_{i=1}^{k} \sum_{j=h}^{H-1} \widetilde{\Delta}_{i,j+1}^{V}(s_{i,j+1})}$$

$$+ \frac{7}{3}HSALG + 2\sqrt{3HSAT_kL} + 5800\sqrt{H^3S^5A^22^AL^3G^2}$$

$$\leqslant \sqrt{28HSAT_kLG} + \sqrt{8HSALG \sum_{i=1}^{k} \sum_{j=h}^{H-1} \widetilde{\Delta}_{i,j+1}^{V}(s_{i,j+1})}$$

$$+ \frac{7}{3}HSALG + 5800\sqrt{H^3S^5A^22^AL^3G^2}.$$

In a similar manner, observing that events $\mathcal{E}_{\text{az}}\left(\mathcal{F}_{\bar{b}^Q,k,h,s}, H^2, L\right)$, and $\mathcal{E}_{\text{az}}\left(\mathcal{F}_{\bar{b}^Q,k,h,s,a}, H^2, L\right)$ also hold under $\mathcal{E}$, we can derive the bounds for the remaining summations, thus concluding the proof. $\qquad\square$

**Lemma D.11** (Summation over typical episodes of state value function bonus term). *Let $k \in [\![K]\!]$ and $h \in [\![H]\!]$. Let $\pi_k$ be the policy followed during episode $k$. Let the UCB bonus for the state value function be defined as:*

$$b_{k,h}^{V}(s) = \sqrt{\frac{4L\,\mathbb{V}\text{ar}_{\mathcal{B} \sim \widehat{C}_k^{\text{ind}}(\cdot|s)}[\widehat{Q}_{k,h}(s, \pi_{k,h}(s,\mathcal{B}))]}{N_k(s)} + \frac{7HL}{3(N_k(s)-1)}}$$

$$+ \sqrt{\frac{4\mathbb{E}_{\mathcal{B} \sim \widehat{C}_k^{\text{ind}}(\cdot|s)}[\min\{\frac{1350^2 H^3 S^3 A 2^A L^3}{N_{k,h}(s, \pi_{k,h}(s,\mathcal{B}))}, H^2\}]}{N_k(s)}}.$$

*Under the events $\mathcal{E}$ and $\Omega_{k,h}$, the following inequalities hold for every $s \in \mathcal{S}$:*

$$\sum_{i=1}^{k} \mathbb{I}\{i \in [k]_{\text{typ}}\} \sum_{j=h}^{H-1} b_{i,j}^{V}(s_{i,j}) \leqslant \sqrt{45HSAT_kLG} + \frac{7}{3}HSLG + 2700\sqrt{H^3S^5A^22^AL^3G^2}$$

$$+ \sqrt{31H^2S^2ALG^2 \sum_{i=1}^{k} \sum_{j=h}^{H-1} \widetilde{\Delta}_{i,j+1}^{V}(s_{i,j+1})},$$

$$\sum_{i=1}^{k} \mathbb{I}\{i \in [k]_{\text{typ},s}, s_{i,h} = s\} \sum_{j=h}^{H-1} b_{i,j}^{V}(s_{i,j}) \leqslant \sqrt{45H^2SAN_{k,h}(s)LG} + \frac{7}{3}HSLG + 2700\sqrt{H^3S^5A^22^AL^3G^2}$$

$$+ \sqrt{31H^2S^2ALG^2 \sum_{i=1}^{k} \mathbb{I}\{s_{i,h} = s\} \sum_{j=h}^{H-1} \widetilde{\Delta}_{i,j+1}^{V}(s_{i,j+1})},$$

$$\sum_{i=1}^{k} \mathbb{I}\{i \in [k]_{\text{typ},s,a}, s_{i,h} = s, a_{i,h}^{\pi_i(\mathcal{B}_{i,h})} = a\} \sum_{j=h}^{H-1} b_{i,j}^{V}(s_{i,j})$$

$$\leqslant \sqrt{45H^2SAN_{k,h}(s,a)LG} + \frac{7}{3}HSLG + 2700\sqrt{H^3S^5A^22^AL^3G^2}$$

$$+ \sqrt{31H^2S^2ALG^2 \sum_{i=1}^{k} \mathbb{I}\{s_{i,h} = s, a_{i,h}^{\pi_i(\mathcal{B}_{i,h})} = a\} \sum_{j=h}^{H-1} \widetilde{\Delta}_{i,j+1}^{V}(s_{i,j+1})}.$$

*where:*

$$[k]_{\text{typ}} := \left\{ i \in [\![k]\!] : (s_{i,h}, a_{i,h}^{\pi_i(\mathcal{B}_{i,h})}) \in [(s,a)]_k, i \geqslant 11600H^3S^3A2^AL^2G, \forall h \in [\![H]\!] \right\},$$

$$[k]_{\text{typ},s} := \left\{ i \in [\![k]\!] : (s_{i,h}, a_{i,h}^{\pi_i(\mathcal{B}_{i,h})}) \in [(s,a)]_k, N_{k,h}(s) \geqslant 11600H^3S^3A2^AL^2G, \forall h \in [\![H]\!] \right\},$$

$$[k]_{\text{typ},s,a} := \left\{ i \in [\![k]\!] : (s_{i,h}, a_{i,h}^{\pi_i(\mathcal{B}_{i,h})}) \in [(s,a)]_k, N_{k,h}(s,a) \geqslant 11600H^3S^3A2^AL^2G, \forall h \in [\![H]\!] \right\},$$

$$[(s,a)]_k := \left\{ (s,a) \in \mathcal{S} \times \mathcal{A} : N_k(s,a) \geqslant H, N_{k,h}(s) \geqslant H, \forall h \in [\![H]\!] \right\}.$$

*Proof.* We can rewrite the summation as:

$$\sum_{i=1}^{k} \mathbb{I}\{i \in [k]_{\text{typ}}\} \sum_{j=h}^{H-1} b_{i,j}^{V}(s_{i,j}) \leqslant \underbrace{\sum_{i=1}^{k} \mathbb{I}\{i \in [k]_{\text{typ}}\} \sum_{j=h}^{H-1} \sqrt{\frac{4L \operatorname{\mathbb{V}ar}_{\mathcal{B} \sim \widehat{C}_i^{\text{ind}}(\cdot|s_{i,j})}[\widehat{Q}_{i,j}(s_{i,j}, a_{i,j}^{\pi_i(\mathcal{B})})]}{N_i(s_{i,j})}}}_{(a)}$$

$$+ \underbrace{\sum_{i=1}^{k} \mathbb{I}\{i \in [k]_{\text{typ}}\} \sum_{j=h}^{H-1} \frac{7HL}{3(N_i(s_{i,j}) - 1)}}_{(b)}$$

$$+ \underbrace{\sum_{i=1}^{k} \mathbb{I}\{i \in [k]_{\text{typ}}\} \sum_{j=h}^{H-1} \sqrt{\frac{4\mathbb{E}_{\mathcal{B} \sim \widehat{C}_i^{\text{ind}}(\cdot|s_{i,j})} \overline{b}_{i,j}^{V}(s_{i,j}, \mathcal{B})}{N_i(s_{i,j})}}}_{(c)}, \tag{44}$$

where $\overline{b}_{i,j}^{V}(s_{i,j}, \mathcal{B}) = \min\{\frac{1350^2 H^3 S^3 A2^A L^3}{N_{i,j}'(s_{i,j}, a_{i,j}^{\pi_i(\mathcal{B})})}, H^2\}$. First of all, we can bound term $(b)$ by applying a pigeonhole argument as:

$$(b) \leqslant \frac{7}{3}HSLG.$$

We now focus on bounding term $(a)$. By applying Cauchy-Schwarz's inequality, we obtain:

$$(a) \leqslant \sqrt{4L} \sqrt{\underbrace{\sum_{i=1}^{k} \sum_{j=h}^{H-1} \widehat{\mathbb{Q}}_{i,j}(s_{i,j})}_{(d)}} \sqrt{\underbrace{\sum_{i=1}^{k} \mathbb{I}\{i \in [k]_{\text{typ}}\} \sum_{j=h}^{H-1} \frac{1}{N_i(s_{i,j})}}_{(e)}}. \tag{45}$$

Using the pigeonhole argument, we bound term $(e)$ with $SG$. We can rewrite term $(d)$ as follows:

$$(d) = \underbrace{\sum_{i=1}^{k} \sum_{j=h}^{H-1} \mathbb{Q}_j^{\pi_i}(s_{i,j})}_{(f)} + \underbrace{\sum_{i=1}^{k} \sum_{j=h}^{H-1} \left( \widehat{\mathbb{Q}}_{i,j}(s_{i,j}) - \mathbb{Q}_j^{\pi_i}(s_{i,j}) \right)}_{(g)}. \tag{46}$$

We can bound term $(f)$ by using Lemma D.7 as:

$$(f) = \sum_{i=1}^{k} \sum_{j=h}^{H-1} \mathbb{Q}_j^{\pi_i}(s_{i,j})$$

$$\leqslant HT_k + 2\sqrt{H^4 T_k L} + \frac{4}{3} H^3 L,$$

and term $(g)$ by using Lemma D.8 as:

$$(g) = \sum_{i=1}^{k} \sum_{j=h}^{H-1} \left( \widehat{\mathbb{Q}}_{i,j}(s_{i,j}) - \mathbb{Q}_j^{\pi_i}(s_{i,j}) \right)$$

$$\leqslant H^2 \sqrt{S 2^A T_k L} + 7 H^2 S \sqrt{A T_k L} + 4 H^3 S L G$$

$$+ 2H \sum_{i=1}^{k} \sum_{j=h}^{H-1} b_{i,j}^Q(s_{i,j}, a_{i,j}^{\pi_i(\mathcal{B}_{i,j})}) + 2H \sum_{i=1}^{k} \sum_{j=h}^{H-1} \widetilde{\Delta}_{i,j+1}^V(s_{i,j+1})$$

By plugging the bounds of terms $(f)$ and $(g)$ into Eq. (46) we get:

$$(d) \leqslant HT_k + 2\sqrt{H^4 T_k L} + \frac{4}{3} H^3 L + H^2 \sqrt{S 2^A T_k L} + 7 H^2 S \sqrt{A T_k L} + 4 H^3 S L G$$

$$+ 2H \sum_{i=1}^{k} \sum_{j=h}^{H-1} b_{i,j}^Q(s_{i,j}, a_{i,j}^{\pi_i(\mathcal{B}_{i,j})}) + 2H \sum_{i=1}^{k} \sum_{j=h}^{H-1} \widetilde{\Delta}_{i,j+1}^V(s_{i,j+1})$$

$$\leqslant HT_k + 2\sqrt{H^4 T_k L} + \frac{4}{3} H^3 L + H^2 \sqrt{S 2^A T_k L} + 7 H^2 S \sqrt{A T_k L} + 4 H^3 S L G$$

$$+ 2H \sqrt{28 H S A T_k L G} + 2H \sqrt{8 H S A L G \sum_{i=1}^{k} \sum_{j=h}^{H-1} \widetilde{\Delta}_{i,j+1}^V(s_{i,j+1})}$$

$$+ \frac{14}{3} H^2 S A L G + 11600 \sqrt{H^5 S^5 A^2 2^A L^3 G^2} + 2H \sum_{i=1}^{k} \sum_{j=h}^{H-1} \widetilde{\Delta}_{i,j+1}^V(s_{i,j+1}) \tag{47}$$

$$\leqslant 8 H T_k + 2H \sum_{i=1}^{k} \sum_{j=h}^{H-1} \widetilde{\Delta}_{i,j+1}^V(s_{i,j+1}) + \sqrt{32 H^3 S A L G \sum_{i=1}^{k} \sum_{j=h}^{H-1} \widetilde{\Delta}_{i,j+1}^V(s_{i,j+1})}, \tag{48}$$

where Eq. (47) follows by applying Lemma D.10 and Eq. (48) holds under $[k]_{\text{typ}}$. By plugging the bounds of terms $(d)$ and $(e)$ into Eq. (45), rearranging the terms, and applying the subadditivity of the square root, we obtain:

$$(a) \leqslant \sqrt{32 H S T_k L G} + \sqrt{31 H^2 S^2 A L G^2 \sum_{i=1}^{k} \sum_{j=h}^{H-1} \widetilde{\Delta}_{i,j+1}^V(s_{i,j+1})}. \tag{49}$$

The derivation of the bound of term $(c)$ is similar to the derivation in Lemma D.10 from Eq. (42) onward. First we apply Cauchy-Schwarz's inequality, rewriting the term as:

$$(c) \leqslant 2 \sqrt{\underbrace{\sum_{i=1}^{k} \sum_{j=h}^{H-1} \mathbb{E}_{\mathcal{B} \sim \widehat{C}_i^{\mathsf{ind}}(\cdot | s_{i,j})} \overline{b}_{i,j}^V(s_{i,j}, \mathcal{B})}_{(h)}} \sqrt{\underbrace{\sum_{i=1}^{k} \mathbb{I}\{i \in [k]_{\mathrm{typ}}\} \sum_{j=h}^{H-1} \frac{1}{N_i(s_{i,j})}}_{(i)}}. \tag{50}$$

With a pigeonhole argument, we bound term $(i)$ with $SG$. We now rewrite term $(h)$ as:

$$
\begin{aligned}
(h) &= \sum_{i=1}^{k} \sum_{j=h}^{H-1} \left( \widehat{C}_i^{\mathsf{ind}} \overline{b}_{i,j}^V \right)(s_{i,j}) \\
&= \underbrace{\sum_{i=1}^{k} \sum_{j=h}^{H-1} \left( (\widehat{C}_i^{\mathsf{ind}} - C^{\mathsf{ind}}) \overline{b}_{i,j}^V \right)(s_{i,j})}_{(j)} + \sum_{i=1}^{k} \sum_{j=h}^{H-1} \left( C^{\mathsf{ind}} \overline{b}_{i,j}^V \right)(s_{i,j}) \\
&= (j) + \underbrace{\sum_{i=1}^{k} \sum_{j=h}^{H-1} \left( \left( C^{\mathsf{ind}} \overline{b}_{i,j}^V \right)(s_{i,j}) - \overline{b}_{i,j}^V(s_{i,j}) \right)}_{(k)} + \underbrace{\sum_{i=1}^{k} \sum_{j=h}^{H-1} \overline{b}_{i,j}^V(s_{i,j})}_{(l)}.
\end{aligned} \tag{51}
$$

By bounding $\overline{b}_{i,j+1}^V$ with $H^2$ and combining the application of (Weissman et al., 2003, Theorem 2.1) and of a pigeonhole argument, we can upper bound term $(j)$ as:

$$(j) \leqslant H^2 \sqrt{S 2^A T_k L}.$$

To bound term $(k)$, we first observe that it is a Martingale difference sequence, and as such we can bound it via the event $\mathcal{E}_{\mathrm{az}}(\mathcal{F}_{\overline{b}^V, k, h}, H^2, L)$, which holds under $\mathcal{E}$, thus obtaining:

$$(k) \leqslant 2 H^2 \sqrt{T_k L}.$$

By applying the definition of $\overline{b}_{i,j}^V$ together with a pigeonhole argument, we can bound term $(l)$ as:

$$(l) \leqslant 1350^2 H^3 S^4 A^2 2^A L^3 G.$$

By putting together the bounds of terms $(j)$, $(k)$, and $(l)$ into Eq. (51), we get:

$$(h) \leqslant H^2 \sqrt{S 2^A T_k L} + 2 H^2 \sqrt{T_k L} + 1350^2 H^3 S^4 A^2 2^A L^3 G. \tag{52}$$

By applying the bounds of terms $(h)$ and $(i)$ to Eq. (50), applying the definition of $[k]_{\mathrm{typ}}$, and applying the subadditivity of the square root, we get:

$$(c) \leqslant \sqrt{H S A T_k L G} + 2700 \sqrt{H^3 S^5 A^2 2^A L^3 G^2}. \tag{53}$$

Finally, we can combine the bounds of terms $(a)$, $(b)$, and $(c)$ into Eq. (44), obtaining the following bound:

$$\sum_{i=1}^{k} \mathbb{I}\{i \in [k]_{\text{typ}}\} \sum_{j=h}^{H-1} b_{i,j}^{V}(s_{i,j}) \leqslant \sqrt{32HST_kLG} + \sqrt{31H^2S^2ALG^2 \sum_{i=1}^{k}\sum_{j=h}^{H-1} \widetilde{\Delta}_{i,j+1}^{V}(s_{i,j+1})}$$

$$+ \frac{7}{3}HSLG + \sqrt{HSAT_kLG} + 2700\sqrt{H^3S^5A^2 2^A L^3 G^2}$$

$$\leqslant \sqrt{45HSAT_kLG} + \sqrt{31H^2S^2ALG^2 \sum_{i=1}^{k}\sum_{j=h}^{H-1} \widetilde{\Delta}_{i,j+1}^{V}(s_{i,j+1})}$$

$$+ \frac{7}{3}HSLG + 2700\sqrt{H^3S^5A^2 2^A L^3 G^2}.$$

In a similar manner, observing that events $\mathcal{E}_{\text{az}}(\mathcal{F}_{\bar{b}^V,k,h,s}, H^2, L)$ and $\mathcal{E}_{\text{az}}(\mathcal{F}_{\bar{b}^V,k,h,s,a}, H^2, L)$ also hold under $\mathcal{E}$, we can derive the bounds for the remaining summations, thus concluding the proof. $\qquad\square$

**Lemma D.12** (Upper bound of state-action value function estimation error). *Let $k \in [\![K]\!]$ and $h \in [\![H]\!]$. Let $\pi_k$ be the policy followed during episode $k$. Under $\mathcal{E}$ and $\Omega_{k,h}$, the following holds for every $(s,a) \in \mathcal{S} \times \mathcal{A}$:*

$$\widehat{Q}_{k,h}(s,a) - Q_h^*(s,a) \leqslant \min\left\{1350\sqrt{\frac{H^3 S^3 A2^A L^2 G}{N_{k,h}(s,a)}}, H\right\}.$$

*Proof.* We begin the proof by observing that, under $\Omega_{k,h}$, $\widehat{Q}_{k,h}(s,a) \geqslant Q_h^*(s,a)$ for every $s \in \mathcal{S}$ and $a \in \mathcal{A}$. We can rewrite:

$$\widehat{Q}_{k,h}(s,a) - Q_h^*(s,a) = \frac{1}{N_{k,h}(s,a)} \sum_{i=1}^{k} \mathbb{I}\{s_{i,h} = s, a_{i,h}^{\pi_i(\mathcal{B}_{i,h})} = a\}\left(\widehat{Q}_{k,h}(s_{i,h}, a_{i,h}^{\pi_i(\mathcal{B}_{i,h})}) - Q_h^*(s_{i,h}, a_{i,h}^{\pi_i(\mathcal{B}_{i,h})})\right)$$

$$\leqslant \frac{1}{N_{k,h}(s,a)} \sum_{i=1}^{k} \mathbb{I}\{s_{i,h} = s, a_{i,h}^{\pi_i(\mathcal{B}_{i,h})} = a\}\left(\widehat{Q}_{i,h}(s_{i,h}, a_{i,h}^{\pi_i(\mathcal{B}_{i,h})}) - Q_h^{\pi_i}(s_{i,h}, a_{i,h}^{\pi_i(\mathcal{B}_{i,h})})\right) \quad (54)$$

$$= \frac{1}{N_{k,h}(s,a)} \sum_{i=1}^{k} \mathbb{I}\{s_{i,h} = s, a_{i,h}^{\pi_i(\mathcal{B}_{i,h})} = a\}\widetilde{\Delta}_{i,h}^{Q}(s_{i,h}, a_{i,h}^{\pi_i(\mathcal{B}_{i,h})}), \quad (55)$$

where Eq. (54) follows from the fact that $\widehat{Q}_{k,h}$ is monotonically decreasing in $k$ by definition and by observing that $Q_h^* \geqslant Q_h^{\pi_i}$.

Recalling the upper bound of $\sum_{i=1}^{k} \mathbb{I}\{s_{i,h} = s, a_{i,h}^{\pi_i(\mathcal{B}_{i,h})} = a\}\widetilde{\Delta}_{i,h}^{Q}(s_{i,h}, a_{i,h}^{\pi_i(\mathcal{B}_{i,h})})$:

$$\sum_{i=1}^{k} \mathbb{I}\{s_{i,h} = s, a_{i,h}^{\pi_i(\mathcal{B}_{i,h})} = a\}\widetilde{\Delta}_{k,h}^{Q}(s_{i,h}, a_{i,h}^{\pi_i(\mathcal{B}_{i,h})})$$

$$\leqslant e \sum_{i=1}^{k} \mathbb{I}\{s_{i,h} = s, a_{i,h}^{\pi_i(\mathcal{B}_{i,h})} = a\} \sum_{j=h}^{H-1}\left[b_{i,j}^{Q}(s_{i,j}, a_{i,j}^{\pi_i(\mathcal{B}_{i,j})}) + 2b_{i,j+1}^{V}(s_{i,j+1})\right.$$

$$+ \left((\widehat{P}_i - P)V_{j+1}^*\right)(s_{i,j}, a_{i,j}^{\pi_i(\mathcal{B}_{i,j})}) + \frac{8H^2SL}{N_i(s_{i,j}, a_{i,j}^{\pi_i(\mathcal{B}_{i,j})})} + 4H\sqrt{\frac{2^A L}{N_i(s_{i,j+1})}}\right]$$

$$+ 2e\sqrt{H^3 N_{k,h}(s,a)L} + 4e\sqrt{HN_{k,h}(s,a)L} + 4e\sqrt{H^3 N_{k,h}(s,a)L},$$

we can apply two pigeonhole arguments, obtaining the following upper bound:

$$\sum_{i=1}^{k} \mathbb{I}\{s_{i,h} = s, a_{i,h}^{\pi_i(\mathcal{B}_{i,h})} = a\} \tilde{\Delta}_{k,h}^{Q}(s_{i,h}, a_{i,h}^{\pi_i(\mathcal{B}_{i,h})})$$

$$\leqslant e \sum_{i=1}^{k} \mathbb{I}\{s_{i,h} = s, a_{i,h}^{\pi_i(\mathcal{B}_{i,h})} = a\} \sum_{j=h}^{H-1} \left[ b_{i,j}^{Q}(s_{i,j}, a_{i,j}^{\pi_i(\mathcal{B}_{i,j})}) + 2b_{i,j+1}^{V}(s_{i,j+1}) \right.$$

$$\left. + \left( (\hat{P}_i - P)V_{j+1}^* \right)(s_{i,j}, a_{i,j}^{\pi_i(\mathcal{B}_{i,j})}) \right] + 8H^2 S^2 ALG + 4\sqrt{H^3 S 2^A N_{k,h}(s,a)L}$$

$$+ 2e\sqrt{H^3 N_{k,h}(s,a)L} + 4e\sqrt{H N_{k,h}(s,a)L} + 4e\sqrt{H^3 N_{k,h}(s,a)L} \tag{56}$$

$$:= U_{k,h,s,a}.$$

We now bound the summations over episodes and stages of the different terms. By applying Lemma D.11, we can bound the summation over typical episodes of the state value function bonus term as:

$$\sum_{i=1}^{k} \mathbb{I}\{i \in [k]_{\text{typ}}, s_{i,h} = s, a_{i,h}^{\pi_i(\mathcal{B}_{i,h})} = a\} \sum_{j=h}^{H-1} b_{i,j}^{V}(s_{i,j})$$

$$\leqslant \sqrt{45H^2 SAN_{k,h}(s,a)LG} + \frac{7}{3}HSLG + 2700\sqrt{H^3 S^5 A^2 2^A L^3 G^2}$$

$$+ \sqrt{31H^2 S^2 ALG^2 \underbrace{\sum_{i=1}^{k} \mathbb{I}\{s_{i,h} = s, a_{i,h}^{\pi_i(\mathcal{B}_{i,h})} = a\} \sum_{j=h}^{H-1} \tilde{\Delta}_{i,j+1}^{V}(s_{i,j+1})}_{(a)}}. \tag{57}$$

By applying the same procedure as in Eq. (26), we can bound term $(a)$ as:

$$(a) \leqslant \sum_{i=1}^{k} \mathbb{I}\{i \in [k]_{\text{typ}}, s_{i,h} = s, a_{i,h}^{\pi_i(\mathcal{B}_{i,h})} = a\} \sum_{j=h}^{H-1} b_{i,j+1}^{V}(s_{i,j+1})$$

$$+ \sum_{i=1}^{k} \mathbb{I}\{i \in [k]_{\text{typ}}, s_{i,h} = s, a_{i,h}^{\pi_i(\mathcal{B}_{i,h})} = a\} \sum_{j=h}^{H-1} \varepsilon_{i,j+1}^{Q}$$

$$+ 2H \sum_{i=1}^{k} \mathbb{I}\{i \in [k]_{\text{typ}}, s_{i,h} = s, a_{i,h}^{\pi_i(\mathcal{B}_{i,h})} = a\} \sum_{j=h}^{H-1} \sqrt{\frac{2^A L}{N_i(s_{i,i+1})}}$$

$$+ \sum_{i=1}^{k} \mathbb{I}\{i \in [k]_{\text{typ}}, s_{i,h} = s, a_{i,h}^{\pi_i(\mathcal{B}_{i,h})} = a\} \sum_{j=h}^{H-1} \tilde{\Delta}_{i,j+1}^{Q}(s_{i,j+1}, a_{i,j+1}^{\pi_i(\mathcal{B}_{i,j+1})})$$

$$\leqslant \sqrt{45H^2 SAN_{k,h}(s,a)LG} + \frac{7}{3}HSLG + 2700\sqrt{H^3 S^5 A^2 2^A L^3 G^2} + \sqrt{31H^3 S^2 ALG^2}$$

$$+ 2\sqrt{H^3 N_{k,h}(s,a)L} + 2\sqrt{H^3 S 2^A N_{k,h}(s,a)L} + HU_{k,h,s,a}, \tag{58}$$

where Eq. (58) follows from the application of Lemma D.3 and Lemma D.11, the application of a pigeonhole argument, and by bounding $\sum_{i=1}^{k} \mathbb{I}\{s_{i,h} = s, a_{i,h}^{\pi_i(\mathcal{B}_{i,h})} = a\} \sum_{j=h}^{H-1} \tilde{\Delta}_{i,j+1}^{Q}(s_{i,j+1}, a_{i,j+1}^{\pi_i(\mathcal{B}_{i,j+1})})$ with $HU_{k,h,s,a}$.

By applying the bound of term $(a)$ into Eq. (57), applying the condition of $[k]_{\text{typ}}$, and rearranging terms, we obtain:

$$\sum_{i=1}^{k} \mathbb{I}\{i \in [k]_{\text{typ}}, s_{i,h} = s, a_{i,h}^{\pi_i(\mathcal{B}_{i,h})} = a\} \sum_{j=h}^{H-1} b_{i,j}^{V}(s_{i,j})$$

$$\leqslant 4\sqrt{45H^2 SAN_{k,h}(s,a)LG} + \sqrt{31H^3 S^2 ALG^2 U_{k,h,s,a}}$$
$$+ 3014\sqrt{H^4 S^5 A^2 2^A L^3 G^2}.$$

By applying Lemma D.10, we can bound the summation over typical episodes of the state-action value function bonus term as:

$$\sum_{i=1}^{k} \mathbb{I}\{i \in [k]_{\text{typ}}, s_{i,h} = s, a_{i,h}^{\pi_i(\mathcal{B}_{i,h})} = a\} \sum_{j=h}^{H-1} b_{i,j}^{Q}(s_{i,j}, a_{i,j}^{\pi_i(\mathcal{B}_{i,j})})$$

$$\leqslant \sqrt{28H^2 SAN_{k,h}(s,a)LG} + \frac{7}{3}HSAL^2 + 5800\sqrt{H^3 S^5 A^2 2^A L^3 G^2}$$

$$+ \sqrt{8HSALG \underbrace{\sum_{i=1}^{k} \mathbb{I}\{, s_{i,h} = s, a_{i,h}^{\pi_i(\mathcal{B}_{i,h})} = a\}\widetilde{\Delta}_{i,j+1}^{V}(s_{i,j+1})}_{(a)}}. \quad (59)$$

By applying the bound of term $(a)$, as computed in Eq. (58), and plugging it into Eq. (59), we get:

$$\sum_{i=1}^{k} \mathbb{I}\{i \in [k]_{\text{typ}}, s_{i,h} = s, a_{i,h}^{\pi_i(\mathcal{B}_{i,h})} = a\} \sum_{j=h}^{H-1} b_{i,j}^{Q}(s_{i,j}, a_{i,j}^{\pi_i(\mathcal{B}_{i,j})})$$

$$\leqslant 4\sqrt{28H^2 SAN_{k,h}(s,a)LG} + \sqrt{8H^2 SALGU_{k,h,s,a}}$$
$$+ 5961\sqrt{H^3 S^5 A^2 2^A L^3 G^2}.$$

By applying Lemma D.9 we can bound the summation over typical episodes of the state-action wise model errors as:

$$\sum_{i=1}^{k} \mathbb{I}\{i \in [k]_{\text{typ}}, s_{i,h} = s, a_{i,h}^{\pi_i(\mathcal{B}_{i,h})} = a\} \sum_{j=h}^{H-1} \left((\widehat{P}_i - P)V_{j+1}^{*}\right)(s_{i,j}, a_{i,j}^{\pi_i(\mathcal{B}_{i,j})})$$

$$\leqslant \sqrt{6H^2 SAN_{k,h}(s,a)LG} + \frac{2}{3}HSALG + 2\sqrt{H^2 SALGU_{k,h,s,a}}.$$

By applying two pigeonhole arguments, combining the bounds, and accounting for the regret of non-typical episodes, we can then upper bound Eq. (56) as:

$$U_{k,h,s,a} \leqslant 92e\sqrt{H^3 SA2^A N_{k,h}(s,a)LG} + 16e\sqrt{H^3 S^2 ALG^2 U_{k,h,s,a}}$$
$$+ 11996e\sqrt{H^4 S^5 A^2 2^A L^3 G^2} + 11600H^3 S^3 A2^A L^3.$$

Letting:

$$\alpha = e\left[92\sqrt{H^3 SA2^A N_{k,h}(s,a)LG} + 11996\sqrt{H^4 S^5 A^2 2^A L^3 G^2}\right] + 11600H^3 S^3 A2^A L^2 G,$$

$$\beta = 16e\sqrt{H^3S^2ALG^2},$$

we can solve for $U_{k,h,s,a}$ as $U_{k,h,s,a} \leqslant 2\alpha + \beta^2$, and obtain the following bound:

$$U_{k,h,s,a} \leqslant 184e\sqrt{H^3SA2^A N_{k,h}(s,a)LG} + 23992e\sqrt{H^4S^5A^22^AL^3G^2} + 23200H^3S^3A2^AL^2G + 256e^2H^3S^2ALG^2$$

$$\leqslant 1350\sqrt{H^3S^3A2^A N_{k,h}(s,a)L^2G}, \tag{60}$$

where Eq. (60) follows from the application of the $[k]_{\text{typ}}$ condition. Plugging this result into Eq. (55), and observing that the error cannot be greater than $H$, we get the following bound to the estimation error of the state-action value function due to the optimistic approach:

$$\widehat{Q}_{k,h}(s,a) - Q_h^*(s,a) \leqslant \min\left\{1350\sqrt{\frac{H^3S^3A2^AL^2G}{N_{k,h}(s,a)}}, H\right\},$$

thus concluding the proof. $\qquad\square$

**Lemma D.13** (Upper bound of state value function estimation error). *Let $k \in [\![K]\!]$ and $h \in [\![H]\!]$. Let $\pi_k$ be the policy followed during episode $k$. Under $\mathcal{E}$ and $\Omega_{k,h}$, the following holds for every $(s,a) \in \mathcal{S} \times \mathcal{A}$:*

$$\widehat{V}_{k,h}(s) - V_h^*(s) \leqslant \min\left\{2900\sqrt{\frac{H^3S^3A2^AL^2G}{N_{k,h}(s)}}, H\right\}.$$

*Proof.* We begin the proof by observing that, under $\Omega_{k,h}$, $\widehat{Q}_{k,h}(s,a) \geqslant Q_h^*(s,a)$ for every $s \in \mathcal{S}$ and $a \in \mathcal{A}$. We can rewrite:

$$\widehat{V}_{k,h}(s) - V_h^*(s) = \frac{1}{N_{k,h}(s)} \sum_{i=1}^{k} \mathbb{I}\{s_{i,h} = s\}\left(\widehat{V}_{k,h}(s_{i,h}) - V_h^*(s_{i,h})\right)$$

$$\leqslant \frac{1}{N_{k,h}(s)} \sum_{i=1}^{k} \mathbb{I}\{s_{i,h} = s\}\left(\widehat{V}_{i,h}(s_{i,h}) - V_h^{\pi_i}(s_{i,h})\right) \tag{61}$$

$$= \frac{1}{N_{k,h}(s)} \sum_{i=1}^{k} \mathbb{I}\{s_{i,h} = s\}\widetilde{\Delta}_{i,h}^V(s_{i,h}), \tag{62}$$

where Eq. (61) follows from the fact that $\widehat{V}_{k,h}$ is monotonically decreasing in $k$ by definition and by observing that $V_h^* \geqslant V_h^{\pi_i}$.

Recalling the upper bound of $\sum_{i=1}^{k} \mathbb{I}\{s_{i,h} = s\}\widetilde{\Delta}_{i,h}^V(s_{i,h})$:

$$\sum_{i=1}^{k} \mathbb{I}\{s_{i,h} = s\}\widetilde{\Delta}_{i,j}^V(s_{i,j}) \leqslant e^2 \sum_{i=1}^{k} \sum_{j=h}^{H-1} \left[ b_{i,j}^V(s_{i,j}) + 2b_{i,j}^Q(s_{i,j}, a_{i,j}^{\pi_i(\mathcal{B}_{i,j})}) + 2\left((\widehat{P}_i - P)V_{j+1}^*\right)(s_{i,j}, a_{i,j}^{\pi_i(\mathcal{B}_{i,j})}) \right.$$

$$+ \mathbb{E}_{\mathcal{B}\sim C^{\text{ind}}(\cdot|s_{i,j})}\left[\frac{8H^2SL}{3N_i(s_{i,j}, a_{i,j}^{\pi_i(B)})}\right] + \frac{8HSL}{3N_i(s_{i,j}, a_{i,j}^{\pi_i(\mathcal{B}_{i,j})})} + \frac{2H^22^AL}{N_i(s_{i,j})} + \frac{2H2^AL}{3N_i(s_{i,j})}$$

$$\left. + \sqrt{\frac{2H^2L}{N_i(s_{i,j})}}\right] + 4e^2\sqrt{H^3N_{k,h}(s)L} + 6e^2\sqrt{HN_{k,h}(s)L} + 4e^2\sqrt{HN_{k,h}(s)L},$$

by applying several pigeonhole arguments, we can derive the following upper bound:

$$\sum_{i=1}^{k} \mathbb{I}\{s_{i,h} = s\}\widetilde{\Delta}_{i,j}^{V}(s_{i,j}) \leqslant e^2 \sum_{i=1}^{k} \sum_{j=h}^{H-1} \left[ b_{i,j}^{V}(s_{i,j}) + 2b_{i,j}^{Q}(s_{i,j}, a_{i,j}^{\pi_i(\mathcal{B}_{i,j})}) + 2\left( (\widehat{P}_i - P)V_{j+1}^* \right)(s_{i,j}, a_{i,j}^{\pi_i(\mathcal{B}_{i,j})}) \right]$$

$$+ 8e^2 H^2 S^2 2^A LG + e^2 \sqrt{2H^3 SN_{k,h}(s)L} + 4e^2 \sqrt{H^3 N_{k,h}(s)L} + 10e^2 \sqrt{HN_{k,h}(s)L}$$

$$\tag{63}$$

$$:= U_{k,h,s},$$

where Eq. (63) is obtained observing that, when applying the pigeonhole argument over a summation in which the argument depends on the expectation over $\mathcal{B}$, the worst-case allocation of summation terms is the one in which all actions are available. We now bound the summations over the typical episodes of the different terms.

By applying Lemma D.10, we can bound the summation over typical episodes of the state-action value function bonus term as:

$$\sum_{i=1}^{k} \mathbb{I}\{i \in [k]_{\text{typ}}, s_{i,h} = s\} \sum_{j=h}^{H-1} b_{i,j}^{Q}(s_{i,j}, a_{i,j}^{\pi_i(\mathcal{B}_{i,j})}) \leqslant \sqrt{28H^2 SAN_{k,h}(s)LG} + \sqrt{8H^2 SALGU_{k,h,s}}$$

$$+ \frac{7}{3}HSALG + 5800\sqrt{H^3 S^5 A^2 2^A L^3 G^2}.$$

By applying Lemma D.11, we can bound the summation over typical episodes of the state value function bonus term as:

$$\sum_{i=1}^{k} \mathbb{I}\{i \in [k]_{\text{typ}}, s_{i,h} = s\} \sum_{j=h}^{H-1} b_{i,j}^{V}(s_{i,j}) \leqslant \sqrt{40H^2 SAN_{k,h}(s)LG} + \sqrt{31H^3 S^2 ALG^2 U_{k,h,s}}$$

$$+ \frac{7}{3}HSLG + 2700\sqrt{H^3 S^5 A^2 2^A L^3 G^2}.$$

By applying Lemma D.9 we can bound the summation over typical episodes of the state-action wise model errors as:

$$\sum_{i=1}^{k} \mathbb{I}\{i \in [k]_{\text{typ}}, s_{i,h} = s\} \sum_{j=h}^{H-1} \left( (\widehat{P}_i - P)V_{j+1}^* \right)(s_{i,j}, a_{i,j}^{\pi_i(\mathcal{B}_{i,j})})$$

$$\leqslant \sqrt{6H^2 SAN_{k,h}(s)LG} + \frac{2}{3}HSALG + 2\sqrt{H^2 SALGU_{k,h,s}}.$$

By combining the bounds into Eq. (63), we get:

$$U_{k,h,s} \leqslant 38e^2 \sqrt{H^3 SA2^A N_{k,h}(s)LG} + 16e^2 \sqrt{H^3 S^2 ALG^2 U_{k,h,s}}$$

$$+ 14317e^2 \sqrt{H^3 S^5 A^2 2^A L^3 G^2} + 11600 H^3 S^3 A 2^A L^3 \tag{64}$$

Letting:

$$\alpha = 38e^2 \sqrt{H^3 SA2^A N_{k,h}(s)LG} + 14317e^2 \sqrt{H^3 S^5 A^2 2^A L^3 G^2} + 11600 H^3 S^3 A 2^A L^2 G,$$

$$\beta = 16e^2 \sqrt{H^3 S^2 ALG^2}, \tag{65}$$

we can solve for $U_{k,h,s}$ as $U_{k,h,s} \leqslant 2\alpha + \beta^2$, and obtain the following bound:

$$U_{k,h,s} \leqslant 76e^2\sqrt{H^3 S A 2^A N_{k,h}(s) L G} + 28634e^2\sqrt{H^3 S^5 A^2 2^A L^3 G^2} + 23200 H^3 S^3 A 2^A L^2 G + 256 e^4 H^3 S^2 A L G^2$$

$$\leqslant 2900\sqrt{H^3 S^3 A 2^A N_{k,h}(s) L^2 G}, \tag{66}$$

where Eq. (66) holds if $N_{k,h}(s) \geqslant \sqrt{11600 H^3 S^3 A 2^A L^2 G}$. Plugging this result into Eq. (62), and observing that the error cannot be greater than $H$, we get the following bound to the estimation error of the state-action value function due to the optimistic approach:

$$\widehat{V}_{k,h}(s) - V_h^*(s) \leqslant \min\left\{2900\sqrt{\frac{H^3 S^3 A 2^A L^2 G}{N_{k,h}(s)}}, H\right\},$$

thus concluding the proof.

$\square$

**Lemma D.14** (Optimism). *Let the optimistic bonuses be defined as:*

$$b_{k,h}^Q(s,a) = \sqrt{\frac{4L\,\mathbb{V}\mathrm{ar}_{s'\sim\widehat{P}_k(\cdot|s,a)}[\widehat{V}_{k,h+1}(s')]}{N_k(s,a)}} + \frac{7HL}{3(N_k(s,a)-1)}$$

$$+ \sqrt{\frac{4\mathbb{E}_{s'\sim\widehat{P}_k(\cdot|s,a)}[\min\{\frac{2900^2 H^3 S^3 A 2^A L^3}{N'_{k,h+1}(s')}, H^2\}]}{N_k(s,a)}},$$

$$b_{k,h}^V(s) = \sqrt{\frac{4L\,\mathbb{V}\mathrm{ar}_{\mathcal{B}\sim\widehat{C}_k^{\mathrm{ind}}(\cdot|s)}[\widehat{Q}_{k,h}(s,\pi_{k,h}(s,\mathcal{B}))]}{N_k(s)}} + \frac{7HL}{3(N_k(s)-1)}$$

$$+ \sqrt{\frac{4\mathbb{E}_{\mathcal{B}\sim\widehat{C}_k^{\mathrm{ind}}(\cdot|s)}[\min\{\frac{1350^2 H^3 S^3 A 2^A L^3}{N_{k,h}(s,\pi_{k,h}(s,\mathcal{B}))}, H^2\}]}{N_k(s)}}.$$

*Then, under event $\mathcal{E}$, the following set of events hold:*

$$\Omega_{k,h} := \left\{\widehat{V}_{i,j}(s) \geqslant V_j^*(s) \wedge \widehat{Q}_{i,j}(s,a) \geqslant Q_j^*(s,a), \forall(i,j) \in [k,h]_{\mathrm{hist}}, s \in \mathcal{S}, a \in \mathcal{A}\right\},$$

*for $k \in [\![K]\!]$ and $h \in [\![H]\!]$, where:*

$$[k,h]_{\mathrm{hist}} := \{(i,j) : i \in [\![K]\!], j \in [\![H]\!], (i < k) \vee (i = k, j \geqslant h)\}.$$

*Proof.* We demonstrate this result by induction. We begin by observing that $\widehat{V}_{k,H+1}(s) = V_{H+1}^*(s) = 0$ and $\widehat{Q}_{k,H+1}(s,a) = Q_{H+1}^*(s,a) = 0$ for every $k \in [\![K]\!]$, $s \in \mathcal{S}$, and $a \in \mathcal{A}$. To prove the induction, we need to prove that, if $\Omega_{k,h+1}$ holds, then also $\Omega_{k,h}$ holds. We prove this result for a generic $k \in [\![K]\!]$, and we observe that we can then apply this procedure recursively for all values of $k$, starting from $k = 1$.

We begin by demonstrating that $\widehat{Q}_{k,h}(s,a) \geqslant Q_h^*(s,a)$. Let us recall the definition of $\widehat{Q}_{k,h}(s,a)$:

$$\widehat{Q}_{k,h}(s,a) = R(s,a) + b_{k,h}^Q(s,a) + \left(\widehat{P}_k \widehat{V}_{k,h+1}\right)(s,a),$$

observing that if $\widehat{Q}_{k,h}(s,a) \geqslant H - h$, the optimism holds trivially. We can write:

$$
\begin{aligned}
\widehat{Q}_{k,h}(s,a) - Q_h^*(s,a) &= b_{k,h}^Q(s,a) + \left(\widehat{P}_k \widehat{V}_{k,h+1}\right)(s,a) - \left(P V_{h+1}^*\right)(s,a) \\
&= b_{k,h}^Q(s,a) + \left(\widehat{P}_k (\widehat{V}_{k,h+1} - V_{h+1}^*)\right)(s,a) + \left((\widehat{P}_k - P)V_{h+1}^*\right)(s,a) \\
&\geqslant b_{k,h}^Q(s,a) + \left((\widehat{P}_k - P)V_{h+1}^*\right)(s,a)
\end{aligned}
\tag{67}
$$

where Eq. (67) follows from the induction assumption. Under event $\mathcal{E}$, we can apply the empirical Bernstein inequality (Maurer & Pontil, 2009):

$$
\left|\left((\widehat{P}_k - P)V_{h+1}^*\right)(s,a)\right| \leqslant \sqrt{\frac{2\widehat{\mathbb{V}}_{k,h+1}^*(s,a)L}{N_k(s,a)}} + \frac{7HL}{3(N_k(s,a) - 1)},
$$

where $\widehat{\mathbb{V}}_{k,h+1}^*(s,a) := \mathbb{V}\mathrm{ar}_{s' \sim \widehat{P}_k(\cdot|s,a)}[V_{h+1}^*(s')]$. As such, we obtain:

$$
\begin{aligned}
\widehat{Q}_{k,h}(s,a) - Q_h^*(s,a) &\geqslant b_{k,h}^Q(s,a) - \sqrt{\frac{2\widehat{\mathbb{V}}_{k,h+1}^*(s,a)L}{N_k(s,a)}} - \frac{7HL}{3(N_k(s,a) - 1)} \\
&= \underbrace{\sqrt{\frac{4\widehat{\mathbb{V}}_{k,h+1}(s,a)L}{N_k(s,a)}} - \sqrt{\frac{2\widehat{\mathbb{V}}_{k,h+1}^*(s,a)L}{N_k(s,a)}}}_{(a)} \\
&\quad + \sqrt{\frac{4\sum_{s'\in\mathcal{S}}\widehat{P}_k(s'|s,a)\min\left\{\frac{2900^2 H^3 S^3 A^2 2^A L^3}{N'_{k,h+1}(s')}, H^2\right\}}{N_k(s,a)}}.
\end{aligned}
\tag{68}
$$

Observing that:

$$
(a) \geqslant \begin{cases}
-\sqrt{\frac{2\widehat{\mathbb{V}}_{k,h+1}^*(s,a) - 4\widehat{\mathbb{V}}_{k,h+1}(s,a)}{N_k(s,a)}} & \text{if } \widehat{\mathbb{V}}_{k,h+1}(s,a) \leqslant \widehat{\mathbb{V}}_{k,h+1}^*(s,a), \\
0 & \text{otherwise,}
\end{cases}
$$

we now bound $\widehat{\mathbb{V}}_{k,h+1}^*$ in terms of $\widehat{\mathbb{V}}_{k,h+1}$ from above. Observing that:

$$
\begin{aligned}
\mathbb{V}\mathrm{ar}[X] &= \mathbb{E}\left[X - \mathbb{E}[X]\right]^2 \\
&= \mathbb{E}\left[X \pm Y - \mathbb{E}[X] \pm \mathbb{E}[Y]\right]^2 \\
&= \mathbb{E}\left[(X - Y) - \mathbb{E}[X - Y] + Y - \mathbb{E}[Y]\right]^2 \\
&\leqslant 2\mathbb{E}\left[(X - Y) - \mathbb{E}[X - Y]\right]^2 + 2\mathbb{E}\left[Y - \mathbb{E}[Y]\right]^2 \\
&= 2\mathbb{V}\mathrm{ar}[X - Y] + 2\mathbb{V}\mathrm{ar}[Y],
\end{aligned}
\tag{69}
$$

we can then write:

$$\widehat{\mathbb{V}}^*_{k,h+1}(s,a) \leqslant 2\widehat{\mathbb{V}}_{k,h+1}(s,a) + 2 \operatorname*{Var}_{y\sim\widehat{P}_k(\cdot|s,a)}[V^*_{h+1}(s') - \widehat{V}_{k,h+1}(s')]$$

$$\leqslant 2\widehat{\mathbb{V}}_{k,h+1}(s,a) + 2 \sum_{s'\in\mathcal{S}} \widehat{P}_k(s'|s,a)\left(\widehat{V}_{k,h+1}(s') - V^*_{h+1}(s')\right)^2.$$

By combining this bound with the result of Lemma D.13, we obtain the following bound on term $(a)$:

$$(a) \geqslant \begin{cases} -\sqrt{\dfrac{4\sum_{s'\in\mathcal{S}}\widehat{P}_k(s'|s,a)\min\left\{\frac{2900^2 H^3 S^3 A^2 2^A L^3}{N'_{k,h+1}(s')}, H^2\right\}}{N_k(s,a)}} & \text{if } \widehat{\mathbb{V}}_{k,h+1}(s,a) \leqslant \widehat{\mathbb{V}}^*_{k,h+1}(s,a), \\ 0 & \text{otherwise.} \end{cases}$$

By plugging this result into Eq. (68), we obtain that $\widehat{Q}_{k,h}(s,a) - Q^*_h(s,a) \geqslant 0$.

We now demonstrate that $\widehat{V}_{k,h}(s) \geqslant V^*_h(s)$. Let us recall the definition of $\widehat{V}_{k,h}(s)$:

$$\widehat{V}_{k,h}(s) = \min\{\widehat{V}_{k-1,h}(s), H, b^V_{k,h}(s) + \mathbb{E}_{\mathcal{B}\sim\widehat{C}^{\mathsf{ind}}_k(\cdot|s)}[\widehat{Q}_{k,h}(s, a^{\pi_k(\mathcal{B})}_{k,h})]\}.$$

Again, observe that, if $\widehat{V}_{k,h}(s) = H$, the optimism holds trivially. Moreover, if $\widehat{V}_{k,h}(s) = \widehat{V}k-1,h(s)$, the optimism holds trivially under $\Omega_{k,h}$. As such, we only need to demonstrate the case in which $\widehat{V}_{k,h}(s) = b^V_{k,h}(s) + \mathbb{E}_{\mathcal{B}\sim\widehat{C}^{\mathsf{ind}}_k(\cdot|s)}[\widehat{Q}_{k,h}(s, a^{\pi_k(\mathcal{B})}_{k,h})]$. We can write:

$$\widehat{V}_{k,h}(s) - V^*_h(s) = b^V_{k,h}(s) + \left(\widehat{C}^{\mathsf{ind}}_k \widehat{Q}_{k,h}\right)(s) - \left(C^{\mathsf{ind}} Q^*_h\right)(s)$$

$$= b^V_{k,h}(s) + \left(\widehat{C}^{\mathsf{ind}}_k(\widehat{Q}_{k,h} - Q^*_h)\right)(s) + \left((\widehat{C}^{\mathsf{ind}}_k - C^{\mathsf{ind}})Q^*_h\right)(s)$$

$$\geqslant b^V_{k,h}(s) + \sum_{\mathcal{B}\in\mathcal{P}(\mathcal{A})} \widehat{C}^{\mathsf{ind}}_k(\mathcal{B}|s)\left(\widehat{Q}_{k,h}(s, \pi^*_h(s,\mathcal{B})) - Q^*_h(s, \pi^*_h(s,\mathcal{B}))\right)$$

$$+ \left((\widehat{C}^{\mathsf{ind}}_k - C^{\mathsf{ind}})Q^*_h\right)(s) \tag{70}$$

$$\geqslant b^V_{k,h}(s) + \left((\widehat{C}^{\mathsf{ind}}_k - C^{\mathsf{ind}})Q^*_h\right)(s), \tag{71}$$

where Eq. (70) derives by observing that $\pi_k$ is the greedy policy w.r.t. $\widehat{V}_{k,h}$, and Eq. (71) follows from the optimism over the state-action value function we just demonstrated.

Under event $\mathcal{E}$, we can apply the empirical Bernstein inequality:

$$\left|\left((\widehat{C}^{\mathsf{ind}}_k - C^{\mathsf{ind}})Q^*_h\right)(s)\right| \leqslant \sqrt{\frac{2\widehat{\mathbb{Q}}^*_{k,h}(s)L}{N_k(s)}} + \frac{7HL}{3(N_k(s)-1)}, \tag{72}$$

where $\widehat{\mathbb{Q}}^*_{k,h}(s) := \mathbb{Var}_{\mathcal{B}\sim\widehat{C}^{\mathsf{ind}}_k(\cdot|s)}[Q^*_h(s, a^{\pi^*(\mathcal{B})}_{k,h})]$. As such, we obtain:

$$\widehat{V}_{k,h}(s) - V^*_h(s) \geqslant b^V_{k,h}(s) - \sqrt{\frac{2\widehat{\mathbb{Q}}^*_{k,h}(s)L}{N_k(s)}} - \frac{7HL}{3(N_k(s)-1)}$$

$$= \underbrace{\sqrt{\frac{4\widehat{\mathbb{Q}}_{k,h}(s)L}{N_k(s)}} - \sqrt{\frac{2\widehat{\mathbb{Q}}^*_{k,h}(s)L}{N_k(s)}}}_{(b)}$$

$$+ \sqrt{\frac{4\sum_{\mathcal{B}\in\mathcal{P}(\mathcal{A})} \widehat{C}^{\mathsf{ind}}_k(\mathcal{B}|s) \min\left\{\frac{1350^2 H^3 S^3 A^2 2^A L^3}{N_{k,h}(s,a^{\pi_k(\mathcal{B})}_{k,h})}, H^2\right\}}{N_k(s)}}. \tag{73}$$

Observing that:

$$(b) \geq \begin{cases} -\sqrt{\frac{2\widehat{\mathbb{Q}}^*_{k,h}(s) - 4\widehat{\mathbb{Q}}_{k,h}(s)}{N_k(s)}} & \text{if } \widehat{\mathbb{Q}}_{k,h}(s) \leq \widehat{\mathbb{Q}}^*_{k,h}(s), \\ 0 & \text{otherwise,} \end{cases}$$

we now bound $\widehat{\mathbb{Q}}^*_{k,h}$ in terms of $\widehat{\mathbb{Q}}_{k,h}$ from above. By applying the result of Eq. (69), we get that:

$$\widehat{\mathbb{Q}}^*_{k,h}(s) \leq 2\widehat{\mathbb{Q}}_{k,h}(s) + 2 \operatorname*{Var}_{\mathcal{B}\sim\widehat{C}^{\mathsf{ind}}_k(\cdot|s)} [Q^*_h(s, a^{\pi_k(\mathcal{B})}_{k,h}) - \widehat{Q}_{k,h}(s, a^{\pi_k(\mathcal{B})}_{k,h})]$$

$$\leq 2\widehat{\mathbb{Q}}_{k,h}(s) + 2 \sum_{\mathcal{B}\in\mathcal{P}(\mathcal{A})} \widehat{C}^{\mathsf{ind}}_k(\mathcal{B}|s) \left(Q^*_h(s, a^{\pi_k(\mathcal{B})}_{k,h}) - \widehat{Q}_{k,h}(s, a^{\pi_k(\mathcal{B})}_{k,h})\right)^2. \tag{74}$$

By combining this bound with the result of Lemma D.12, we obtain the following bound on term $(b)$:

$$(b) \geq \begin{cases} -\sqrt{\frac{4\sum_{\mathcal{B}\in\mathcal{P}(\mathcal{A})} \widehat{C}^{\mathsf{ind}}_k(\mathcal{B}|s) \min\left\{\frac{1350^2 H^3 S^3 A^2 2^A L^3}{N_{k,h}(s,a^{\pi_k(\mathcal{B})}_{k,h})}, H^2\right\}}{N_k(s)}} & \text{if } \widehat{\mathbb{Q}}_{k,h}(s) \leq \widehat{\mathbb{Q}}^*_{k,h}(s), \\ 0 & \text{otherwise.} \end{cases}$$

By plugging this result into Eq. (73), we finally obtain that $\widehat{V}_{k,h}(s) \geq V^*_h(s)$, thus demonstrating optimism. □

**Theorem 5.2** (Regret Upper Bound S-UCBVI with independent availability and per-stage disclosure). *For any $\delta \in (0,1)$, with probability $1 - \delta$, the per-stage disclosure regret of* S-UCBVI *on any SleMDP with per-stage disclosure independent action availabilities is bounded by:*

$$R_{\mathsf{PS}}(\text{S-UCBVI}, T) \leq 512 H\sqrt{SATLG}$$
$$+ 498^2 H^6 S^3 A 2^A L^2 G,$$

*where $L = \log(80 H S^2 A 2^A T/\delta)$ and $G = \log(HSAT)$. In particular, for $T \geq \Omega(H^{10} S^5 A^4 2^{2A})$ and selecting $\delta = 2^A/T$, we have:*

$$\mathbb{E}[R_{\mathsf{PS}}(\text{S-UCBVI}, T)] \leq \widetilde{\mathcal{O}}\left(H\sqrt{SAT}\right).$$

*Proof.* This proof adapts the result of (Azar et al., 2017, Theorem 2) to the Sleeping MDP setting in the case of i.i.d. action set availability and per-stage disclosure. We start by considering a Sleeping MDP in which the transition probabilities are stage-independent, as the generalization is straightforward and we consider it at the end of the proof.

Let events $\mathcal{E}$ and $\Omega_{k,h}$ hold. Under these events, Lemma D.14 holds.

As such, we define the regret suffered by an algorithm $\mathfrak{A}$ after $T$ time steps as:

$$R^{\mathsf{SD}}_T(\mathfrak{A}) = \sum_{i=1}^{K} (V^*_1(s_{i,1}) - V^{\pi_i}_1(s_{i,1})) := \Delta^V_{i,1}(s_{i,1}).$$

We can define a pseudo-regret suffered by an algorithm $\mathfrak{A}$ as:

$$\tilde{R}_T^{\mathrm{SD}}(\mathfrak{A}) = \sum_{i=1}^{K} \left( \widehat{V}_{i,1}(s_{i,1}) - \widehat{V}_1^{\pi_i}(s_{i,1}) \right) := \widetilde{\Delta}_{i,1}^V(s_{i,1}).$$

By applying Lemma D.1, we first observe that $\sum_{i=1}^{k} \Delta_{i,1}^V(s_{i,1}) \leqslant \sum_{i=1}^{k} \widetilde{\Delta}_{i,1}^V(s_{i,1})$, and we decompose the pseudo-regret as:

$$\sum_{i=1}^{K} \widetilde{\Delta}_{i,h}^V(s_{i,h}) \leqslant e^2 \sum_{i=1}^{K} \sum_{j=1}^{H-1} \left[ b_{i,j}^V(s_{i,j}) + \mathbb{E}_{\mathcal{B} \sim C^{\mathrm{ind}}(\cdot|s_{i,j})}[b_{i,j}^Q(s_{i,j}, a_{i,j}^{\pi_i(\mathcal{B})})] + \frac{1}{H} b_{i,j}^Q(s_{i,j}, a_{i,j}^{\pi_i(\mathcal{B}_{i,j})}) \right.$$

$$+ \mathbb{E}_{\mathcal{B} \sim C^{\mathrm{ind}}(\cdot|s_{i,j})} \left[ \varepsilon_{i,j,\mathcal{B}}^V \right] + \frac{1}{H} \varepsilon_{i,j}^V + \mathbb{E}_{\mathcal{B} \sim C^{\mathrm{ind}}(\cdot|s_{i,j})} \left[ \sqrt{2L} \bar{\varepsilon}_{i,j,\mathcal{B}}^V \right] + \frac{\sqrt{2L}}{H} \bar{\varepsilon}_{i,j}^V + \sqrt{2L} \bar{\varepsilon}_{i,j}^Q$$

$$+ \mathbb{E}_{\mathcal{B} \sim C^{\mathrm{ind}}(\cdot|s_{i,j})} \left[ \left( (\widehat{P}_i - P) V_{j+1}^* \right) (s_{i,j}, a_{i,j}^{\pi_i(\mathcal{B})}) \right] + \frac{1}{H} \left( (\widehat{P}_i - P) V_{j+1}^* \right) (s_{i,j}, a_{i,j}^{\pi_i(\mathcal{B}_{i,j})})$$

$$+ \mathbb{E}_{\mathcal{B} \sim C^{\mathrm{ind}}(\cdot|s_{i,j})} \left[ \frac{8H^2 SL}{3N_i(s_{i,j}, a_{i,j}^{\pi_i(B)})} \right] + \frac{8HSL}{3N_i(s_{i,j}, a_{i,j}^{\pi_i(\mathcal{B}_{i,j})})}$$

$$\left. + \frac{2H^2 2^A L}{N_i(s_{i,j})} + \frac{2H 2^A L}{3N_i(s_{i,j})} + \sqrt{\frac{2\mathbb{Q}_j^{\pi_i}(s_{i,j})L}{N_i(s_{i,j})}} + \frac{2HL}{3N_i(s_{i,j})} \right].$$

By applying several pigeonhole arguments, we obtain:

$$\sum_{i=1}^{K} \widetilde{\Delta}_{i,h}^V(s_{i,h}) \leqslant e^2 \sum_{i=1}^{K} \sum_{j=1}^{H-1} \left[ b_{i,j}^V(s_{i,j}) + \mathbb{E}_{\mathcal{B} \sim C^{\mathrm{ind}}(\cdot|s_{i,j})}[b_{i,j}^Q(s_{i,j}, a_{i,j}^{\pi_i(\mathcal{B})})] + \frac{1}{H} b_{i,j}^Q(s_{i,j}, a_{i,j}^{\pi_i(\mathcal{B}_{i,j})}) \right.$$

$$+ \mathbb{E}_{\mathcal{B} \sim C^{\mathrm{ind}}(\cdot|s_{i,j})} \left[ \varepsilon_{i,j,\mathcal{B}}^V \right] + \frac{1}{H} \varepsilon_{i,j}^V + \mathbb{E}_{\mathcal{B} \sim C^{\mathrm{ind}}(\cdot|s_{i,j})} \left[ \sqrt{2L} \bar{\varepsilon}_{i,j,\mathcal{B}}^V \right] + \frac{\sqrt{2L}}{H} \bar{\varepsilon}_{i,j}^V + \sqrt{2L} \bar{\varepsilon}_{i,j}^Q$$

$$+ \mathbb{E}_{\mathcal{B} \sim C^{\mathrm{ind}}(\cdot|s_{i,j})} \left[ \left( (\widehat{P}_i - P) V_{j+1}^* \right) (s_{i,j}, a_{i,j}^{\pi_i(\mathcal{B})}) \right] + \frac{1}{H} \left( (\widehat{P}_i - P) V_{j+1}^* \right) (s_{i,j}, a_{i,j}^{\pi_i(\mathcal{B}_{i,j})})$$

$$\left. + \underbrace{\sqrt{\frac{2\mathbb{Q}_j^{\pi_i}(s_{i,j})L}{N_i(s_{i,j})}}}_{(a)} \right] + 9e^2 H^2 S^2 A 2^A LG \tag{75}$$

$$:= U_{K,1}.$$

We can bound the summation of term $(a)$ over typical episodes as follows:

$$\sum_{i=1}^{K} \mathbb{I}\{i \in [k]_{\mathrm{typ}}\} \sum_{j=1}^{H-1} (a) = \sum_{i=1}^{K} \mathbb{I}\{i \in [k]_{\mathrm{typ}}\} \sum_{j=1}^{H-1} \sqrt{\frac{2\mathbb{Q}_j^{\pi_i}(s_{i,j})L}{N_i(s_{i,j})}}$$

$$\leqslant \sqrt{2L} \sqrt{\sum_{i=1}^{K} \sum_{j=1}^{H-1} \mathbb{Q}_j^{\pi_i}(s_{i,j})} \sqrt{\sum_{i=1}^{K} \mathbb{I}\{i \in [k]_{\mathrm{typ}}\} \sum_{j=1}^{H-1} \frac{1}{N_i(s_{i,j})}} \tag{76}$$

$$\leqslant \sqrt{2L} \sqrt{HT + 2\sqrt{H^4 TL} + \frac{4}{3} H^3 L \sqrt{SG}} \tag{77}$$

$$\leqslant \sqrt{4HSTLG}, \tag{78}$$

where Eq. (76) is obtained by applying Cauchy-Schwarz's inequality, Eq. (77) follows from the application of Lemma D.7 and of a pigeonhole argument, and Eq. (78) holds under $[k]_{\mathrm{typ}}$.

By applying this bound, together with Lemmas D.3, D.9, D.10, D.11, the condition of $[k]_{\text{typ}}$, accounting for the regret of non-typical episodes, and rearranging terms, we get that:

$$U_{K,1} \leqslant 256\sqrt{HSAT_k LG} + 117392H^3 S^3 A2^A L^2 G + 113\sqrt{H^2 S^2 ALG^2 U_{K,1}}. \tag{79}$$

By letting:

$$\begin{aligned}
\alpha &= 256\sqrt{HSATLG} + 117392H^3 S^3 A2^A L^2 G, \\
\beta &= 113\sqrt{H^2 S^2 ALG^2},
\end{aligned} \tag{80}$$

we can solve for $U_{K,1}$ as $U_{K,1} \leqslant 2\alpha + \beta^2$, and obtain the following bound:

$$U_{K,1} \leqslant 512\sqrt{HSATLG} + 247533H^3 S^3 A2^A L^2 G,$$

which we can plug into Eq. (75) to get the following upper bound on the regret:

$$R_{\mathsf{PS}}(\text{S-UCBVI}, T) \leqslant 512\sqrt{HSATLG} + 498^2 H^3 S^3 A2^A L^2 G.$$

Considering now stage-dependent state transitions, we can address this generalization by considering a new Sleeping MDP with state space $\mathcal{X}$ such that $|\mathcal{X}| = SH$. As such, the regret in this case is upper bounded by:

$$R_{\mathsf{PS}}(\text{S-UCBVI}, T) \leqslant \mathcal{O}\left(\sqrt{H^2 SATLG} + H^6 S^3 A2^A L^2 G\right) =: \text{UB}_{\mathsf{PS}}(\delta),$$

w.p. at least $1 - \delta$.

To obtain an upper bound for the expected regret in the case of stage-independent transitions, we select $\delta = 2^A/T$, and obtain that, if $T \geqslant 2^A$, it holds that:

$$\begin{aligned}
\mathbb{E}\left[R_{\mathsf{PS}}(\text{S-UCBVI}, T)\right] &= \mathbb{E}\left[R_{\mathsf{PS}}(\text{S-UCBVI}, T)\mathbb{I}\{R_{\mathsf{PS}}(\text{S-UCBVI}, T) \leqslant \text{UB}_{\mathsf{PS}}(\delta)\}\right] \\
&\quad + \mathbb{E}\left[R_{\mathsf{PS}}(\text{S-UCBVI}, T)\mathbb{I}\{R_{\mathsf{PS}}(\text{S-UCBVI}, T) > \text{UB}_{\mathsf{PS}}(\delta)\}\right] \\
&\leqslant \text{UB}_{\mathsf{PS}}(\delta) + T\delta \tag{81} \\
&\leqslant 512\sqrt{HSAT \log\left(80HS^2 AT^2\right)^2 \log(HSAT)} \\
&\quad + 498^2 H^3 S^3 A2^A \log(80HS^2 AT^2)^2 \log(HSAT) + 2^A, \tag{82}
\end{aligned}$$

where Equation (81) follows by applying the high probability regret upper bound and by observing that $R_{\mathsf{PS}}(\text{S-UCBVI}, T) \leqslant T$. Moving to stage-dependent state transitions, we can generalize as above and rewrite Equation (82) as:

$$\begin{aligned}
\mathbb{E}\left[R_{\mathsf{PS}}(\text{S-UCBVI}, T)\right] &\leqslant 512H\sqrt{SAT \log\left(80H^3 S^2 AT^2\right)^2 \log(H^2 SAT)} \\
&\quad + 498^2 H^6 S^3 A2^A \log(80H^3 S^2 AT^2)^2 \log(H^2 SAT) + 2^A \\
&= \tilde{\mathcal{O}}\left(H\sqrt{SAT} + H^6 S^3 A2^A\right). \tag{83}
\end{aligned}$$

Finally, we observe that whenever $T \geqslant \Omega(H^{10} S^5 A2^{2A})$, the upper bound of the expected regret is of order $\tilde{\mathcal{O}}(H\sqrt{SAT})$, thus concluding the proof. $\qquad\square$

# E. Numerical Validation

In this appendix, we propose the *StochasticFrozenLake* setting and numerically validate our `S-UCBVI` against `UCBVI`, showing the efficacy of exploiting the knowledge of action availability. The code to reproduce the experiments is available at https://github.com/marcomussi/SleepingRL.

**Setting.** The StochasticFrozenLake environment is a modification of the well-known FrozenLake to allow holes in the lake to open and close stochastically, effectively limiting the action availability of the agent stochastically during the episode. The probability of a cell of the grid being a hole at any given stage is denoted via parameter $p$, except for the goal cell and the cell in which the agent is located at the beginning of the stage, which cannot be holes. We vary the probability of holes in the lake as $p \in \{0, 0.5, 0.75\}$ and the grid size of the lake as $G \in \{2, 3, 4\}$. We consider a horizon $H = 10$ to ensure that the agent can reach the goal. We consider $K = 2 \cdot 10^5$ episodes, and we compare `S-UCBVI` and `UCBVI` in terms of *instantaneous reward* averaged over 5 runs, with a 95% confidence interval. We also report the optimum computed apriori for reference.

**Results.** The results of the experiment are reported in Figure 7. We observe that, when $p = 0$, i.e., there are no holes in the lake, both `S-UCBVI` and `UCBVI` manage to achieve the optimum instantaneous reward. As $p$ and $G$ increase, we observe that `S-UCBVI` manages to achieve the optimum, whereas `UCBVI` settles to a suboptimal value, with the gap between the two algorithms increasing in a directly proportional manner w.r.t. the two parameters.

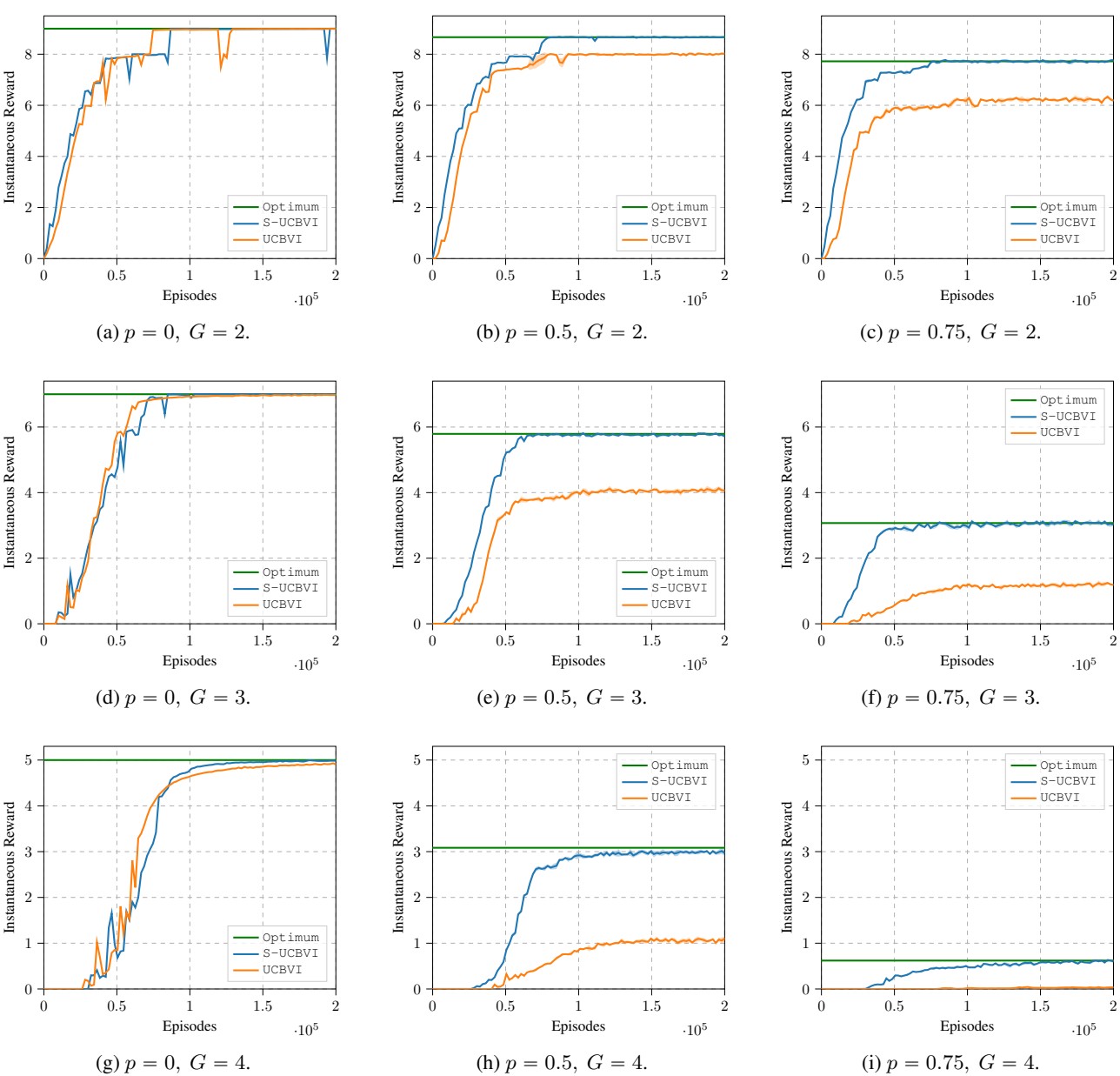

*Figure 7.* Performances in terms of instantaneous reward in the *StochasticFrozenLake* environment with horizon $H = 10$, number of episodes $K = 2 \cdot 10^5$, hole probability $p \in \{0, 0.5, 0.75\}$ and grid size $G \in \{2, 3, 4\}$ (5 runs, mean $\pm$ 95% C.I.).

