# OpenReview forum: "Sleeping Reinforcement Learning"
_ICML.cc/2025/Conference — ICML 2025 poster_

### Official Review · Reviewer_EnYm · 2025-03-13

**Overall Recommendation:** 4

**Summary:**

This paper considers a tabular episodic reinforcement learning setting where the set of available actions is not fixed but varying over episodes, states and time steps. The paper studies two different ways the available actions are revealed to the learner: per-episode (available actions are revealed at the beginning of each episode) and per-stage (available actions are revealed only at each time step within an episode). In the per-episode regime, the paper proposes an algorithm that works for both adversarial and stochastic cases, and proves that its sleeping regret is not larger (in big-O notation) than standard RL. In the per-stage regime, the paper proves both lower and upper bounds in this regime for two different types of action distributions - one where the action availability is independent of past states and actions and one where the action availability is dependent on the previous state and action.

**Claims And Evidence:**

This is a theory paper. All theorems are clearly stated and accompanied by their proofs in the appendix.

**Essential References Not Discussed:**

I am not aware of any missing important references.

**Experimental Designs Or Analyses:**

The paper has no experiments.

**Methods And Evaluation Criteria:**

The approach is sound and the results are significant. There are two different notions of regret, one for the per-episode regime (Definition 3.1) and one for the per-stage regime (Definition 4.2). These notions of regret make sense.

**Other Comments Or Suggestions:**

No other comments.

**Other Strengths And Weaknesses:**

No other comments.

**Questions For Authors:**

1. Could the authors outline any significant technical challenges for the stochastic per-stage regime where the *type* of the distribution of the action availability is unknown? In particular, why doesn't your current estimation method for $C^{ind}$ works for $C^{Markov}$? The distribution does *not* depend on the episode, so couldn't we simply estimate the conditional probability that an action might be available given the current state, previous state, previous action and previous availability?

2. Going back to the exponential dependency on $A$: the lower bound construction in Theorem 4.2 requires the availability to change based on previous availability $\mathcal{A}\_{k,h-1}$. Do you think the exponential dependency on A can be removed if the availability depends only on $s\_{k,h}, s\_{k, h-1}$ and $a\_{k,h-1}$?

**Relation To Broader Scientific Literature:**

The paper extends the existing literature on sleeping bandits to reinforcement learning. Sleeping reinforcement learning is much harder than sleeping bandits, so the contributions of the paper are significant. The paper also make novel technical contributions in deriving optimistic algorithms for sleeping reinforcement learning. I find it interesting that such an optimism-based approach works (in the per-episode regime) even for adversarial action availability, which was not the case for sleeping adversarial bandits (e.g. Nguyen and Mehta, AISTATS 2024).

In the independent and stochastic per-stage regime, the paper proposes an approach that estimates the distribution of action availability. This approach is similar to existing approaches in sleeping bandits with stochastic availabilities (e.g. Saha et al, ICML 2020).

The most surprising result to me is that on both independent and Markovian stochastic per-stage regimes, the dependency on the number of actions is exponential. This was not the case in sleeping bandits.

**Theoretical Claims:**

For the upper bounds, I only skimmed their proofs in the appendix. Since the algorithms are based on the standard approach of optimism, all the upper bound proofs seem fine.

For the lower bounds, I did carefully check the proofs. In particular, I checked the proof of Theorem 4.2, which is the most important lower bound in this paper. I found the proof correct.

---

> ### Author Rebuttal · Authors · 2025-03-31
>
> We thank the Reviewer for the time spent reviewing our work, and for having appreciated the significativeness of our work. We also thank the Reviewer for the comments on the results which we will use to expand the discussion on the results exploiting the additional page. Below, our answer to the Reviewer's questions.
>
> > Could the authors outline any significant technical challenges for the stochastic per-stage regime where the type of the distribution of the action availability is unknown? In particular, why doesn't your current estimation method for $C^{ind}$ works for $C^{markov}$? The distribution does not depend on the episode, so couldn't we simply estimate the conditional probability that an action might be available given the current state, previous state, previous action and previous availability?
>
> We can adapt the design of the estimator for $C^{ind}$ to handle $C^{markov}$ by incorporating **conditional probabilities**, as the Reviewer correctly suggests. However, we conjecture that the resulting **regret bound** would remain **of the same order (with exponential dependence on $A$)** as in our current approach (Theorem 4.1), which relies on the **augmented MDP** approach.
> That said, we believe our method allows us to avoid several non-strictly necessary calculations. We will add a comment on that in the paper.
>
> > Going back to the exponential dependency on $A$: the lower bound construction in Theorem 4.2 requires the availability to change based on previous availability $\mathcal{A}\_{k,h-1}$. Do you think the exponential dependency on A can be removed if the availability depends only on $s_{k,h}, s_{k,h-1}$ and $a_{k,h-1}$?
>
> Yes, this dependency can be removed in the described scenario and the regret bound we would obtain should be **very similar** to the **independent availability** scenario, as the additional complexity in the Markovian case is generated, as the Reviewer properly noticed by looking at the lower bound construction, by the challenge in estimating the transition probabilities over the action sets. We will add a comment on that using the additional page. Thank you for pointing it out.

---

### Official Review · Reviewer_nkFn · 2025-03-13

**Overall Recommendation:** 3

**Summary:**

The paper introduces Sleeping Reinforcement Learning (SleRL), a new reinforcement learning paradigm where the set of available actions varies over time due to external constraints or stochastic processes.  Two settings are considered: per-episode disclosure where available actions for all states are revealed at the start of each episode and per-stage disclosure where available actions are revealed only at the current step. They show an upper bound and a lower bound for the regret of a modification of UCBVI in each setting.

**Claims And Evidence:**

The regret bounds are supported by proofs.

**Essential References Not Discussed:**

This paper has discussed essential references.

**Experimental Designs Or Analyses:**

No experimental designs.

**Methods And Evaluation Criteria:**

The algorithm is a modification of a classical algorithm called UCBVI,
and it makes sense for this problem.

**Other Comments Or Suggestions:**

1. In the definition of three regrets, is there a typo? I guess $T$ and $K$ should be the same thing, the number of episodes.
2. It would be better to show some numerical results to verify the bounds.

**Other Strengths And Weaknesses:**

Strengths:

1. This paper is a pure theoretical paper  and offers detailed analysis about the upper bound and lower bound for the algorithm.
2. This paper uses a novel construction (Figure 3) to show that an exponential dependence on the
number of actions $A$ is unavoidable in the regret and proves a lower bound  (Theorem 4.2).

Weaknesses:

1. This paper lacks numerical analysis to test their theoretical results.
2. The gap between upper bound and lower bound may be too loose for Markovian Per-stage Disclosure.

**Questions For Authors:**

1. Is it possible to add some numerical results to verify your results?
2. Is it possible to reduce the bound gaps, i.e. the exponential term $2^{A}$ and $2^{A/2}$, between Theorem 4.1 and 4.2?
3. Is it possible to reduce the lower bound condition for $T$ with a factor $H^{10}$ in Theorem 5.1?

## update after rebuttal: I raise the score from 2 to 3. Please refer to the comment below for reason.

**Relation To Broader Scientific Literature:**

This work is related to Multi-Armed Bandits under the name of “Sleeping”
MABs and Reinforcement Learning with constrained action spaces.

**Theoretical Claims:**

I have roughly checked the proof of Theorem 3.1 and it looks correct.

---

> ### Author Rebuttal · Authors · 2025-04-01
>
> We thank the Reviewer for the time spent reviewing our work. Below, our answer to the Reviewer's questions and concerns.
>
> > Is it possible to add some numerical results to verify your results?
>
> To numerically validate the algorithm and show the impact of action availability on performance, we consider a modification of the well-known Frozen Lake environment (https://gymnasium.farama.org/environments/toy_text/frozen_lake/). In the original version, the agent must traverse a frozen lake (i.e., a grid) from a start to a goal position, avoiding holes in the surface (i.e., unavailable positions in the grid). We modify it by letting such holes open and close stochastically (i.e., the unavailability of each position is sampled at each time step from a Bernoulli with parameter $p$).
> We assume the start and goal to be fixed in the top-left and bottom-right corners of the grid, respectively.
>
> The agent can move up, down, left, right, or stay. If the agent selects an action that would move it to an unavailable state, it stays in the current position.
> The reward function is defined as follows: the agent receives reward $1$ for state-action pairs that have the goal as next state, and $0$ otherwise. The episode stops at the end of the time horizon (not when we reach the goal state).
> Given that all positions (except the start, the goal, and the position occupied by the agent) have the same probability of being frozen at any time step, it is easy to see that the optimal policy tries to move along the diagonal that connects the start to the goal.
> To compute the value of the optimal policy, we perform a Montecarlo simulation for each evaluated setting.
> We compare our S-UCBVI (Algorithm 7) with standard UCBVI (Azar et al., 2017), observing that the latter interacts with the environment as shown in Figure 1. We compare them in terms of reward, to highlight that both algorithms converge, yet UCBVI converges to a suboptimal objective.
>
> We evaluate grids of size $G \times G$, with $G \in \\{2,3,4\\}$, varying the stochasticity parameter as $p \in \\{0,0.5,0.75\\}$. We consider a time horizon of $H=10$ for each episode and $K= 2 \cdot 10^5$ episodes.
>
> The Reviewer can find the plots of the experiments here:
>
> https://drive.google.com/file/d/19WlZpKUHoSoYxx4PgT3jNLqyfmrm9twu/view?usp=sharing
>
> and the code here:
>
> https://drive.google.com/file/d/1NBoDjr9P_eaUcO1Na4nyzE71q4zgAKwR/view?usp=sharing
>
> As expected, when $p=0$, we observe no performance difference. Instead, as the environment stochasticity increases, we observe a greater gap in the performances of the two algorithms, with S-UCBVI (our algorithm) achieving the optimal value (obtained via Montecarlo simulations as described above), represented as an horizontal line, and UCBVI not reaching such optimum. This is due to the fact that UCBVI cannot observe action availabilities, effectively playing in an environment with a lower optimum.
>
> > Is it possible to reduce the bound gaps [...] between Theorem 4.1 and 4.2?
>
> The lower bound of Theorem 4.2 that shows an exponential dependence in the cardinality of the action set represents a **statistical barrier** in the learning for Sleeping MDPs with **Markovian** per-stage availability. For this reason, as common in the literature (e.g., MDPs with adversarial transitions [1]), the goal is no longer matching such an exponential lower bound, but rather finding **structures** (e.g., independent action availability) that overcome such barriers. The result of Theorem 4.1, indeed, has to be interpreted as just an exemplification of the fact that, when accepting such an exponential dependence, the Sleeping MDPs with Markovian per-stage availability can be addressed through the augmented MDP method. We will clarify this in the paper.
>
> [1] Tian, Y., Wang, Y., Yu, T. and Sra, S. Online Learning in Unknown Markov Games. ICML 2021
>
> > In the definition of three regrets, is there a typo? I guess $T$ and $K$ should be the same thing, the number of episodes.
>
> We checked and the notation is correct and compliant with (Azar et al., 2017) where $T = KH$ is the total number of interactions (see Section 2), $K$ is the number of episodes and $H$ is the horizon of the single episode.
>
> > Is it possible to reduce the lower bound condition for $T$ with a factor $H^{10}$ in Theorem 5.1?
>
> Just like in (Azar et al., 2017), for the UCBVI algorithm and its analysis, it is challenging to remove such a dependence on the horizon $H$ from the minimum $T$ condition. This is exacerbated, in our case compared to (Azar et al., 2017), since we consider stage-dependent transitions (doubling the exponent of $H$). Nevertheless, there exist works [2] that succeed in mitigating such a condition at the price of more complex (and less effective in practice) algorithms. Importing such techniques to the Sleeping MDP setting can be an interesting future work.
>
> [2] Zhang, Z., Chen, Y., Lee, J. D., and Du, S. S. Settling the sample complexity of online reinforcement learning. COLT 2024.

---

> > ### Comment · Reviewer_nkFn · 2025-04-04
> >
> > I appreciate your responses for my questions.  The numerical experiment shows the advantage of your algorithm over UCBVI, and your responses answer my questions about the regret bound. I have adjusted my score accordingly.

---

### Official Review · Reviewer_iuwR · 2025-03-13

**Overall Recommendation:** 3

**Summary:**

This paper studies a new paradigm called Sleeping Reinforcement Learning, where the available action set varies during the interaction with the environment. The authors study several settings, including the per-episode disclosure, in which the available action sets are revealed at the beginning of each episode, and the per-stage disclosure, in which the available actions are disclosed only at each decision stage. The authors provide algorithms, upper and lower bounds for the problem.

**Claims And Evidence:**

Yes.

**Essential References Not Discussed:**

No.

**Experimental Designs Or Analyses:**

No.

**Methods And Evaluation Criteria:**

Yes.

**Other Comments Or Suggestions:**

There is a typo in Line 195.

**Other Strengths And Weaknesses:**

Strengths:

1. This is the first work to theoretically study the important sleeping RL setting. It is well motivated, the authors provide many examples to show the importance of this setting.
2. The paper is well-written and easy-to-follow.
3. The authors conduct a complete study in this topic, providing algorithms with regret upper bounds, and proving novel lower bounds.

Weaknesses:

1. It may be better to also provide a lower bound for the setting of independent per-stage disclosure.
2. In Lines 130-132, $V^*_{ED}(A)$, $V^*_{SD}(A)$, $V^*_{LLC}(A)$ are used before definition.

**Questions For Authors:**

It is possible to also provide a lower bound for the setting of independent per-stage disclosure? I believe this may help better understand the problem.

**Relation To Broader Scientific Literature:**

The work is related to the works on sleeping bandits, where the action sets vary in time. To the best of my knowledge, this is the first theoretical work on sleeping RL.

**Theoretical Claims:**

No.

---

> ### Author Rebuttal · Authors · 2025-03-31
>
> We thank the Reviewer for the time spent reviewing our paper and for having appreciated the motivation, clarity and novelty of our work. Below, our answer to the Reviewer's comments.
>
> > It may be better to also provide a lower bound for the setting of independent per-stage disclosure.
>
> > It is possible to also provide a lower bound for the setting of independent per-stage disclosure? I believe this may help better understand the problem.
>
> We thank the Reviewer for giving us the opportunity to elaborate on this. After careful consideration, we argue that a **tight lower bound for the independent per-stage disclosure case is the same as that of standard MDPs, i.e.,  $\mathbb{E}[R(T)] \ge {\Omega}(H\sqrt{SAT})$** (see Domingues et al., 2021). Clearly, the lower bound for standard MDPs is a lower bound for the independent per-stage disclosure case, since the latter includes a smaller class of problems in which all actions are always available, i.e., $C^{\text{ind}}_h(\mathcal{A}|s) = 1$ for every $(s,h) \in \mathcal{S}\times [H]$. Concerning the tightness, we need to carefully compare it with the upper bound of Theorem 5.1. First of all, the lower bound $\mathbb{E}[R(T)] \ge {\Omega}(H\sqrt{SAT})$ (Domingues et al., 2021) is for the **expected regret** (as customary in the literature), while our upper bound for the per-stage disclosure case of Theorem 5.1 is for the **regret in high probability**. Thus, we need to convert the latter to the expected regret in order to perform the comparison. To do so, we start from the first regret bound of Theorem 5.1 (disregarding constants and logarithmic terms, but not terms depending on $\delta$):
> $$
>    R(T) \le  \text{UB}(\delta) := \widetilde{O} \left( H \sqrt{SAT \log\left(\frac{2^A}{\delta} \right)} + H^6S^3A 2^A \log\left(\frac{2^A}{\delta} \right)^2 \right), \qquad \text{w.p.} \quad 1-\delta.
> $$
> Notice that by disregarding the dependence on $\delta$ and for sufficiently large $T$, we obtain the second regret bound of Theorem 5.1 (still in high probability). To get the upper bound for the expected regret, we make the choice of $\delta = \frac{2^A}{T}$, whenever $T \ge 2^A$, obtaining (since $R(T) \le T$ always):
> $$
> \begin{aligned}
>     \mathbb{E}[R(T)] & = \mathbb{E}[R(T) \mathbb{1}\\{R(T) \le \text{UB}(\delta) \\}] +  \mathbb{E}[R(T) \mathbb{1}\\{R(T) > \text{UB}(\delta) \\}] \\\\
>     & \le\text{UB}(\delta) + T \delta \\\\
>     &  \le \widetilde{O} \left( H \sqrt{SAT \log\left(T \right)} + H^6S^3A 2^A \log\left(T \right)^2 + 2^A\right) \\\\
>     & = \widetilde{O} \left( H \sqrt{SAT } + H^6S^3A 2^A \right).
>     \end{aligned}
> $$
> The latter is of order  $\widetilde{O} \left( H \sqrt{SAT }\right)$ whenever $T \ge \Omega(H^{10}S^5A2^{2A})$. We will adjust the presentation of the result in the final version of the paper, clarifying the difference between expected regret and regret with high probability bounds, and, consequently, the discussion on the result and the future works. Thank you for rising this point.
>
> > In Lines 130-132, $V_{ED}^* (A)$, $V_{SD}^*(A)$, $V_{LLC}^*(A)$ are used before definition.
>
> We agree with the Reviewer that (at least) an informal definition is needed also in this part. It was an oversight, we fixed it. Thank you.
>
> > Typo in Line 195.
>
> Thanks, we fixed it.

---

### Decision · Program_Chairs · 2025-05-01

**Decision:**

Accept (poster)

**Comment:**

This paper introduces Sleeping Reinforcement Learning, in which available actions vary over time. It provides a theoretical analysis for this problem, along with algorithms with regret guarantees and matching lower bounds. While minor concerns were raised regarding exponential dependencies and missing lower bounds in specific cases, the rebuttal and added experiments addressed these concerns. Reviewers have converged on the significance of the contribution.